# Simultaneous measurement of $\delta^{13}C$, $\delta^{18}O$ and $\delta^{17}O$ of atmospheric $CO_2$ - Performance assessment of a dual-laser absorption spectrometer

Pharahilda M. Steur[1], Hubertus A. Scheeren[1], Dave D. Nelson[2], J. Barry McManus[2], and Harro A. J. Meijer[1]

[1]Centre for Isotope Research, University of Groningen, Nijenborgh 6, 9747 AG Groningen, The Netherlands
[2]Aerodyne Research Inc., 45 Manning Road, Billerica, MA 01821-3976, USA

**Correspondence:** P.M. Steur (p.m.steur@rug.nl)

**Abstract.** Using laser absorption spectrometry for the measurement of stable isotopes of atmospheric $CO_2$ instead of the traditional Isotope Ratio Mass Spectrometry method decreases sample preparation time significantly, and uncertainties in the measurement accuracy due to $CO_2$ extraction and isobaric interferences are avoided. In this study we present the measurement performance of a new dual-laser instrument developed for the simultaneous measurement of the $\delta^{13}C$, $\delta^{18}O$ and $\delta^{17}O$ of atmospheric $CO_2$ in discrete air samples, referred to as the Stable Isotopes of $CO_2$ Absorption Spectrometer (SICAS). We compare two different calibration methods: the ratio method based on measured isotope ratio and a $CO_2$ mole fraction dependency correction, and the isotopologue method based on measured isotopologue abundances. Calibration with the ratio method and isotopologue method is based on three different assigned whole air references calibrated on the VPBD and the WMO 2007 scale for their stable isotope compositions and their $CO_2$ mole fractions, respectively. An additional quality control tank is included in both methods to follow long-term instrument performance. Measurements of the quality control tank show that the measurement precision and accuracy of both calibration methods is of similar quality for $\delta^{13}C$ and $\delta^{18}O$ measurements. During one specific measurement period the precision and accuracy of the quality control tank reach WMO compatibility requirements, being 0.01‰ for $\delta^{13}C$ and 0.05‰ for $\delta^{18}O$. Uncertainty contributions of the scale uncertainties of the reference gases add another 0.03 and 0.05‰ to the combined uncertainty of the sample measurements. Hence, reaching WMO compatibility for sample measurements on the SICAS requires reduction of the scale uncertainty of the reference gases used for calibration. An inter-comparison of flask samples over a wide range of $CO_2$ mole fractions has been conducted with the Max Planck Institute for Biogeochemistry resulting in a mean residual of 0.01 and -0.01‰ and a standard deviation of 0.05 and 0.07‰ for the $\delta^{13}C$ measurements calibrated using the ratio method and the isotopologue method, respectively. The $\delta^{18}O$ could not be compared due to depletion of the $\delta^{18}O$ signal in our sample flasks because of too long storage times. Finally, we evaluate the potential of our $\Delta^{17}O$ measurements as a tracer for gross primary production by vegetation through photosynthesis. Here, a measurement precision of $<0.01‰$ would be a prerequisite for capturing seasonal variations in the $\Delta^{17}O$ signal. Lowest standard errors for the $\delta^{17}O$ and $\Delta^{17}O$ of the ratio method and the isotopologue method are 0.02 and 0.02‰, and 0.01 and 0.02‰, respectively. The accuracy results show consequently too enriched results for both the $\delta^{17}O$ and $\Delta^{17}O$ measurements for both methods. This is probably due to the fact that two of our reference gases were not measured directly,

but were determined indirectly. The ratio method shows residuals ranging from 0.06 to 0.08‰ and from 0.06 to 0.1‰ for the $\delta^{17}O$ and $\Delta^{17}O$ results, respectively. The isotopologue method shows residuals ranging from 0.04 to 0.1‰ and from 0.05 and 0.13‰ for the $\delta^{17}O$ and $\Delta^{17}O$ results, respectively. Direct determination of the $\delta^{17}O$ of all reference gases would improve the accuracy of the $\delta^{17}O$, and thereby of the $\Delta^{17}O$ measurements.

## 1    Introduction

As atmospheric $CO_2$ (atm-$CO_2$) is the most important contributor to anthropogenic global warming, keeping track of its sources and sinks is essential for understanding and predicting the consequences of climate change for natural systems and societies, and for assessing and quantifying the possible mitigating measures. The stable isotope (si) composition of atm-$CO_2$ is often used as an additional tool to distinguish between anthropogenic emissions and the influence of the biosphere on varying $CO_2$ mole fractions (Pataki et al., 2003; Zhou et al., 2005). For this reason, the si composition of atm-$CO_2$ is monitored at a

considerable number of atmospheric measurement stations around the globe. Due to the large size of the carbon reservoir of the atmosphere and the long lifetime of $CO_2$ in the atmosphere, the effects of sources and sinks on the atmospheric composition are heavily diluted. Changes in the isotope composition of atm-$CO_2$ are therefore relatively small compared to the actual changes in carbon fluxes (IAEA, 2002). Hence, current climate change- and meteorological research, as well as the monitoring of $CO_2$ emissions, require accurate and precise greenhouse gas measurements that can meet the WMO/GAW inter-laboratory

compatibility goals of 0.01‰ for $\delta^{13}C$ and 0.05‰ for $\delta^{18}O$ of atm-$CO_2$ for the Northern Hemisphere (Crotwell et al., 2020).

Traditionally, high precision stable isotope measurements are done using Isotope Ratio Mass Spectrometry (IRMS) (Roeloffzen et al., 1991; Trolier et al., 1996; Allison and Francey, 1995) which requires extraction of $CO_2$ from the air sample before a measurement is possible. This is a time-consuming process wherein very strict, 100% extraction procedures need to be applied to avoid isotope fractionation and to prevent isotope exchange of $CO_2$ molecules with other gases or water. Extraction of $CO_2$

from air is a major contributor to both random and systematic scale differences between laboratories and thus complicates the comparison of measurements (Wendeberg et al., 2013). Further, due to the isobaric interferences of both different $CO_2$ isotopologues and $N_2O$ molecules, which are also trapped with the (cryogenic) extraction of $CO_2$ from air, corrections need to be applied for the determination of the $\delta^{13}C$ and $\delta^{18}O$ values. Due to the mass interference of the $^{12}C^{17}O^{16}O$ isotopologue with $^{13}C^{16}O^{16}O$ (and to a lesser extent $^{13}C^{17}O^{16}O$ and $^{12}C^{17}O^{17}O$ with $^{12}C^{18}O^{16}O$), the $\delta^{13}C$ results need a correction (usually

referred to as "ion correction") that builds upon an assumed fixed relation between $\delta^{17}O$ and $\delta^{18}O$. This assumed relation has varied in the past (Santrock et al., 1985; Allison et al., 1995; Assonov and Brenninkmeijer, 2003; Brand et al., 2010) giving rise to again systematic differences (and confusion) between laboratories. Determination of the $\delta^{17}O$ of $CO_2$ samples itself using IRMS is extremely complex, due to the mass overlap of the $^{13}C$ and $^{17}O$ containing isotopologues, and can only be done using very advanced techniques restricted to just a few laboratories at the moment (see Adnew et al., 2019, and references therein).

As the $\delta^{17}O$ in addition to the $\delta^{18}O$ values in atmospheric $CO_2$ have the potential to be a tracer for gross primary production and anthropogenic emissions (Laskar et al., 2016; Luz et al., 1999; Koren et al., 2019), a less labor-intensive method that would enable to analyze all three stable isotopologues of atm-$CO_2$ at a sufficient precision would be an asset.

Optical (infrared) spectroscopy now offers this possibility following strong developments in recent years in FTIR and especially for the laser light sources, to perform isotopologue measurements showing precisions close to, or even surpassing IRMS measurements (Tuzson et al., 2008; Vogel et al., 2013; McManus et al., 2015). The technique was developed in the 1990s to a level where useful isotope signals could be measured, first on pure compounds such as water vapour (Kerstel et al., 1999), and soon also directly on $CO_2$ in dry whole air samples (Becker et al., 1992; Murnick and Peer, 1994; Erdélyi et al., 2002; Gagliardi et al., 2003). Extraction of $CO_2$ from the air can therefore be avoided and smaller sample sizes suffice. Finally, optical spectroscopy is truly isotopologue-specific and is thus free of isobaric interferences, hence giving the possibility to directly measure the $\delta^{17}O$ in addition to the $\delta^{13}C$ and $\delta^{18}O$. Recent studies already showed the effectiveness of optical spectroscopy for the measurement of $\delta^{17}O$ in pure $CO_2$ for various applications (Sakai et al., 2017; Stoltmann et al., 2017; Prokhorov et al., 2019).

In this paper we present the performance, in terms of precisions and accuracy, of an Aerodyne dual laser optical spectrometer (CW-IC-TILDAS-D) in use since September 2017, for the simultaneous measurement of $\delta^{13}C$, $\delta^{18}O$ and $\delta^{17}O$ of atm-$CO_2$, which we refer to as "Stable Isotopes of $CO_2$ Absorption Spectrometer" (SICAS). The instrument performance over time is discussed, followed by an analysis of the $CO_2$ mole fraction dependency of the instrument. We report $CO_2$ mole fractions in $\mu mol/mol$, also referred to as ppm. The actual ways of performing a calibrated measurement using either individual isotopologue measurements or isotope ratios is discussed and whole air measurement results of both calibration methods are evaluated for their compatibility with IRMS stable isotope measurements. Conclusively, the usefulness of the triple oxygen isotope measurements for capturing signals of atmospheric $CO_2$ sources and sinks is evaluated.

## 2 Instrument description

### 2.1 Instrumental set-up

The optical bench as depicted in figure 1 includes the two lasers, several mirrors to combine and deflect the laser beams, the optical cell and two detectors. The two interband cascade lasers (ICL) (Nanoplus GmbH, Germany) operate in the mid infrared region (MIR). The isotopologues that are measured are $^{12}C^{16}O_2$, $^{13}C^{16}O_2$, $^{12}C^{16}O^{18}O$ and $^{12}C^{16}O^{17}O$, which from now on will be indicated as 626, 636, 628 and 627 respectively, following the HITRAN database notation (Gordon et al., 2017). Application of a small current ramp causes small frequency variations so the lasers are swept (with a sweep frequency of 1.7kHz) over a spectral range in which ro-vibrational transitions of the isotopologues occur with similar optical depths (Tuzson et al., 2008). Laser 1 operates in the spectral range of 2350 cm-1 (4.25 $\mu$m) for measurement of 627 (and 626) and laser 2 operates around 2310 cm-1 (4.33 $\mu$m) for the measurement of 626, 636 and 628. The lasers are thermoelectrically cooled and stabilized to temperatures of -1.1°C and 9.9°C, respectively. The beams are introduced in a multi-pass aluminum cell with a volume of 0.16 L in which an air sample is present at low pressure ($\sim$50 mbar). The total path length of the laser light in the optical cell is 36 meters.

After passing the cell, the lasers are led to a thermoelectrically cooled infrared detector, measuring the signal from the lasers in the spectral range (figure 2). The lasers, optical cell and detectors are all in a housing that is continuously flushed with $N_2$ gas

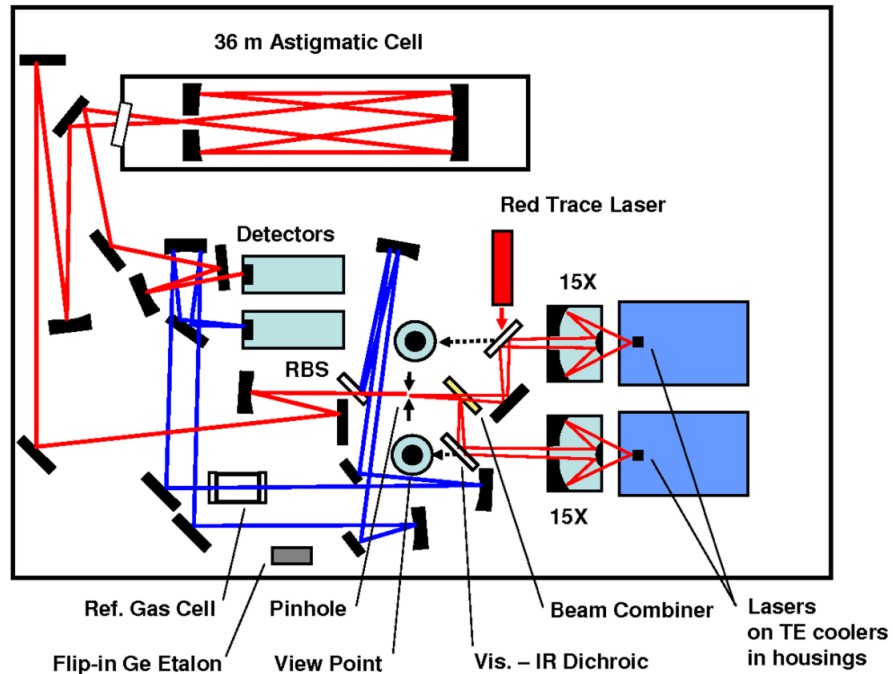

**Figure 1.** Scheme of the optical board of the SICAS (figure adapted from McManus et al., (2015)). Two pathways are shown, both consisting of signals from both lasers: the sample measurement beam in red and the reference beam in blue. The reference pathway is in our case only used for fitting purposes. RBS stands for reference beam splitter. One of the detectors is used to read the signal of the sample beam, the other for the reference beam. The red trace laser is co-aligned with the sample path to visualize the sample pathway to ease alignment.

to avoid any other absorption by $CO_2$ than from gas in the optical cell. The temperature within the housing is controlled using a re-circulating liquid chiller set at a temperature of 20°C to keep the temperature in the cell stable. Within a measurement sequence (~12 hours) the temperature does normally not fluctuate more than 0.05°C. The absorption spectra are derived by the software TDLWintel (McManus et al., 2005) that fits the measured signal based on known molecular absorption profiles from

the HITRAN database (Rothman et al., 2013). On basis of the integration of the peaks at the specific wavelengths, measured pressure and temperature in the optical cell and the constant path length, the isotopologue mole fractions are calculated by the TDLWintel software with an output frequency of 1Hz. For convenience, the default output for the isotopologue mole fractions are scaled for 'the natural abundances' of the 626, 636, 628 and 627 as defined in Rothman et al. (2013), but for obtaining the raw mole fractions this scaling is avoided.

The gas inlet system, depicted in figure 3, is designed to measure discrete air samples in static mode, such that one can quickly switch between measurements of different samples. The system consists of Swagelok stainless steel tubing and connections and pneumatic valves. There are two inlet ports (11 and 14) which are connected to the sample cross at the heart of system (from now on indicated as inlet volume), where a sample is collected at the target pressure of 200± 0.25 mbar before

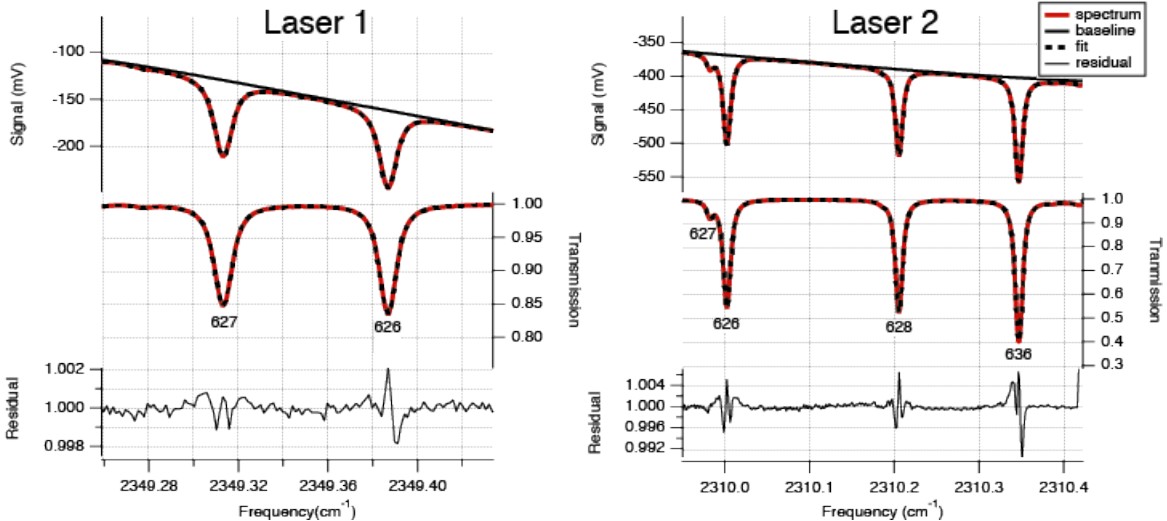

**Figure 2.** Typical absorption spectrum, transmission spectrum and residual for laser 1 (left) for measurement of 627 and 626, and laser 2 (right) for measurement of 626, 628 and 636. The residuals show systematic deviations at the line positions. These deviations are primarily due to the use of the Voigt lineshape function in the spectral fitting model, rather than a more complex lineshape function such as Hartmann-Tran. Careful analysis has shown that the use of the more convenient Voigt lineshape function does not add noise, drift or calibration error as implemented in the isotope analyser.

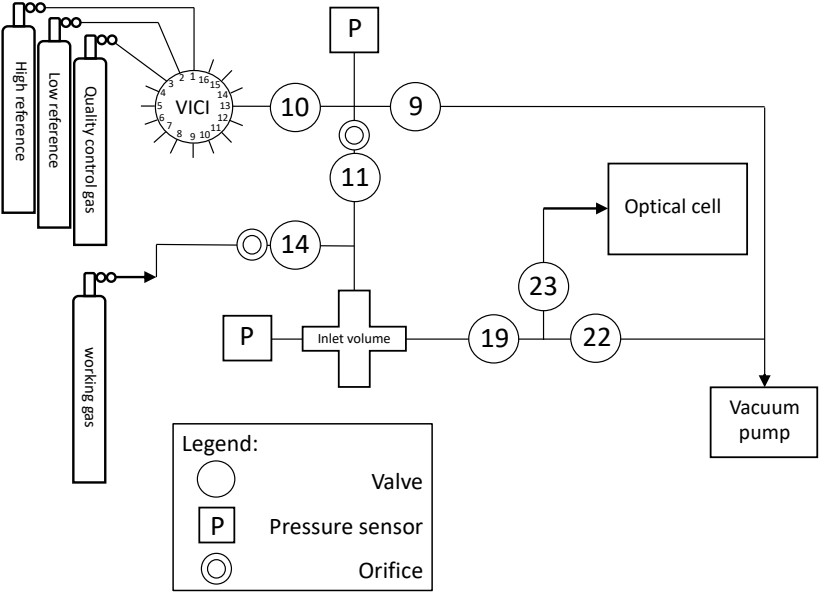

**Figure 3.** Gas inlet system of the SICAS with one VICI multivalve inlet port, connected to three high pressure natural air tanks and 12 free ports for samples. The includes an extra inlet port for the working gas, also a high pressure natural air tank.

it is connected to the optical cell. One of the inlet ports (11) is connected to a 1/8" VICI multivalve (Valco Instruments) with 15 potential positions for flask samples or cylinders. The cylinders depicted in figure 3 will be defined in section 2.2 and 3.2. When the VICI valve switches from position, the volume between port 10 and 9 is flushed 7 times with the sample gas to prevent memory effects due to the dead volume of the VICI valve. A sample gas is led into the inlet volume at reduced flow, as a critical orifice is placed right before the inlet valve, while another gas is being measured inside the optical cell. Since the closing and opening of the valves is controlled by the TDLWintel software, it also controls the duration of the flow into the inlet volume. The target pressure is reached using input from a pressure sensor placed inside the inlet volume. After evacuation of the optical cell (opening valve 22 and 23) the gas from the inlet volume can immediately be brought into the optical cell (opening valve 19 and 23) thereby reducing the sample pressure to $\sim$50 mbar.

The gas handling procedures are different for measurements of air from cylinders or flasks. For the cylinders, single stage pressure regulators are in use (Rotarex, model SMT SI220), set at an outlet pressure of 600-1000 mbar (absolute). If measurements are started after more than two days of inactivity, the internal volume of the regulators is flushed 10 times to prevent fractionation effects. To open and close the flasks we use a custom-built click-on electromotor valve system (Neubert et al., 2004), making it possible to open the flasks automatically before the measurement. Before opening the flask, the volume between valve 9 and the closed flask is evacuated so there is no need to flush extensively and less sample gas is lost. The actions described above are all steered by a command program developed by Aerodyne Research Inc. called the Switcher program. A bespoke script writing program developed in FileMaker Pro enables us to quickly write scripts for measurement sequences and to directly link those measurements to an internal database.

## 2.2 Instrument performance

The SICAS measurement performance was evaluated by determining the Allan variance of the four measured isotopologue abundances and the three isotope ratios as function of measurement time on a single whole air sample in the sealed optical cell. The isotope ratios, defined here as the ratio of the rare isotopologue (636, 628 and 627) and the most abundant 626 isotopologue, are r636, r628 and r627 [1]. This experiment was first done in September 2017 and repeated in July 2019 to see the development in time of the measurement precision (figure 4). In all cases, drifts outweigh the averaging process after time periods ranging from 16 seconds to 75 seconds, and this is short compared to the duration of the normal measurement sequences. This is a firm indication that continuous drift correction using gas from a high pressurized cylinder, of comparable $CO_2$ concentration and isotope composition as atmospheric samples is necessary for optimal results. The cylinder used for drift correction which we define as the working gas contains natural air of which the isotope composition and the $CO_2$ concentration is known.

The precision became significantly worse for all species but isotopologue 627 in the time period between September 2017 and July 2019. In this same period a gradual but significant decrease (of about 50%) in the measured laser intensity was observed. For most species this led to an increase of the optimal integration time, which is logical given the fact that the minimal precision was higher, such that the increase due to drift influences the acquired precision at a higher integration time, and also at a higher variance level. Figure 4 shows the rapid variance increase due to drift for all isotopologues after less than

---

[1]Note that the r628 and r627 differ strictly speaking from the isotope ratios (r18 and r17) by a factor 2.

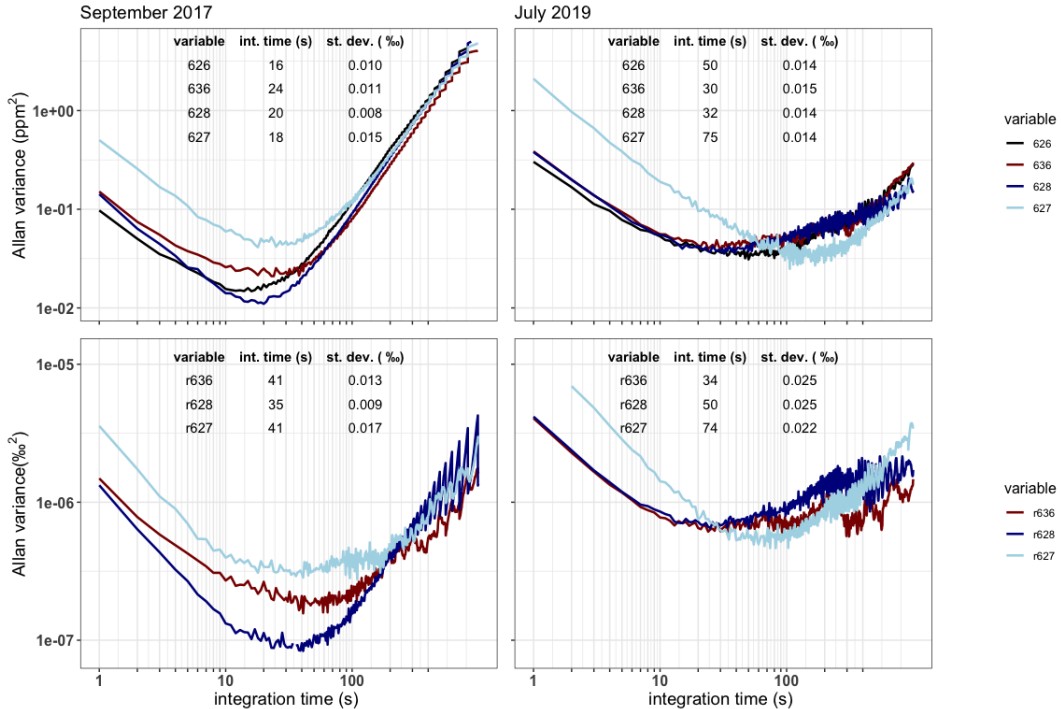

**Figure 4.** The Allan variance as a function of the integration time in seconds for a single gas measurement plotted for both the measured isotopologue abdundance (top) and the isotope ratios (bottom) at September 2017 (left) and July 2019 (right). The best achieved precisions and corresponding integration times are shown as a table in the plots.

one minute for the September 2017 measurements, and the same happens for the July 2019 measurements, only less visible due to the higher minimal variance levels.

The decreased laser intensity, potentially leading to a deteriorated signal-to-noise ratio, was caused by contamination of the
140 mirrors in the optical cell, most likely due to precipitation of ultra-fine salt-based aerosols from the sample air occurring during evacuation of the cell. The majority of flask samples measured on the SICAS are from the atmospheric measurement station Lutjewad which is located at the Northern coast of the Netherlands in a rural area dominated by cropland and grassland mainly used for dairy cows. The aerosol composition at Lutjewad is therefore expected to be dominated by sea-salt and ammonium-nitrate from agricultural emissions. Hence, we were able to clean the mirrors and retrieve $\sim 80\%$ of the original laser signal
by flushing the mirrors with ultrapure water and ethanol (in that order). This procedure, performed at the $31^{st}$ of October in 2019, deviates from the recommended mirror cleaning instructions in which it is advised to use ethanol only to clean the mirrors. The additional use of ultrapure water was in our case necessary since the precipitated aerosols were not dissolved in ethanol and were therefore not removed when we used ethanol only. Despite the increase of the laser signal due to the cleaning procedure, precisions did not improve as a consequence of it. This indicates that other, still unidentified, issues played a role in
the decrease of measurement precision.

To reduce short-term instrumental drift, all sample measurements needed to be alternated with measurements of the working gas, as then the drift corrected signal can be expressed as:

$$I_{S(t)} = \frac{M_{S(t)}}{M_{WG(t)}} \tag{1}$$

Where M stands for measurement which can be either the measured isotope ratio or isotopologue abundance, S stands for sample, WG stands for working gas, t stands for time of the sample gas measurement and I for index referring to the drift corrected sample measurement, of either the isotope ratio or the isotopologue abundance. WG(t) is the measured working gas at time t derived from the time-dependent linear regression of the measurements of the working gas bracketing the sample gas measurement. The effectiveness of this drift correction method was tested for the measured isotope ratios only. Although one of the tested calibration methods uses isotopologue abundances for the initial calibration, the isotope composition is expressed as a delta value and will therefore eventually be calculated using isotope ratios (section 3.3.3). The precision of the isotope ratios will therefore always determine the measurement precision.

The calibration procedure conducted directly on isotopologue abundances will calculate isotope ratios from the calibrated isotopologue abundances, so precision of the isotope ratios will determine the measurement precision (see section 3.3.3). A tank was measured >10 times alternately with the working gas. The relative standard deviations were calculated for n=5 and n=10, both with and without drift correction (table 1). It is expected that, if the drift correction is effective, the standard deviations of the uncorrected values are higher than the standard deviations of the corrected values. The drift correction is effective as the standard deviations of the corrected values are always lower than of the uncorrected values. Although the drift correction procedure is not perfect, as we see a small increase of the standard deviation between n=5 and n=10 between 0.005 and 0.012‰, we can still conclude that the drift correction will result in a better repeatability of the isotope ratios.

| All st. dev. | n=5 | | n=10 | |
|---|---|---|---|---|
| in ‰ | uncor | cor | uncor | cor |
| r636 | 0.04 | 0.020 | 0.06 | 0.025 |
| r628 | 0.05 | 0.021 | 0.10 | 0.029 |
| r627 | 0.06 | 0.018 | 0.18 | 0.03 |

**Table 1.** Relative standard deviations for n=5 and n=10 of uncorrected (uncor) and corrected (cor) isototope ratio sample measurements. Sample measurements were always bracketed by measurements of the working gas. Standard deviations of the uncorrected measurements only use the sample measurements, standard deviations of the corrected measurements use drift corrected (equation 1) sample measurements using the working gas measurements.

Cross-contamination, being the dilution of a small volume of the working gas in the sample aliquot that is being measured, and vice versa, as described for a Dual-Inlet IRMS in Meijer et al. (2000), will occur in the SICAS due to the continuous switching between sample and machine working gas. If cross-contamination is not corrected for DI-IRMS measurements inaccuracies can occur when samples of a highly deviating isotope composition are measured. On the SICAS only atmospheric samples are measured that are of very similar isotope values. The $CO_2$ mole fraction of the samples can deviate quite strongly

from the machine working gas, so effects of cross-contamination will have an influence on the $CO_2$ mole fraction in the optical cell. From experimental data we quantified the fraction of the preceding sample that affects a sample measurement to be max 0.01%. A sensitivity analysis was performed using this fraction and showed that this is such a small amount that scale effects due to cross-contamination are well below the precisions found in this study (for a detailed description of the analysis, see Appendix E). If samples of $CO_2$ concentrations outside the range of atmospheric samples are measured it will be essential to also take into account the surface adsorption effects of the aluminum cell which is known to adsorb $CO_2$ (Leuenberger et al., 2015). $CO_2$ adsorption in the cell of the SICAS was clearly visible as a drop of measured $CO_2$ concentration when an atmospheric sample was let into the cell right after the cell was flushed with a $CO_2$ free flush gas (hence stripped from $CO_2$ molecules sticking to the cell surface).

## 3   Calibration experiments

### 3.1   The $CO_2$ mole fraction dependency

The stable isotope composition of atmospheric $CO_2$ is expressed as a delta value on the VPDB ($^{13}C$) / VPDB-$CO_2$ ($^{17}O$ and $^{18}O$) scales, which are realized by producing $CO_2$ gas (using phosphoric acid under well-defined circumstances) from the IAEA-603 marble primary reference material (successor to the now obsolete NBS-19) (IAEA, 2016). A complication when compared to classical DI-IRMS isotope measurements (or to optical measurements of pure $CO_2$ for that matter) is that in the practice of laser absorption spectroscopy the mole fraction of $CO_2$ in a gas affects the measured stable isotope ratios (and thus delta values) of $CO_2$. Quantification, let alone elimination of this $CO_2$ mole fraction dependence (CMFD) is difficult (McManus et al., 2015), but two sources of CMFD were identified by Wen et al. (2013) and related to different calibration strategies. In the first place, CMFD results from non-ideal fitting of the absorption spectra which will to some extent always occur. Capturing the true absorption spectrum is very complicated, due to among others line broadening effects of the various components of the air, far wing overlap of distant but strong absorptions, temperature and pressure variability and the choice of lineshape function (see figure 2 and caption). Secondly, a more "trivial" CMFD is introduced when calibration is done on measured isotopologue ratios and the intercepts of the relation between the isotopologues and the $CO_2$ mole fraction is non-zero (Griffith et al., 2012). This effect can be explained by expressing the calculation of the isotopologue ratio by:

$$r* = \frac{X_*}{X_{626}} \tag{2}$$

In which $X_*$ is the measured isotopologue mole fraction and * indicates which of the rare isotopologues is used. When the relation of the measured isotopologue mole fraction and the $CO_2$ mole fraction is linear, this can be described by:

$$X_* = X_{CO2} * \alpha + \beta \tag{3}$$

When equations 3 is brought into 2 for either or both of the rare and the abundant isotopologue mole fraction, and $\beta$ is non-zero for one of those, this leads to an approximate inverse dependence of the measured ratios on the concentration (Griffith et al., 2012).

### 3.1.1 Experiment description

Three experiments have been conducted over the last two years to determine the CMFD and to assess its stability over time. These experiments were conducted in December 2017 (experiment 1), in December 2018 (experiment 2) and in May 2019 (experiment 3). Experiment 1 has been conducted in cooperation with the Institute for Marine and Atmospheric research
Utrecht (IMAU) and served as the initial determination of the CMFD on the SICAS. Experiments 2 and 3 were meant to assess the stability of the CMFD over time. A methodology to determine the CMFD of the $r^{636}$ for a comparable dual-laser instrument has been described by McManus et al. (2015). In their study, a pure $CO_2$ working gas was diluted back to different $CO_2$ mole fractions using a set-up including computer controlled valves connected to a flow of air without $CO_2$ ("zero-air"). $CO_2$ and zero-air mixtures were led directly into the continuous flow dual laser instrument. In this way it was possible to measure the
CMFD over a wide range of $CO_2$ mole fractions, from $\sim$0 to 1000 ppm. The CMFD correction function for the isotope ratios was derived by applying a fourth order polynomial fit to these measurements.

For determination of the CMFD on the SICAS this approach was used with some adjustments. The SICAS is designed for the measurement of atmospheric samples of which the relevant range of $CO_2$ mole fractions is $\sim$370 – 500 ppm, and CMFD experiments were therefore for the most part conducted in this range. The SICAS measures discrete air samples, hence air
mixtures were manually prepared in sample flasks by back-diluting a well-known pure $CO_2$ in-house reference gas to different $CO_2$ mole fractions in the ambient range. Air samples for experiment 1 were prepared at the IMAU, Utrecht University.

Air samples for experiment 2 and 3 were prepared manually in our own laboratory, for the detailed procedure see Appendix A. The dilutor gas consists of natural air scrubbed of $CO_2$ and $H_2O$ using Ascarite® (sodium hydroxide coated silica, Sigma-Aldrich) and Sicapent® (phosphoric anhydride, phosphorus(V) oxide), which results in dry, $CO_2$-free natural air. For
experiment 2, additional samples were prepared using synthetic air mixtures with and without 1% Argon as dilutor gas for evaluation of the effect of air composition on the CMFD (see also section 3.1.6). With our manual preparation system we were able to prepare 10, 12 (with dilutor being whole air) and 7 flasks for experiment 1, 2 and 3 respectively, that were within our relevant range of atmospheric $CO_2$ mole fractions. McManus et al. (2015) applied a polynomial curve fit on the isotope ratio as a function of the $CO_2$ mole fraction. In this study we focus on a narrower range of $CO_2$ mole fractions and therefore
we expect that a linear or quadratic relationship is sufficient to describe the measured ratios as a function of the $CO_2$ mole fraction. We therefore considered the lower number of samples that were used for the three experiments in comparison to the continuous flow experiment by McManus et al. (2015) to be sufficient. Griffith (2018) showed that a combination of a linear and inverse relationship to the $CO_2$ mole fraction is theoretically expected, and this relationship fitted the data used in his study in practice. As we expect to have a relation of the measured delta values and the $CO_2$ mole fraction which is close to linear,
we use a quadratic relation which approximates this expected theoretical relation closely.

In the next two paragraphs we will discuss the results of the above described experiments for evaluation of the two sources of CMFD according to Wen et al. (2013) for the SICAS.

### 3.1.2 Spectroscopic non-linearities of measured isotopologues

The first source described by Wen et al. (2013), non-linearity of the relation between the measured isotopologue mole fraction and the $CO_2$ mole fraction, is determined by analysis of the linear fits of the measured rare isotopologue mole fractions ($X_{636}$, $X_{628}$ and $X_{627}$) as a function of the measured $X_{626}$. We used the $I_{S(t)}$ from equation 1 for both the rare isotopologue and the abundant isotopologue mole fractions. The 626 mole fraction is calculated by multiplying $I_{626(t)dc}$ by the known 626 mole fraction of the working gas. The residuals of the linear fits are manipulated such that residuals of the lowest mole fractions are zero (figure 5). A linear relation would result in residuals scattering around zero, without a pattern, while systematic non-linearities would result in a significant pattern, recurring for the different experiments. From the results in figure 5 we can conclude that non-linearities occur, however, these are only clearly visible in experiment 1 for the $X_{636}$ and the $X_{627}$ isotopologue, and to a lesser degree in experiment 2 for the $X_{636}$ isotopologue. The maximum residuals of both the $X_{636}$ and the $X_{627}$ are highest in experiment 1, which is also the experiment covering the highest range of $CO_2$ mole fractions. From these experiments we can therefore conclude that non-linearities of the measured rare isotopologue mole fractions and the $X_{626}$ isotopologue occur, but are only significant if the range of $CO_2$ mole fraction is higher than 100 ppm. For the $X_{628}$ we do not see significant non-linearities, even if the $CO_2$ mole fraction is much higher than 100 ppm. The maximum residuals of the $X_{628}$ are not influenced by the $CO_2$ mole fraction, and we therefore conclude that non-linearities are below the level of detection in these experiments.

### 3.1.3 Introduced dependency on measured delta values

The second source for CMFD, described by Wen et al. (2013), is the introduced dependency on measured isotope ratios if intercepts of the different isotopologues of the analyser's signal are non-zero, or as in our case for some experiments, if different isotopologues of the analyser's signal are non-linear in a different way. In this paragraph we look into the different possibilities to correct for the CMFD of the measured deltas based on observations of the experiments that were described in the section above.

Isotope ratios are susceptible to instrumental drift, but delta values are drift corrected as the uncalibrated delta value $\delta_S$ is calculated by:

$$\delta_S^* = (\frac{r_{S(t)}^*}{r_{WG(t)}^*} - 1) \tag{4}$$

Where S(t) and WG(t) stand for sample and working gas at the time of the sample measurement, respectively and * stands for the rare isotopologue of which the delta is calculated. The $r_{WG(t)}^*$ is calculated using the same method as $M_{WG(t)}$ is calculated in equation 1. The CMFDs of the deltas are determined by conducting a linear fit on the measured delta values as a function of the measured $CO_2$ mole fraction.

The results for $\delta^{636}$ are shown in figure 6 and slopes of all deltas and the standard errors of the slopes are shown in table 2. Note that in some cases the standard error of the slope is close to the slope itself and it is therefore questionable whether a significant CMFD is measured at all. As the $CO_2$ used for the different experiments was not of similar isotope composition,

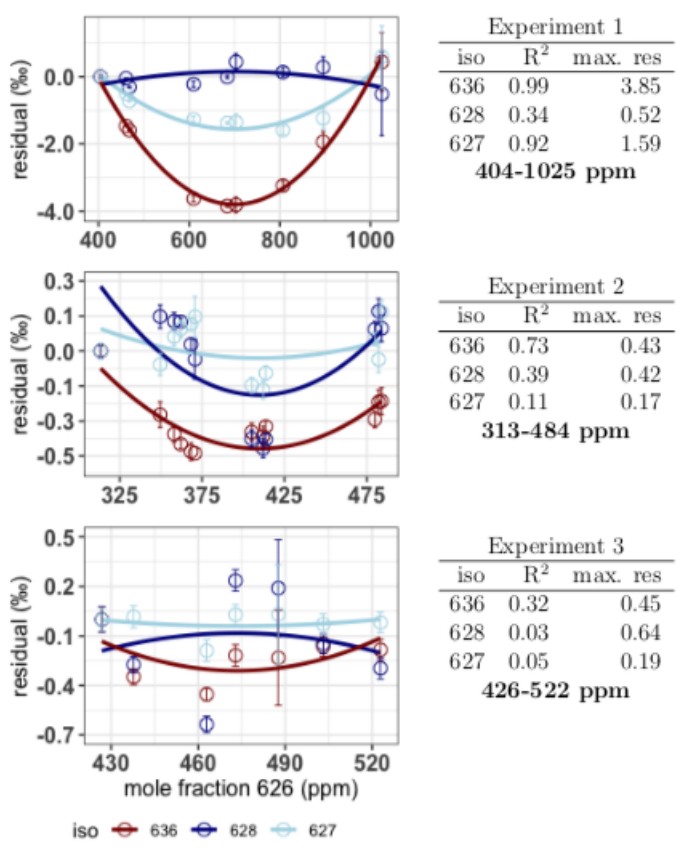

**Experiment 1**

| iso | $R^2$ | max. res |
|-----|-------|----------|
| 636 | 0.99 | 3.85 |
| 628 | 0.34 | 0.52 |
| 627 | 0.92 | 1.59 |

**404-1025 ppm**

**Experiment 2**

| iso | $R^2$ | max. res |
|-----|-------|----------|
| 636 | 0.73 | 0.43 |
| 628 | 0.39 | 0.42 |
| 627 | 0.11 | 0.17 |

**313-484 ppm**

**Experiment 3**

| iso | $R^2$ | max. res |
|-----|-------|----------|
| 636 | 0.32 | 0.45 |
| 628 | 0.03 | 0.64 |
| 627 | 0.05 | 0.19 |

**426-522 ppm**

iso ⬤ 636 ⬤ 628 ⬤ 627

**Figure 5.** Residuals (expressed in ‰ relative to the measured amount fraction) of the linear fit of the rare isotopologue abundancies as a function of the $X_{626}$ and the quadratic fit on the residuals. From top to bottom: Experiment 1, experiment 2 and experiment 3. The colours red, dark blue and light blue are used for the isotopologues 636, 628 and 627 respectively. Error bars are the combined standard deviations of the 626 and rare isotopologue measurements. Per isotopologue the $R^2$ of the quadratic fit on the residuals is indicated in the tables on the right, as well as the maximum residual (in ‰) on the linear fit of the rare isotopologue as a function of the $X_{626}$.

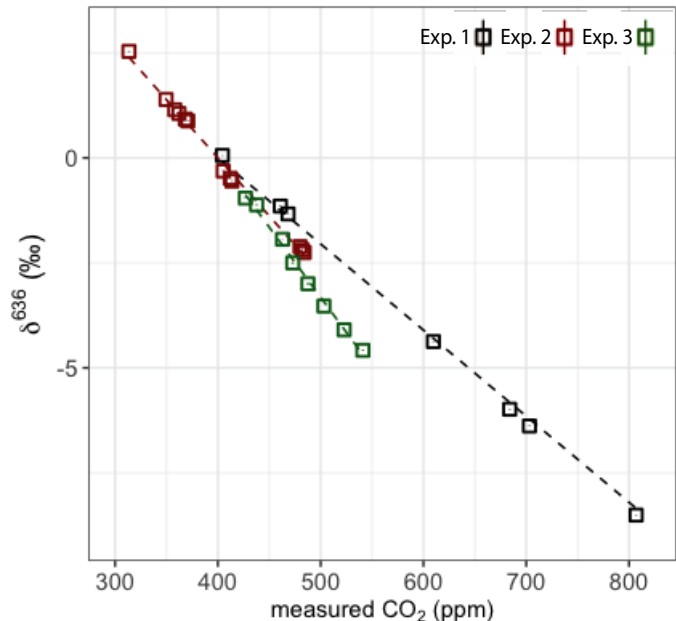

**Figure 6.** Measured $\delta^{636}$ of three experiments, black points are experiment 1, red points are experiment 2, green points are experiment 3.

the $\delta^{636}$ measurements in figure 6 were normalized such, that at the $CO_2$ mole fraction of 400 ppm all ratios are 1. Only the calculated slope is therefore of importance when considering the CMFD of the different experiments. From table 2 it is clear that the $\delta^{636}$ shows the strongest CMFD. The results show that the CMFD varies for the three different experiments for all measured deltas. Changing instrumental conditions can be an explanation for this change in the CMFD. A drop in measured laser intensity, for instance, was observed over the period between experiment 1 and experiment 3. We should, however, also

consider the different range of $CO_2$ mole fractions of the different experiments.

| all values | $\delta^{636}$ | | $\delta^{628}$ | | $\delta^{627}$ | |
| --- | --- | --- | --- | --- | --- | --- |
| in ‰/ppm | slope | se | slope | se | slope | se |
| exp. 1 | -0.0205 | 0.0003 | -0.0013 | 0.0003 | -0.0040 | 0.0004 |
| exp. 2 | -0.0277 | 0.0006 | -0.0027 | 0.0012 | 0.0029 | 0.0007 |
| exp. 3 | -0.0333 | 0.0011 | -0.004 | 0.003 | -0.0022 | 0.0005 |

**Table 2.** Slopes derived from the linear fits of the three measured deltas and $CO_2$ mole fractions, and the standard errors of the slopes. Delta values calculated with equation 4.

Although most of the variance occurring in the observed CMFD of the deltas (especially of the $\delta^{636}$) can be explained by the linear relationship we found with the measured $CO_2$ mole fraction, we can, from the observed non-linearities of the measured isotopologues, expect that these relations are better explained by a polynomial relation. We compare therefore both linear and

| all values in ‰ | | $\delta^{636}$ | $\delta^{628}$ | $\delta^{627}$ |
|---|---|---|---|---|
| exp.1 (404-1025ppm) | lin | 0.871 | 0.120 | 0.376 |
| | q | 0.072 | 0.142 | 0.100 |
| | fit lin | 0.141 | 0.090 | 0.169 |
| | fit q | 0.034 | 0.092 | 0.078 |
| exp. 2 (313-484ppm) | lin | 0.095 | 0.181 | 0.095 |
| | q | 0.054 | 0.164 | 0.097 |
| | fit lin | 0.086 | 0.175 | 0.093 |
| | fit q | 0.049 | 0.155 | 0.093 |
| exp. 3 (426-522ppm) | lin | 0.075 | 0.186 | 0.048 |
| | q | 0.084 | 0.162 | 0.032 |
| | fit lin | 0.093 | 0.191 | 0.037 |
| | fit q | 0.082 | 0.161 | 0.028 |

**Table 3.** Mean residuals for correction of the CMFD of the 3 measured deltas using 3 different scenarios; lin and q are calculated relations, using the linear and quadratic fit, respectively, of the rare isotopologue as a function of the abundant isotopologue. Fit and fit q are the linear and quadratic fit, respectively, of the measured delta values as a function of the $CO_2$ mole fraction. The minimum and maximum $CO_2$ mole fractions that were used per experiment are shown in the first column.

quadratic fits of the measured deltas with calculated relations derived from the fits of the rare isotopologues as a function of
the measured 626 isotopologue mole fraction. The theoretically expected combination of a linear and inverse relationship as
described in Griffith (2018) showed very similar results as the quadratic fit results, so we consider the quadratic fit to be a good
approximation of the theoretically expected relationship. Two relations are calculated: assuming a linear dependency of the
rare isotopologue on the abundant isotopologue and assuming a quadratic dependency of the rare isotopologue on the abundant
isotopologue. To compare all four scenarios (assuming a linear or quadratic CMFD of the measured deltas and calculation of
the CMFD of the deltas assuming a linear and a quadratic dependency of the rare isotopologues on the abundant isotopologue),
the mean of the absolute residuals of the observations was calculated for all three experiments and shown in table 3. The
quadratic fit of the deltas (fit q) shows the lowest mean residuals (except the $\delta^{13}C$ in experiment 3), followed by the calculated
relation of the deltas when using a quadratic relation of the individual isotopologues and the $CO_2$ mole fraction (q). From these
results it can therefore be concluded that determination of the quadratic CMFD of the deltas will give the most accurate results
in most cases. It is, however, the question whether this is feasible in practice, as we also know that the CMFD can change
through time due to changing instrumental conditions. Determination of a (accurate) quadratic relation requires at least three
measurement points (but preferably more) of atm-$CO_2$ of the same isotope composition. In our lab $CO_2$ in air samples of the
same isotope composition but deviating $CO_2$ mole fractions are prepared manually, introducing again uncertainties, and doing
these experiments regularly is therefore labor- and time intensive. Note as well that the range of the $CO_2$ mole fractions in the
3 experiments is quite high, considering the range of $CO_2$ mole fractions in atmospheric samples. The differences between

the four scenarios are significantly smaller in experiment 3 (covering 96 ppm) than in experiment 1 (covering 621 ppm). In the daily procedure of the SICAS there are at least two $CO_2$-in-air reference gases (in short reference gases), high pressurized cylinders containing gas of known isotope composition and $CO_2$ mole fraction, measured bracketing most of the $CO_2$ mole fractions (covering 82 ppm) that occur in atmospheric samples. As all sample and reference measurements are divided by measurements of the working gas when the delta values are calculated, the measured delta value of the working gas should always be zero. The two reference cylinders, together with the zero point for the working gas provide us with three points to determine a quadratic CMFD of the measured deltas. In this way it is possible to apply a quadratic CMFD correction on the measured deltas. It should be noted that tests showed that the improvements of a quadratic fit (in this form) compared to a linear fit were very small within the narrow range of $CO_2$ mole fractions occurring in the atmosphere, in line with the results of table 3. However, when samples of very deviating $CO_2$ mole fractions are measured, a quadratic fit will certainly improve the accuracy of the measurement.

## 3.2 Standard materials and reference scales

Four high pressure gas tanks (40 L Luxfer aluminum, alloy 6061, max. pressure of 200 bar) containing reference gases are used in the daily measurement procedure of the SICAS: a working gas used for drift correction and possibly for a first calibration step; a quality control tank that is being treated as a sample; and two tanks containing a high mole fraction reference gas and a low mole fraction reference gas, from now defined as the high reference and the low reference, which can thus be used for CMFD corrections. The high and low reference cover a great part of the $CO_2$ mole fraction range occurring in atmospheric samples.

It is known that for laser spectroscopy the composition of the sample air affects the absorption line profiles by pressure broadening effects ("matrix effects"), with non-negligible consequences (Nakamichi et al., 2006; Nara et al., 2012; Harris et al., 2020). Hence, it is likely that air composition affects $CO_2$ isotope measurements for the SICAS as well. The possible effects of air composition on the CMFD have been tested by measurement of samples of the same $CO_2$, mixed to different $CO_2$ mole fractions, prepared according to the method described in section 3.1.1 and Appendix A, using three different dilutor gases. The gases that have been used in addition to the $CO_2$ free natural air (whole air), were synthetic air (20% $O_2$ and 80% $N_2$, purity is $>= 99.99\%$) and the same synthetic air with addition of 1% of Argon, both prepared by Linde Gas. Linear fits on the measured $r^{636}$ as a function of the $CO_2$ mole fraction show a small but significant difference of the resulting slopes of $0.0014‰$ per ppm (table 4) between the synthetic air and whole air samples. For the $r^{628}$ and $r^{627}$ the slope was much smaller and the standard error of the slope was too large to determine a significant difference between the use of the synthetic dilutors and whole air. Nevertheless, to avoid inaccuracies due to a different CMFD of $r^{636}$ of samples and references, we solely use gas consisting of natural, dried air as then the effects of the (very small) variability in air composition are negligible.

The gas tanks were produced in-house from dry compressed natural air collected at the roof of our institute using a RIX compressor (model SA-3). The high and low reference were produced as follows: the high reference cylinder was filled up to $\sim 150$ bar in winter at the $15^{th}$ of January 2018, so the resulting $CO_2$ mole fraction is relatively high (423.77 $\pm 0.07$ ppm). The low reference cylinder was subsequently produced by transferring air from the high reference cylinder to an empty cylinder,

| dilutor gas | slope (‰ per ppm) | se. slope (‰) |
|---|---|---|
| whole air | -0.0272 | 0.0006 |
| synthetic air+Ar. | -0.0265 | 0.0008 |
| synthetic air | -0.0258 | 0.0007 |

**Table 4.** CMFD for samples of the same $CO_2$ diluted back with different dilutors. Per dilutor the slopes, resulting from the linear fits of measured $r^{636}$ and 626 isotopologue mole fraction (ppm), and the standard errors of the slopes are indicated.

using the pressure difference, while completely removing $CO_2$ from the air as it flew through a tube filled with Ascarite®. After the low reference cylinder was filled up to ~13 bar with $CO_2$ free air, the Ascarite® filled tube was removed and the filling was continued until the pressure of both cylinders was ~ 70 bar. In this way the $CO_2$ mole fraction of the low reference cylinder was reduced in comparison with the high reference cylinder, without influencing the $CO_2$ isotope ratios. The resulting $CO_2$ mole fraction of the low reference was 342.81±0.01 ppm. A scheme of the whole set-up and detailed description of the

procedure can be found in Appendix B.

     The $CO_2$ mole fraction of the tanks was measured on a PICARRO G2401 gas mole fraction analyzer and calibrated using in-house working standards, linked to the WMO 2007 scale for $CO_2$ with a suite of of four primary standards provided by the Earth System Research Laboratory (ESRL) of the National Oceanic and Atmospheric Administration (NOAA). The uncertainty of the WMO 2007 scale was estimated to be 0.07 $\mu mol/mol$. The typical measurement precision of the PICARRO G2401

measurements is 0.01 $\mu mol /mol$ resulting in a combined uncertainty of 0.07 $\mu mol/mol$ for the assigned $CO_2$ mole fraction values of the calibration tanks, while difference between the two cylinders is known with a much lower uncertainty. The PICARRO analysis is based on the 626 isotopologue mole fraction, not on whole $CO_2$. This is a potential source of error if the isotope composition of different reference gases varies significantly. As the isotope compositions of the used reference gases are close (see table 5), the variation is not significant for this error (Griffith, 2018).

Aliquots of all four tanks have been analyzed at the MPI-BGC in Jena by IRMS to link the $\delta^{13}C$ and $\delta^{18}O$ directly to the JRAS-06 scale (Jena Reference Air Set for isotope measurements of $CO_2$ in air (VPDB/VPDB-$CO_2$ scale)) (Wendeberg et al., 2013). The JRAS-06 scale uses calcites mixed into $CO_2$-free whole air to link isotope measurements of atm-$CO_2$ to the VPDB scale. An overview of our reference gases measured at the MPI-BGC and their final propagated error is presented in table 5 and it can be seen that the low and high reference are very close in isotope composition but seem to differ slightly in their $\delta^{13}C$

composition (by 0.05‰).

     Aliquots of the working gas and quality control gas were analyzed for their $\delta^{18}O$ and $\delta^{17}O$ values at the IMAU in Utrecht. These values were related to the VSMOW scale using two pure in-house reference gases. The $\delta^{17}O$ values are converted to the VPDB-$CO_2$ scale using the known relations between the reference materials VSMOW and VPDB. As the low reference and high reference were not measured at the IMAU, the $\delta^{17}O$ values were calculated from experimental results in which a linear

CMFD correction was conducted using the measured $\delta_S^{17}$ (as in equation 4) of the low and high reference, assuming that the $\delta^{17}O$ values of both gases are similar. Subsequently another linear fit is conducted on the CMFD corrected $\delta^{17}O$ values using

the known values of the working gas and quality control gas, deriving the calibrated $\delta^{17}O$ values of the low reference and high reference. Note that for measurement of our reference gases by the MPI-BGC and IMAU aliquots were prepared using the 'sausage' method, meaning that several (in this case 5) flasks are connected and flushed with the sample gas, resulting in a similar air sample in all flasks. However, deviations of the sampled air and the air in reference cylinders due to small leakages or other gas handling problems might be introduced.

| Tank | $CO_2$ (ppm) | $\delta^{13}C(‰)$ | $\delta^{18}O(‰)$ | $\delta^{17}O(‰)$ |
|---|---|---|---|---|
| working gas | 405.74 ±0.07 | -8.63 ±0.02 | -4.05 ±0.03 | -2.18 ±0.05 |
| quality control gas | 417.10 ±0.07 | -9.13 ±0.03 | -3.25 ±0.02 | -1.78 ±0.03 |
| low reference | 342.81 ±0.07 | -9.40 ±0.02 | -3.65 ±0.03 | -1.90 ±0.05 |
| high reference | 424.52 ±0.07 | -9.45 ±0.02 | -3.65 ±0.05 | -1.90 ±0.05 |

**Table 5.** Calibrated whole air working standards used in daily operation of the SICAS measurements. $CO_2$ measurements were conducted in our lab on a PICARRO G2401 gas mole fraction analyzer and the $\delta^{13}C$ and $\delta^{18}O$ values were measured at the MPI-BGC with a MAT-252 Dual-Inlet IRMS. The $\delta^{17}O$ values of the working gas and the quality control tank were measured at the IMAU, while the $\delta^{17}O$ of the low and high references were indirectly determined using our own measurements on the SICAS. Errors are all combined errors, including measurement precision, measurement accuracy and scale uncertainty.

## 3.3 Calibration methods

Two different calibration strategies are discussed in this section. The calibration strategies are based on the two main approaches for calibration of isotope measurements, as also described by Griffith et al. (2012) and, more recently by Griffith (2018), being (1) determine the isotopologue ratios, and calibrate those, taking the introduced CMFD into account, from now on defined as the ratio method (RM), and (2) first calibrate the absolute isotopologue mole fractions individually and then calculate the isotopologue ratios, from now on defined as the isotopologue method (IM). We give a brief introduction of the two calibration methods, as described in literature and we describe the measurement procedure that is used for both calibration methods. This section ends with a detailed description of both methods as applied for the SICAS measurements.

The RM, being very similar to calibration strategies applied by isotope measurements using DI-IRMS (Meijer, 2009), is usually based on reference gases covering delta values of a range which is similar to the range of the measured samples. Determination of the CMFD can be done by measuring different tanks of varying $CO_2$ mole fractions or by dynamical dilution of pure $CO_2$ with $CO_2$ free air (Braden-Behrens et al., 2017; Sturm et al., 2012; Griffith et al., 2012; McManus et al., 2015; Tuzson et al., 2008), again covering the $CO_2$ mole fraction range of the measured samples.

The IM has the advantage that there is no need to take the introduced CMFD into account (Griffith, 2018). As all isotopologues are calibrated independently, it is only necessary to use reference gases covering the range of isotopologue abundances as occurring in the samples. This can be realised by using reference gases containing $CO_2$ of similar isotope composition but varying $CO_2$ mole fractions as described in (Griffith, 2018) and successfully implemented in (Griffith et al., 2012; Flores et al., 2017; Wehr et al., 2013; Tans et al., 2017). The range of delta values that is measured in samples of atmospheric background

air is limited (range in unpolluted troposphere is -9.5 to -7.5‰ en -2 to +2‰ for $\delta^{13}C$ and $\delta^{18}O$, respectively (Crotwell et al., 2020)), hence this also applies to the range of delta values that should be covered by the reference gases when applying the RM. We decided therefore to use the same reference gases to test both calibration methods, varying mainly in $CO_2$ mole fraction (342.81-424.52 $\mu mol/mol$).

### 3.3.1  Measurement procedure

The measurement procedure that is used for both calibration methods is based on the alternating measurements of samples/reference gases and the working gas, so the drift corrected measurement value can be calculated as in equation 1. Per sample/reference gas measurement, there are 9 iterations of successive sample and working gas measurements, from now on called a measurement series, before switching to the next sample/reference gas measurement series. One measurement series lasts ∼ 30 minutes. Sample series are conducted once, while the reference gases series (low and high reference) are repeated 4

times throughout a measurement sequence. The quality control gas, a gas of known isotope composition which is not included in the calibration procedure, is also measured 4 times throughout the measurement sequence. One measurement sequence in which 12 samples are measured lasts therefore ∼12 hours. For the 9 measurement values of each measurement series outliers are determined using the outlier identification method for very small samples by Rousseeuw and Verboven (2002), and the mean values of the measurement series are calculated. For a complete step-by-step guide of all calculation steps for both

calibration methods, please see Appendix C.

### 3.3.2  Ratio method

In the RM measured isotopologue mole fractions are used for the estimation of isotope ratios (equation 1), which are calibrated to the international VPDB-$CO_2$ scale by measurement of several in-house $CO_2$-in-air references within the same measurement sequence. The working gas is used both for drift correction and a first calibration step, and the uncalibrated delta value $\delta_S$ is

calculated by:

$$\delta_S^* = (\frac{r_S^*}{r_{WG}^*} - 1) \tag{5}$$

Where S and WG stand for sample and working gas respectively. The calibrated $\delta^{13}C$, $\delta^{18}O$ and $\delta^{17}O$ based on the working gas that is used is then derived by:

$$\delta_{SCal} = (1 + \delta_{WG}) * \delta_S + \delta_{WG} \tag{6}$$

In which $\delta_{WG}$ is the known delta value of the working gas on the VPDB-$CO_2$ scale.

Up to this point, the procedures are more or less identical to those for IRMS measurements (but without the here unnecessary 'ion correction' and $N_2O$ correction). CMFD correction is specific for laser absorption spectroscopy and is crucial (as can be concluded from section 3.1.3) to derive accurate measurement results when calibration is done using the isotope ratios. We developed a calibration method based on the idea that including the measurement of two reference gases covering the $CO_2$

mole fraction range of the measured samples (in our case the low and high reference gas) enables the correction of the measured

isotope ratios. These two reference gases are measured several times throughout the measurement sequence and a quadratic fit of the mean of the residuals (measured $\delta_{SCal}$ - assigned $\delta_{VPDB}$), including the residual of zero for the (hypothetical) working gas measurement, as a function of the $CO_2$ mole fraction is done, so the following calibration formula can then be determined:

$$\delta_{VPDB} = \delta_{SCal} - ([CO_2]^2 * a + [CO_2] * b + c) \tag{7}$$

In which a and b are the second and first order coefficients respectively and c is the intercept of the quadratic fit of the residuals and the $CO_2$ mole fractions of the two reference gases, $[CO_2]$ is the measured $CO_2$ mole fraction and $\delta_{VPDB}$ is the calibrated $\delta$ value on the VPDB scale.

### 3.3.3   Isotopologue method

The IM as described by Flores et al. (2017) following methods earlier described by Griffith et al. (2012) will be briefly explained
here for clarity, before explaining the application of the IM on the SCIAS. Basically, the method treats the $CO_2$ isotopologues as if they were independent species, calibrates their mixing ratios individually, and only then combines the results to build isotope ratios and delta values. The mole fraction (X) of the four most abundant isotopologues of a measured $CO_2$ sample are determined using a suite (in our case the working gas and the high and low reference gas) of references gases with known $CO_2$ mole fractions and isotope compositions. The $CO_2$ mole fractions are ideally chosen such that normally occurring $CO_2$
mole fractions in atmospheric air are bracketed by the two reference gases. The low and high reference gases cover the range between 324.81 and 424.52 ppm, meaning that this method is only valid for samples within that range of $CO_2$ concentrations. The actual (or assigned) mole fractions ($X_a$) of the four most abundant isotopologues of the reference gases can be calculated using calculations 1-11 in Flores et al. (2017) which are listed in the Appendix B. Although the non-linearity of isotopologues as a function of the absolute $CO_2$ mole fraction has not been investigated in this study, it is very likely that non-linearities occur,
according to the results discussed in section 3.1.2. The broad range of $CO_2$ mole fractions that are covered by the reference gases, together with a hypothetical measurement of the working gas (of which the normalized isotopologue abundance will always be 1) enables to do a quadratic fit of the measured isotopologue abundance as a function of the assigned isotopologue mole fractions, by:

$$X_a = X_m^2 * c + X_m * d + e \tag{8}$$

In which c and d are the second and first order coefficients, respectively, and e is the intercept of the quadratic fit of $X_m$ as a function of $X_a$ of the reference gases. The resulting $X_a$s are used to calculate the isotope composition using calculation 1-11 in Appendix B. The introduced CMFD due to calibration on measured isotope ratios will not occur with this method, and a CMFD correction is therefore not necessary to yield accurate results.

    A complete overview of all calculation steps of both the RM and IM can be found in Appendix C.

 **4   Results and discussion**

## 4.1   Monitoring measurement quality and comparison of calibration methods of $\delta^{13}C$ and $\delta^{18}O$

To capture the very small signals in time-series of the isotope composition of atm-$CO_2$ it is crucial to keep track of the instrument's performance over the course of longer measurement periods. Variations in precision and accuracy of the isotope measurements on the SICAS are monitored by measurement of a quality control gas in every measurement sequence. Since the quality control gas measurement is not used for any correction or calibration procedures it can be considered as a known sample measurement that gives an indication of the overall instrument performance. Based on the WMO compatibility goals required for isotope measurements of atm-$CO_2$ we categorized (high quality (H), medium quality (M) and low quality (L)) three measurement periods for both the RM and IM. A period is rated as H if both the mean accuracy and the mean precision (expressed as the standard error) of the quality control gas measurements over that period are within the WMO compatibility goals (0.01‰ for $\delta^{13}C$ and 0.05‰ for $\delta^{18}O$ (Crotwell et al., 2020)), if the accuracy or precision is within the requirements but the other one is not, it is rated M, if both accuracy and precision do not fulfil the requirements it is rated L. Measurements of the quality control gas done over the period of 20th of November 2019 until the 4th of February 2020 are shown in figure 7 and we assigned three distinct measurement periods based on the quality of the measurements. The mean residuals and standard errors of all quality control gas measurements during the three periods are shown in table 6.

From the results we learn that the differences in performance between the two methods is minimal. The precision of the quality control gas measurements show the same results, while the accuracy shows small differences between the methods for the different periods. High quality performances are reached in period 1 for the $\delta^{13}C$ measurements, but in periods 2 and 3 both the precision and the accuracy are worse than 0.01‰, hence the measurement quality is low. The $\delta^{18}O$ measurements show high quality performance over the whole period.

| All values in ‰ | Ratio method | | Isotopologue method | |
|---|---|---|---|---|
| period | $\delta^{13}C$ residual | $\delta^{13}C$ st.error | $\delta^{13}C$ residual | $\delta^{13}C$ st.error |
| 1 | 0.002 | 0.008 | 0.006 | 0.008 |
| 2 | -0.03 | 0.02 | -0.03 | 0.02 |
| 3 | 0.04 | 0.03 | 0.03 | 0.03 |
| | $\delta^{18}O$ residual | $\delta^{18}O$ st.error | $\delta^{18}O$ residual | $\delta^{18}O$ st.error |
| 1 | 0.016 | 0.008 | 0.021 | 0.008 |
| 2 | -0.043 | 0.007 | -0.039 | 0.007 |
| 3 | -0.05 | 0.01 | -0.03 | 0.01 |

**Table 6.** Mean residuals and standard errors of the quality control measurements in the three different measurement periods.

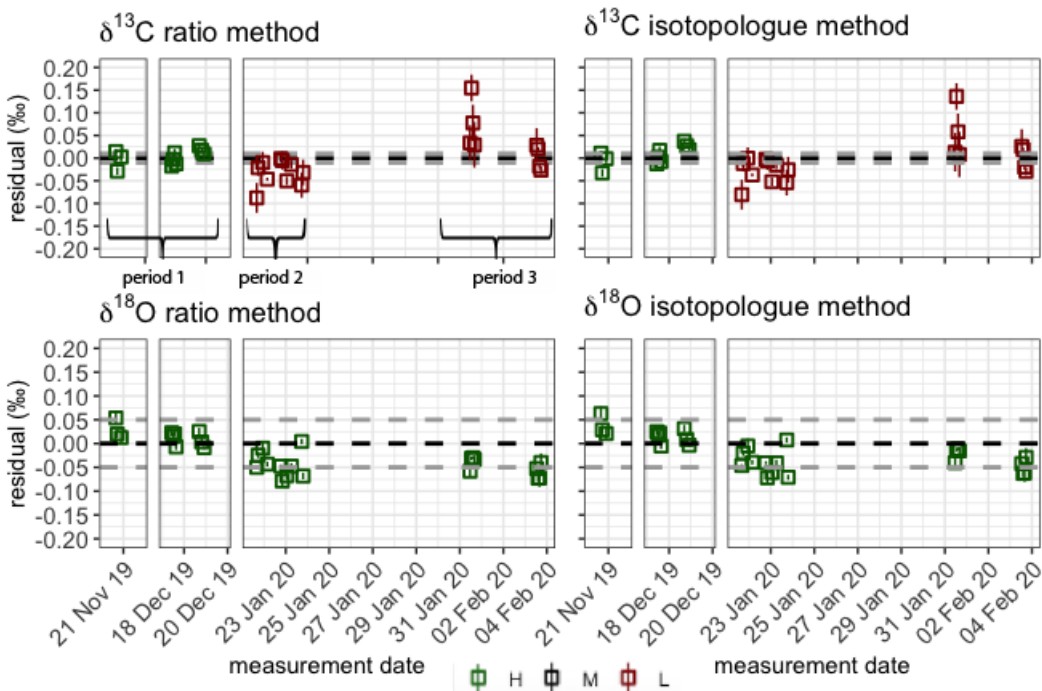

**Figure 7.** Quality control gas $\delta^{13}C$ (upper panels) and $\delta^{18}O$ (lower panels) measurements for both the RM (left) and IM (right). The assigned value of the quality control gas is indicated by the black dotted line and the WMO compatibility goals are indicated by the grey dotted lines. The error bars show the standard error of the measurements. Colour of the points indicates whether the measurements were performed in a High quality (green), Medium quality (black) or Low quality (red) measurement period (see section 4.1 for definitions).

## 460    **4.2   Uncertainty budget**

A combined uncertainty consisting of measurement uncertainties and scale uncertainties is calculated for the sample measurements. Measurement uncertainties include the standard error of the sample measurement, the repeatability of all (usually four) measurements of the quality control gas throughout the measurement sequence, and the residual of the mean of the quality control gas measurements from the assigned value. The measurement uncertainties will therefore vary with each measure-
ment/measurement sequence. We observe a high repeatability in all sequences included in the analysis of figure 7 (8 in total); with standard errors ranging between 0.008 and 0.03‰ and a mean of 0.02‰ for $\delta^{13}C$, and standard errors ranging between 0.007 and 0.01‰ and a mean of 0.008‰ for $\delta^{18}O$, for both methods. The residuals in these sequences show a higher contribution to the combined uncertainty and a small difference between the two calibration methods. The absolute residuals of the RM range between 0.002 and 0.04‰ with a mean of 0.024‰ for $\delta^{13}C$, and between 0.016 and 0.05‰ with a mean of 0.04‰
for $\delta^{18}O$. For the IM the residuals range between 0.006 and 0.03‰ with a mean of 0.02‰ for $\delta^{13}C$, and between 0.012 and 0.04‰ with a mean of 0.03‰ for $\delta^{18}O$. Hence, the RM shows slightly higher contributions to the combined uncertainty as a result of the accuracy of the quality control gas measurements.

The scale uncertainties, which are fixed for all measurement sequences in which the working gas, low reference and high reference are used for the sample calibration, were simulated using the Monte Carlo method. Input values were generated by choosing random numbers of normal distribution with the assigned value and uncertainty as in table 5, being the mean and the standard deviation around the mean, respectively. As the RM and IM follow different calibration schemes, the Monte Carlo simulations are discussed separately; for the RM the scale uncertainties of the assigned delta values result in an uncertainty in the calculated residuals which are quadratically fitted against the measured $CO_2$ mole fraction. The average uncertainties in the calibrated delta values of the 5 simulations are 0.03 and 0.05‰ for $\delta^{13}C$, and $\delta^{18}O$, respectively.

Besides the uncertainties introduced by the scale uncertainties of the delta values, the calibrated measurements of the IM are also affected by the scale uncertainties of the $CO_2$ mole fractions. Both the uncertainties in the delta values and in the $CO_2$ mole fractions affect the calculated assigned isotopologue abundances, which are quadratically fitted against the measured isotopologue abundances. The uncertainties in the assigned delta values result in average uncertainties of 0.03 and 0.06‰ for $\delta^{13}C$ and $\delta^{18}O$, respectively. The uncertainties in the assigned $CO_2$ mole fractions result in uncertainties of 0.005 and 0.018‰ for $\delta^{13}C$ and $\delta^{18}O$, respectively, and are small compared to the uncertainties of the assigned delta values.

Reducing the combined uncertainty of the $\delta^{13}C$ and $\delta^{18}O$ measurements of the SICAS will be most effective by determining the isotope composition of the reference gases with a lower uncertainty on the VPDB-$CO_2$ scale.

### 4.3   Intercomparison flask measurements

To test the accuracy of SICAS flask measurements over a wide range of $CO_2$ mixing ratios, as well as testing the lab compatibility of the SICAS measurements, we measured flask samples that are part of an ongoing lab intercomparison of atmospheric trace gas measurements including the $\delta^{13}C$ and $\delta^{18}O$ of $CO_2$ (Levin et al., 2004). The sausage flask Intercomparison Program (from now on defined as ICP) has provided since 2002 every 2 to 3 months (occasionally longer periods) aliquots of three high pressure cylinders containing natural air covering a $CO_2$ mixing ratio range of 340-450 $\mu mol/mol$. Participating laboratories send 6 flasks to the ICOS-CAL lab in Jena where these are filled with air from the three cylinders (two flasks per cylinder) with the so called 'sausage method'. The ICP provides therefore the opportunity to compare flask measurements on the SICAS with IRMS flask measurements of the MPI-BGC and other groups. We measured sausage series 90-94, which were filled between April 2018 and January 2020, and calibrated the isotope measurements both with the RM and the IM. SICAS measurements took place in the period from December 2019 to April 2020, with the consequence that the storage time of the flasks varies between 3 and 20 months. To place these results in context of intercomparison results of well established isotope and measurement laboratories, the ICP results of the Earth System Research Laboratory of the National Oceanic and Atmospheric Administration (NOAA) (Trolier et al., 1996) were also compared to the MPI-BGC results for the same sausage series. The lab-inter-comparison is presented in the usual way: the mean and standard deviation of the differences between our SICAS $\delta^{13}C$ and $\delta^{18}O$ results (both RM and IM calibrated) and the MPI-BGC ones are shown in table 7, along with the NOAA-MPI-BGC differences. The mean of the differences for the SICAS RM and NOAA results are both below 0.01‰, while the standard deviations of the differences are 0.05 and 0.07‰, respectively. The SICAS results calibrated with the IM show an offset with MPI-BGC of -0.013‰ and a standard deviation of the differences of 0.07‰. We can therefore conclude

| All values in ‰ | | $\delta^{13}C$ | | $\delta^{18}O$ | |
|---|---|---|---|---|---|
| | | mean | st. dev. | mean | st. dev. |
| SICAS - MPI-BGC | RM | 0.009 | 0.05 | -0.4 | 0.16 |
| | IM | -0.013 | 0.07 | -0.4 | 0.16 |
| NOAA - MPI-BGC | | -0.007 | 0.07 | 0.130 | 0.08 |

**Table 7.** Lab intercomparison of ICP sausage 90-94 results, only including datapoints within the $CO_2$ mole fraction range of the used calibration tanks (342.81-424.52 $\mu mol/mol^{-1}$). Differences between the SICAS, of both calibration methods, as well as the NOAA IRMS results and the MPI-BGC IRMS results are shown. The mean difference as well as the standard deviation of the differences of the $\delta^{13}C$ and $\delta^{18}O$ are shown.

that the differences in performance between the RM and the IM are minimal and both methods show comparable results for the measured differences between MPI-BGC as for the differences between the NOAA and the MPI-BGC.

When we compare the $\delta^{18}O$ measurements, we find that the SICAS results are consequently significantly more depleted with an average difference of -0.4‰ compared to the MPI-BGC results and that the differences vary strongly with a standard deviation of 0.16‰. $\delta^{18}O$ results of the ICP program show in general a larger scatter among the labs than $\delta^{13}C$ results (Levin et al., 2004), as is also visible in table 7 for the NOAA-MPI-BGC differences. The differences between the SICAS- and the MPI-BGC results, however, are far larger than those (or than in fact all differences in the ICP programme). The reason for this too depleted signal is presumably equilibration of $CO_2$ with water molecules on the glass surface inside the CIO-type sample flasks during storage. Earlier (unpublished) results from our $CO_2$ extraction system indicated that the water content of our dried atmospheric air samples increased as a function of time inside the flasks. Our atmospheric samples are stored at atmospheric pressure or lower (down to 800 mbar) when part of the sample has been consumed by different measurement devices. The CIO flasks are sealed with two Louwers-Hapert valves and Viton O-rings of which it is known that permeation of water vapour (as well as other gases) occurs over time (Sturm et al., 2004). Both the pressure gradient and the water vapour gradient between the lab atmosphere and the dry sample air inside the flask lead to permeation of water molecules through the valve seals. To check this hypothesis an experiment was conducted in which CIO flasks were filled with quality control gas and were measured the same day of the filling procedure and one week and three months later (see table 8). The results show no significant change in the $\delta^{13}C$, while for the $\delta^{18}O$ there is a strong depletion of the flask measurements after 3 months, deviating more than -0.2‰ in comparison to the cylinder measurements. After 1 week there is no change in the $\delta^{18}O$, indicating that depletion of the $\delta^{18}O$ in the CIO flasks occurs over longer time periods. As the flasks from the ICP were measured at the SICAS after relatively long storage times, sometimes longer than two years, this is likely the explanation of the too depleted values in comparison to the MPI-BGC results. A depletion twice as small as for $\delta^{18}O$ is observed in the $\delta^{17}O$ values, as one would expect for isotopic exchange with water. Further investigations about the changing oxygen isotope signal in CIO-sample flasks are being conducted with the aim to be able to make reliable assessments on the quality of $\delta^{18}O$ and $\delta^{17}O$ flasks measurements on the SICAS.

| Storage time | Flasks | | | | | | |
|---|---|---|---|---|---|---|---|
| All values in ‰ | $\delta^{13}C$ | std. | $\delta^{18}O$ | std. | $\delta^{17}O$ | std. | n |
| 1 day | -9.177 | 0.023 | -3.336 | 0.002 | -1.835 | 0.011 | 2 |
| 1 week | -9.14 | 0.04 | -3.312 | 0.012 | -1.854 | 0.011 | 2 |
| 3 months | -9.191 | 0.019 | -3.51 | 0.12 | -1.92 | 0.04 | 4 |
| | Cylinder | | | | | | |
| 1 day | -9.178 | 0.024 | -3.332 | 0.009 | -1.854 | 0.017 | 4 |
| 1 week | -9.160 | 0.023 | -3.299 | 0.009 | -1.857 | 0.024 | 3 |
| 3 months | -9.180 | 0.020 | -3.299 | 0.028 | -1.893 | 0.011 | 4 |

**Table 8.** Results of isotope measurements of quality control gas from the tank and quality control gas air in flasks (calibrated with the RM) at different periods after the flask filling procedure. The last column shows the number of cylinder measurements or the number of flasks that were used to calculate the average and the standard deviation.

To check the performance of the SICAS for both the IM and RM over the wide $CO_2$ range that is covered by the ICP sausage samples, the differences between the MPI-BGC and the SICAS results are plotted in figure 8 against the measured $CO_2$ mole fraction. Shown is that for both methods the highest differences are seen at the higher end of the $CO_2$ mole fraction above 425 ppm, and therefore far out of the range that is covered by the high and low references ($\sim$343-425 ppm). Extrapolation of the calibration methods outside the $CO_2$ mole fraction range of the reference gases yields worse compatibility with MPI-BGC, possibly due to the non-linear character of both the isotopologue $CO_2$ dependency and the ratio $CO_2$ dependency. It should therefore be concluded that, to achieve highly accurate results of isotope measurements over the whole range of $CO_2$ mole fractions found in atmospheric samples, the range covered by the reference gases would ideally be changed to $\sim$380-450 ppm. The results of the IM are slightly better in the $CO_2$ range above 425 ppm, specifically the point closest to 440 ppm shows a significantly smaller residual ($\sim$0.1‰ less) than the RM. The better result of extrapolation of the determined calibration curves for the IM method could be due to the lesser degree of non-linearity of the measured isotopologue abundances as a function of the assigned isotopologue abundances, in comparison to the non-linearity of the measured isotope ratios as a function of the $CO_2$ mole fraction. More points in this higher range are needed, however, to draw any further conclusions on this matter.

### 4.4 Potential of SICAS $\Delta^{17}O$ measurements for atmospheric research

With the direct measurement of $\delta^{17}O$ in addition to $\delta^{18}O$ (triple oxygen isotope composition) of atm-$CO_2$, the $\delta^{17}O$ excess ($\Delta^{17}O$) can be calculated. $\Delta^{17}O$ measurements can be a tracer for biosphere activity (Hoag et al., 2005), atmospheric circulation patterns (Mrozek et al., 2016) and different combustion processes (Horváth et al., 2012). The $\Delta^{17}O$ is usually defined as:

$$\Delta^{17}O = ln(1 + \delta^{17}O) - \lambda * ln(1 + \delta^{18}O) \tag{9}$$

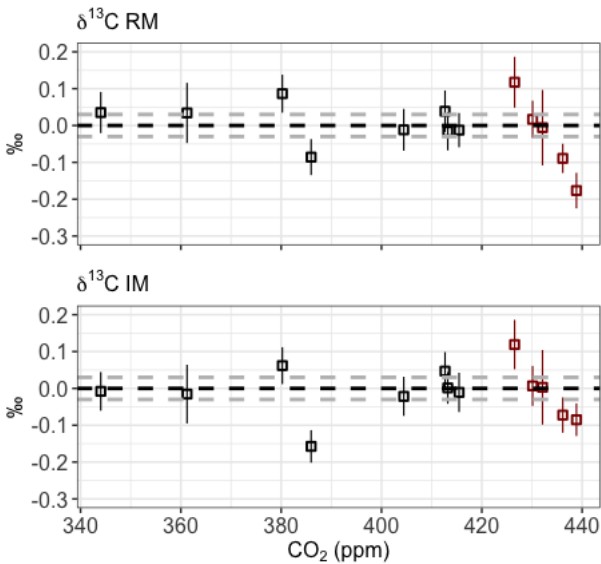

**Figure 8.** Results of the intercomparison of $\delta^{13}C$ measurements on the SICAS and on the IRMS facility at the MPI-BGC for both the RM (upper) and IM (lower). The MPI-BGC results were subtracted from the SICAS results, the error bars show the combined uncertainty of the SICAS measurements. The grey dotted lines show the 0.03‰ range of residuals. Red data points are outside of the $CO_2$ mole fraction range of the reference gases.

Variations in the $\Delta^{17}O$ signal in the troposphere are mainly depending on biosphere activity and the influx of stratospheric $CO_2$ (Koren et al., 2019; Hofmann et al., 2017; Hoag et al., 2005). High measurement precision and accuracy of both the $\delta^{18}O$ and the $\delta^{17}O$ is needed to capture spatial gradients and seasonal cycles in the $\Delta^{17}O$, of which seasonal variations of 0.13‰ (Hofmann et al., 2017) and 0.211‰ (Liang et al., 2017) have been reported. So far it has been an extremely complex and time intensive process to measure $\delta^{17}O$ of $CO_2$ using DI-IRMS (Hofmann and Pack, 2010; Barkan and Luz, 2012; Mahata

et al., 2013; Adnew et al., 2019). Dual-laser absorption spectroscopy as presented in this paper does not require any sample preparation and would therefore be a great step forward in the use of $\Delta^{17}O$ as a tracer for atm-$CO_2$. Here we present the measurement precision and stability of the $\delta^{17}O$ as well as the $\Delta^{17}O$ measurements of our quality control tank in figure 9 and table 9, and we evaluate the potential for contributing in the field of triple oxygen isotope composition studies.

     All results show too enriched values according to the assigned values, which is probably due to the fact that the assigned

$\delta^{17}O$ values of the low and high references have been determined indirectly, as discussed in section 3.2. A direct determination of the $\delta^{17}O$ values of our low and high references would supposedly improve the accuracy of both methods. The $\Delta^{17}O$ accuracy is dependent on both the $\delta^{17}O$ and $\delta^{18}O$ results, where $\Delta^{17}O$ values will deviate more if those results deviate in opposite directions and vice versa. Furthermore, it is striking that the mean standard errors of measurement periods 2 and 3 are twice as low for the IM than for the RM. The $r^{627}$, used for the RM, is calculated by dividing $X_{627}$, derived from laser 1,

by $X_{626}$ derived from laser 2. It can be that the two lasers do not drift in the same direction and the advantage of cancelling

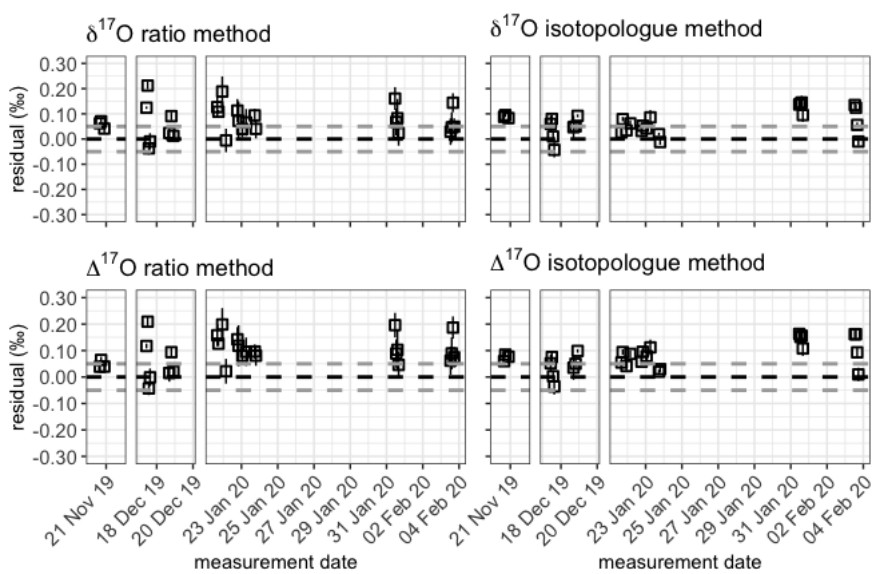

**Figure 9.** Quality control gas $\delta^{17}O$ (upper panels) and $\Delta^{17}O$ (lower panels) measurement averages for the three measurement periods for both the RM (left) and the IM (right). The averages are indicated by the black dotted line and the grey dotted line show the 0.05‰ range around the average. The error bars show the standard error of the measurements.

| All values in ‰ | Ratio method | | Isotopologue method | |
| --- | --- | --- | --- | --- |
| period | $\delta^{17}O$ residual | $\delta^{17}O$ st.error | $\delta^{17}O$ residual | $\delta^{17}O$ st. error |
| 1 | 0.06 | 0.02 | 0.06 | 0.02 |
| 2 | 0.08 | 0.04 | 0.04 | 0.01 |
| 3 | 0.08 | 0.05 | 0.10 | 0.02 |
| | $\Delta^{17}O$ residual | $\Delta^{17}O$ st. error | $\Delta^{17}O$ residual | $\Delta^{17}O$ st. error |
| 1 | 0.06 | 0.02 | 0.05 | 0.02 |
| 2 | 0.04 | 0.04 | 0.07 | 0.02 |
| 3 | 0.10 | 0.05 | 0.13 | 0.02 |

**Table 9.** Average of the residuals from the assigned value and mean of the standard error of the quality control gas $\delta^{17}O$ and $\Delta^{17}O$ measurements per period for both the RM and the IM.

out these drifts by dividing the two measured values will not apply. The outlier analysis of the IM might in that case be more effective as it is performed on both the measured $^{16}O$ and $^{17}O$ abundances, while for the RM it is only performed on the $r^{627}$. A comparison of the correlation coefficients between the 627 peak results and the 626 peak results from both lasers shows no significant difference (and a value of $\sim 0.65$), meaning that using the 626 peak of laser 1 for the $\delta^{17}O$ calibration will not improve the precision of the RM results.

Due to the lower seasonal variations of the $\Delta^{17}O$ values, even higher measurement precisions are a prerequisite and in Hofmann et al. (2017) it is stated that a measurement precision of 0.01‰ or better is required to capture these variations and to use the $\Delta^{17}O$ value as a potential tracer for GPP. These precisions are now not yet achieved, but the results of the IM calibrated values show that small improvements in the measurement precision of the SICAS can bring the $\Delta^{17}O$ measurements close to the 0.01‰ precision. This could for instance be accomplished by deciding to conduct more iterations per measurement, if sample size allows this. In section 2.2 the contamination of the mirrors was discussed as the potential cause for the decreased signal-to-noise ratio in over the period September 2017-July 2019. Placing new mirrors in the optical cell might therefore improve the quality of the measurements further. As the quality of the $\Delta^{17}O$ measurements depends directly on the quality of the the the $\delta^{18}O$ and the $\delta^{17}O$ measurements, it will be important to monitor the measurement quality of both isotope values over time using the measurements of the quality control gas. If SICAS measurements are to be used for comparison with $\Delta^{17}O$ measurements from other labs or measurement devices, it is necessary to add the error introduced by the scale uncertainties of the reference gases as well. For both the $\delta^{17}O$ and $\Delta^{17}O$ these uncertainties are 0.08‰, as calculated with a Monte-Carlo simulation as described in section 4.2. As long as only measurements from this device are used, seasonal and diurnal cycles are measured with much lower uncertainties. The high residuals found for the quality control gas measurements of the $\delta^{17}O$ and $\Delta^{17}O$ show that these uncertainties are probably an underestimation, as the assigned values of the low and high reference, which were not directly measured at the IMAU, are not known with high accuracy. For reducing the combined uncertainty it is therefore crucial to have all reference gases directly determined for their $\delta^{17}O$ values, as well as reducing the scale uncertainties of both the $\delta^{17}O$ and $\delta^{18}O$ values of the reference tanks.

## 5  Conclusions and Outlook

In this study we discuss the measurement performances of our Aerodyne dual-laser absorption spectrometer in static mode for stable isotope measurements of atm-$CO_2$ in dry whole air samples. We implemented two different calibration methods based on the same measurement procedure, the ratio method (RM) and the isotopologue method (IM). Short-term instrumental drift can effectively be corrected by continuously alternating sample measurements with measurements of a machine working gas. Nine aliquots are measured per sample/reference gas and two reference gases covering a wide range of $CO_2$ mole fractions, as well as a quality control tank serving as a known unknown, are measured four times throughout a measurement sequence. The RM is based on calibration of measured isotopologue ratios (or delta values, calculated directly from the measured sample and working gas ratios), including correction for a non-linear $CO_2$ mole fraction dependency. This correction is determined by doing a quadratic fit of the residuals of the calibrated delta values of the reference gases as a function of the measured $CO_2$

mole fraction. The IM is based on calibration of measured isotopologue abundances, using a quadratic fit of the measured values of the reference gases as a function of the assigned isotopologue values. In optimal measurement conditions, precisions and accuracies of $<0.01$ and $<0.05$‰ for $\delta^{13}C$ and $\delta^{18}O$ are reached for measurements of the quality control tank for both calibration methods. The combined uncertainty of the measurements includes also the repeatability of the four quality control gas measurements throughout the measurement sequence, with mean values of 0.014 and 0.012‰. The last components in the combined uncertainty calculation are caused by scale uncertainties of the reference gases used for the sample calibration, which are 0.03 and 0.05‰ for $\delta^{13}C$ and $\delta^{18}O$ of the RM, respectively and 0.03 and 0.06‰ for $\delta^{13}C$ and $\delta^{18}O$ of the IM, respectively.

A comparison of SICAS results, for both calibration methods, with results from the MPI-BGC from the sausage ICP show that sample results within the $CO_2$ mole fraction range of both methods are of similar quality when compared to the MPI-BGC results. Better results were achieved for the IM for samples outside the $CO_2$ mole fraction range, but more measurements are needed to determine whether the IM is indeed less vulnerable to extrapolation of the calibration. As extrapolation should at any time be avoided, using reference gases that cover the range of atmospheric samples is advisable for more reliable measurement results. We found that $\delta^{18}O$ measurements were consequently too depleted due to too long storage times of the CIO flasks before measurement. Future investigations will give more insight in the stability of the oxygen isotopes within the CIO flasks and we will evaluate the possibility of a correction based on storage time.

$\delta^{17}O$ and $\Delta^{17}O$ results of the quality control gas show consequently too enriched values, which is probably caused due to the indirect determination of the $\delta^{17}O$ values of two of the reference gases. The measurement precision is significantly better for the IM, with standard errors not higher than 0.02‰, while the measurement precision of the RM shows standard errors ranging between 0.02 and 0.05‰. Results of the IM come close to the required 0.01‰ precision to capture seasonal variations of the atmospheric $\Delta^{17}O$ signal. For a combined uncertainty of the $\delta^{17}O$ and $\Delta^{17}O$ values, an additional uncertainty of 0.08‰ must be added due to effects of the scale uncertainties of the the reference gases, indicating that improved determination of the oxygen stable isotope values of the reference gases will be essential to reach high precision $\Delta^{17}O$ measurements that are compatible with measurements from other labs. We will show the actual achievements of $\Delta^{17}O$ measurements with this instrument for a record of atmospheric $CO_2$ samples of our atmospheric monitoring station Lutjewad in a forthcoming paper.

*Data availability.* All data that has been used for this study which was measured at the SICAS can be found in the supplementary material.

**Appendix A: Preparation procedure for $CO_2$-in-air samples**

The pure $CO_2$ aliquots were prepared by connecting a 20 mL flask containing a pure $CO_2$ local reference gas to a calibrated adjustable volume. The required amount of $CO_2$ in the adjustable volume could be determined by measuring the pressure at a resolution of 1 mbar using a pressure sensor (Keller LEO 2). Both the sample flask and adjustable volume were connected to a vacuum ($3.3 * 10^{-5}$ mbar) glass line. The $CO_2$ in the adjustable volume was transferred cryogenically (using liquid

nitrogen) into a small glass tube shape attachment on the side of the evacuated sample flask which was custom-made for this purpose and subsequently the zero-air dilutor gas was added. The dilutor gas consists of natural air scrubbed of $CO_2$ and $H_2O$ using Ascarite® (sodium hydroxide coated silica, Sigma-Aldrich) and Sicapent® (phosphoric anhydride, phosphorus(V) oxide), which results in dry, $CO_2$-free natural air. For experiment 2, additional samples were prepared using synthetic air mixtures with and without 1% Argon as dilutor gas for evaluation of the effect of air composition on the CMFD (see also section 3.1.6).

After closing the flask, the mixture was put to rest for at least one night before measurement to ensure the $CO_2$ and the dilutor were completely mixed.

## Appendix B:  Equations for calculation of isotopologue mole fractions

Individual isotopologues of standards of known $CO_2$ mole fractions and isotope composition are calculated for the IM calibration method by the equations below, according to Flores et al. (2017), starting with equations for the atomic abundances X

in each of the calibration gas mixtures (B1-B5):

$$X(^{12}C) = \frac{1}{1 + R^{13}} \tag{B1}$$

$$X(^{13}C) = \frac{R^{13}}{1 + R^{13}} \tag{B2}$$

$$X(^{16}O) = \frac{1}{1 + R^{18} + R^{17}} \tag{B3}$$

$$X(^{17}O) = \frac{R^{17}}{1 + R^{18} + R^{17}} \tag{B4}$$

$$X(^{18}O) = \frac{R^{18}}{1 + R^{18} + R^{17}} \tag{B5}$$

where

$$R^{13} = R^{13}_{VPDB-CO_2} * (1 + \delta^{13}C) \tag{B6}$$

$$R^{17} = R^{17}_{VPDB-CO_2} * (1 + \delta^{18}O)^{\lambda} \tag{B7}$$

$$R^{18} = R^{18}_{VPDB-CO_2} * (1 + \delta^{18}O) \tag{B8}$$

and $\delta^{13}C$ and $\delta^{18}O$ are the delta values. $R^{13}_{VPDB-CO_2}$ (0.011180), $R^{17}_{VPDB-CO_2}$ (0.0003931) and $R^{18}_{VPDB-CO_2}$ (0.00208835) values were taken from Brand et al. (2010) for $VPDB-CO_2$. Then each carbon dioxide isotopologue mole fraction in the reference gas was calculated according to its composition using equations B9-B10:

$$X_{626} = (X(^{12}C) * X(^{16}O) * X(^{16}O)) * X_{CO_2} \tag{B9}$$


$$X_{636} = (X(^{13}C) * X(^{16}O) * X(^{16}O)) * X_{CO_2} \tag{B10}$$

$$X_{628} = (X(^{12}C) * X(^{16}O) * X(^{18}O)) * 2 * X_{CO_2} \tag{B11}$$

$\quad X_{627} = (X(^{12}C) * X(^{16}O) * X(^{17}O)) * 2 * X_{CO_2} \tag{B12}$

For a more elaborated explanation of these equations, see Flores et al. (2017).

## Appendix C: Step-by-step calculation steps

### C1    Ratio method

1. Calculate ratios from the measured isotopologue abundances:

$$r^* = \frac{X_*}{X_{626}} \tag{C1}$$

With r being the ratio, X the measured isotopologue abundance as the default output, so scaled for the natural abundance, * stands for one of the three rare isotopologue (636, 628 or 627) and 626 standard for the abundant isotopologue. The $CO_2$ mole fraction is calculated by:

$$[CO_2] = A_{626} + A_{636} + A_{628} + A_{627} \tag{C2}$$

With A being the actual measured abundance, so calculated back using the natural abundance values for the isotopologues as defined in Rothman et al. (2013.

2. Use only the relevant interval (in our case 30-60 seconds) from measured ratio and $[CO_2]$ per measurement

3. Do a drift correction and calculate the uncalibrated delta value by:

$$\delta^*_S(t)dc = \frac{r_{S(t)}}{r_{WG(t)}} - 1 \tag{C3}$$

With S standing for sample, t for time of the measurement, dc for drift corrected and WG for working gas. With $r*_{WG(t)}$
derived from applying a time dependent linear fit of the $r^*_{WG(t-1)}$ and $r^*_{WG(t+1)}$, following:

$$r^*_{WG(t)} = \alpha + \beta * t \tag{C4}$$

The $[CO_2]$ is also drift corrected by:

$$[CO_2]_{S(t)dc} = \frac{CO_{2S(t)}}{CO_{2WG(t)}} \tag{C5}$$

4. Group all $\delta^*$ values and $[CO_2]$ values per measurement series and do an outlier analysis per series. We adapted the method as described in Rousseeuw and Verboven (citepRousseeuw2002):

   (a) Define the variable "sborder" (sborder=2), defining the strictness of filtering.

   (b) Calculate for all values in the series the (absolute) deviation from the median of the series, resulting in a new series containing the distance from the median (DM)

(c) Calculate the MAD (median absolute deviation), by:

   $$MAD = 1.483 * median(DM) \tag{C6}$$

   (d) Calculate per value of the series the deviation with the following equation:

   $$deviation = \frac{abs(x_{1,2..N} - median)}{sborder * MAD} \tag{C7}$$

   with $x_{1,2..N}$ standing for the measurement values from the measurement series.

(e) If the deviation of a value is higher than 1, the value is identified as an outlier.

5. calculate the mean and standard error per measurement series, excluding the identified outliers

6. do first a one-point-calibration on all mean values using the known values of the working gas, by:

   $$\delta^*A = (1 + WG^*A) * \delta^* + WG^*A \tag{C8}$$

   With A standing for atom (C or O) and $WG^*$ being the assigned isotope value of the working gas. The $[CO_2]$ is calibrated
by:

   $$[CO_2]_c = [CO_2]_S * [CO_2]_{WG} \tag{C9}$$

   With $[CO_2]_c$ being the calibrated $[CO_2]$ value, $[CO_2]_S$ being the mean of sample $[CO_2]$ measurement and $[CO_2]_{WG}$ being the assigned $CO_2$ mole fraction value of the working gas.

7. Calculate the means of the $\delta^*A$ values and the $[CO_2]_c$ values of the high and the low reference gas measurements that
were done throughout the measurement sequence (we normally do four measurements of both reference cylinder) and calculate the residual of the means from their assigned $\delta^*A$ values.

8. Use the two calculated residuals, together with a residual of 0 for a hypothetical working gas measurement to do a quadratic fit ( $ax^2 + bx + c$) of the residuals as a function of the $[CO_2]_c$ and calculate the final $\delta^* A$ on the VPDB scale by:

$$\delta^* A_{VPDB} = \delta^* A - ([CO_2]_c^2 * a + [CO_2]_c * b + c) \tag{C10}$$

9. calculate the combined uncertainty by:

$$cu\delta^* A = \sqrt{sud^2 + QC_{ste}^2 + QC_{res}^2 + se_m^2} \tag{C11}$$

In which sud is the scale uncertainty of delta values (derived from a Monte Carlo simulation), $QC_{ste}$ is the standard error of all (usually four) quality control gas measurements throughout the measurement sequence, $QC_{res}$ is the residual of the mean of all quality control gas measurements and $se_m$ is the standard error of the measurement.

## C2 Isotopologue method

1. Use only the relevant interval (30-60 seconds) from measured isotopologue abundances per measurement

2. Do a drift correction by:

$$a_{S(t)dc}^* = \frac{a_{S(t)}}{a_{WG(t)}} \tag{C12}$$

The $a_{WG(t)}$ is derived with the same method as for the RM.

3. Group all $a^*$ per measurement series and do an outlier analysis per sample. The same method as for the RM is used.

4. calculate the mean per measurement series, excluding the identified outliers

5. calculate the quadratic calibration curves ($\alpha * x^2 + \beta * x + \gamma$) for all four isotopologues, by fitting the mean of all low and high reference measurements (usually four per measurement sequence) and an additional value of 1 for the hypothetical working gas measurement as a function of the assigned isotopologue mole fraction

6. calculate the calibrated isotopologue mole fractions of all four isotopologues for all measurements, so not for the mean of the grouped measurements but for all drift corrected $a^*$s from step 2:

$$a_{S(cal)}^* = \alpha * a_S^{*2} + \beta * a_S^* + \gamma \tag{C13}$$

7. calculate the calibrated delta values using the calibrated isotopologue abundances for all sample measurements

8. Group all $\delta^* A$ per measurement series and do an outlier analysis per sample, using again the same method as described in the RM

9. calculate the mean and standard error of all $\delta^* A$s per measurement series, excluding the identified outliers

10. Calculate the combined uncertainty of the measurement:

$$cu\delta^* A = \sqrt{sud^2 + suc^2 + QC_{ste}^2 + QC_{res}^2 + se_m^2} \qquad \text{(C14)}$$

In which suc is uncertainty introduced by the scale uncertainty of the $CO_2$ mole fractions

### Appendix D: Set-up for preparation of low reference

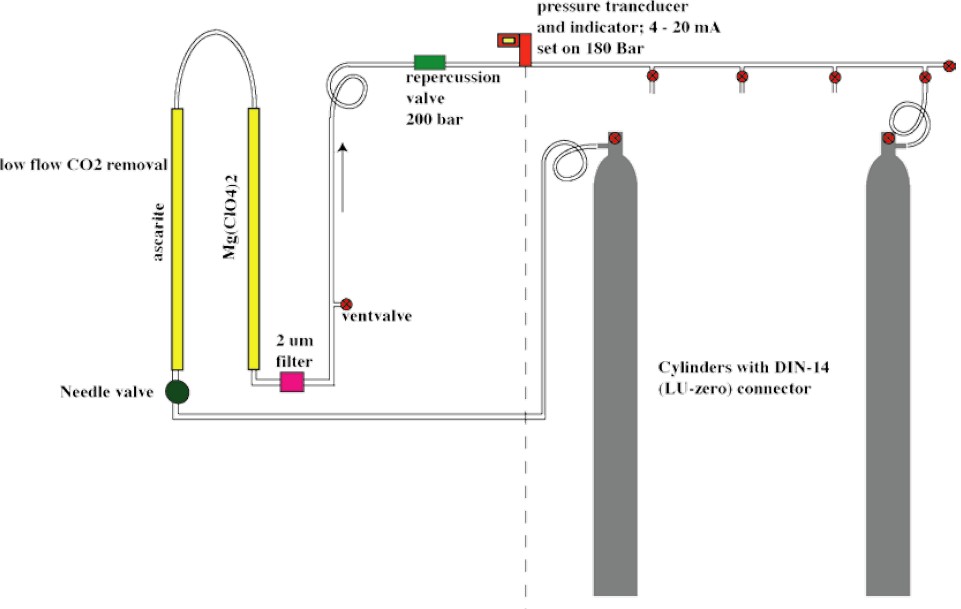

**Figure D1.**

The set-up is as follows: a high reference, filled up to ~150 bar with dry natural air, is connected to a similar, empty cylinder. Half of the air in the high reference tank will be transferred (passive transfer using the pressure difference) into the empty cylinder to produce the low reference. The $CO_2$ mole fraction in the low reference is reduced by leading part of the air over an Ascarite® filled cartridge that removes all $CO_2$ from the air, so no isotope fractionation will occur. Successively it is led over a magnesium perchlorate filled cartridge to remove water from the air that is potentially stored in the hydrophilic Ascarite®. A needle valve installed before the cartridges creates a low flow to ensure the complete removal of the $CO_2$ from the air. The pressure sensor installed after the repercussion valve enables to estimate when the low reference cylinder is filled with the amount of $CO_2$ free air needed to obtain the preferred $CO_2$ mole fraction. When the preferred amount of $CO_2$ free air is transferred into the low reference cylinder, the cartridges are decoupled from the system to transfer the rest of the air from the high reference cylinder to the low reference cylinder.

## Appendix E: Analysis cross-contamination

To determine whether cross-contamination has the potential to affect isotope measurements on the SICAS, a simulation was conducted in which we use the measurement procedure described in this paper. Input in the simulation is an by experiment derived value which expresses how much a measured sample is affected by the sample that was measured in the optical cell before. The experiment was conducted as follows: the high reference was measured 8 times in a row, each time letting in a new aliquot, followed by the low reference which was measured also 8 times in a row and this procedure was repeated 3 times. The usual flushing procedure was applied every time there was a switch between the cylinders. It can be expected that the first measurement of a series of 8 of the low reference is affected the most by the preceding measurement of the high reference gas. The last measurements of a series of 8 will be affected less, and will be closer to the 'true' value. We quantified this effect by applying the following equation to all series of measured isotopologues:

$$CC = \frac{M_1^* - mean(M_{6,7,8}^*)}{mean(M_{6,7,8}^*)} * 100 \tag{E1}$$

In which CC is the cross-contamination in percent, M stands for measurement, with * being the isotopologue and the number indicated at the underscore is the number of the measurement. The CC values we observed were low, ranging from indetecable up to 0.01% at most. We used this highest value for our simulation.

A simulation for a measurement sequence was set up in Excel, following the measurement procedure as described in this paper, only using 3 sample measurements per measurement series instead of 9. Included in the simulation are measurements of the low and high reference gas and two hypothetical samples with $CO_2$ concentrations of 480 and 340 ppm, a $\delta^{13}C$ values of -7 and -11‰ and $\delta^{18}O$ values of -1 and -4‰, respectively. All measurements are alternated with measurements of the working gas, according to the measurement procedure described in this paper. We use the actual values for $CO_2$ concentration and isotope composition of all reference gases in the simulation. The measurements were simulated by:

$$M_t^* = M_{t-1}^* * 0.01 * 10^{-2} + (1 - 0.01 * 10^{-2}) * Tr^* \tag{E2}$$

With $M_t^*$ being the simulated measurement at time t with * indicating which isotopologue measurement is simulated, $M_t - 1$ being the preceding simulated measurement and Tr being the true isotopologue abundance of the sample or reference gas that is being measured at time t. The first value that is put in the simulation contains the true values for all measured isotopologue abundances. For all sample measurements a normalized measurement is calculated by dividing $M*_t$ by $M*_{t-1}$ (being the working gas measurement).

These simulated, normalized measurements of the low and high reference gases are used to do a linear fit as a function of the true value, and so calculating the calibration curves. these curves are used to calculate the calibrated sample measurements, and the measured $_.13C$ and $\delta^{18}O$ measurements can be calculated. We find deviations from the measured simulation values of maximum 0.0002‰ for both $\delta^{13}C$ and $\delta^{18}O$, so two orders of magnitudes lower than the measurement precision.

*Author contributions.* P. S., H. S. and H. M. conceived the experiments, which were conducted by P. S. and H. S., P. S. carried out the data-analysis. D. N. and B. M. optimised the fit and contributed with technical assistance for development of the gas handling system as well as solving problems with the instrumentation. The manuscript was written by P. S., H. S. and H. M. contributed with discussions and comments throughout the writing process.

*Competing interests.* Authors D.N. and B.M. work for Aerodyne Research Inc. which is the company that developed the instrument described in this study.

*Acknowledgements.* We would like to thank H. M. Moossen, as well as his colleagues at the MPI-BGC, for measuring our reference cylinders and providing us with the data that we required for the intercomparison. We greatly acknowledge the help of G. Adnew from IMAU who prepared the samples for the first CMFD experiment and helped measuring them. He also measured the $\delta^{17}O$ composition of our working gas and quality control gas. We also want to thank M. de Vries, B.A.M. Kers and R. Ritchie for helping us to develop the sampling system as well as the the required software development, and D. Paul for general assistance. We thank the anonymous reviewer, D. Griffith and E. Flores for their valuable comments, as well as the associate editor T. Arnold for his contributions. This research was partly funded by the European Metrology Programme for Innovation and Research 16ENV06 "Metrology for Stable Isotope Reference Standards (SIRS)".

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
