# Peer review of "Simultaneous measurement of $\delta^{13}C$ , $\delta^{18}O$ and $\delta^{17}O$ of atmospheric $CO_2$ - Performance assessment of a dual-laser absorption spectrometer"

_Atmospheric Measurement Techniques, 2020_

## Referee Comment (RC1) · Anonymous Referee #1 · 10 Dec 2020

This is a good paper that deserves publication in AMT after revision. The authors did a great job in describing the mid-IR laser-based spectroscopic instrument for direct 17O-CO2 detection in air and reporting all relevant aspects of long-term operation and calibration of this instrument. The paper contains a lot of technical details and high-quality data. However, underlying spectroscopic principles are only partially covered. This might be outside the main scope of the paper, but some brief explanations are desirable.

The authors might consider addressing the following points:

[Figure]

1. The abstract of the paper ends with the sentence, which is only partially supported by the main text. Authors might want to elaborate improvements in measurement procedure, spec. fit and 17O calibration.

2. In recent years, significant progress has been made towards high-precision optical measurements of rare 17O-CO2 isotopologue: doi.org/10.1021/acs.analchem.7b03582, doi.org/10.1021/acs.analchem.7b02853, doi.org/10.1021/acs.analchem.9b03316. An overview of these works could be mentioned in the introduction.

3. Spectrometer's description still lacks some important details: a) what was the output power of ICLs? b) were measurements realised in static or flow-through mode? c) CO2 is known to be absorbed by aluminium, did authors encounter losses of gas in the optical cell? d) typical level of residuals, absorption line profiles, and spectroscopic line parameters are not mentioned in the text. Figure 2 might be improved by adding subplots with fit residuals and reporting noise level.

4. The authors tested the novel calibration scheme based on the isotopologue mole fraction and compare it with a conventional isotope ratio calibration. Several groups demonstrated successful application of the isotopologue mole fraction calibration recently (doi.org/10.1088/1681-7575/ab948c and doi.org/10.1002/rcm.8836). Not too much effort has been made towards the explanation of the advantages and disadvantages of both methods. Discussion on spectroscopic principles and limiting factors of the methods fit the scope of the paper. The discussion given in the end of 4.1 is very brief. Authors might want to expand this section.

5. This paper might attract more readers if it ends with a crisp recommendation summary on how to operate the laser-based isotope ratio spectrometer in practice. The findings reported in the paper are sufficient for this.

6. The paper would benefit from reduced use of abbreviations. I also suggest using roman typesetting for chemical formula and conventional (not AFGL) notation for
isotopologues, e.g., \ce{ˆ{12}Cˆ{16}O_2} instead of 626. Captions of the figures and tables might be extended for Figs. 1 – 3, and Tabs. 1 – 4.

7. Parts of the main text with technical details that are not directly related to the main subject of the paper, e.g., p.9 starting line 199, might be moved to the Appendix.

Some technical corrections: page 2, line 40: isobaric interferences of m/z = 46 with m/z = 45 ? Please elaborate. page 3, line 80: citation to out-dated HITRAN version. page 4, figure 1: label in the figure contradicts main text (QCL vs ICL). page 4, figure 2: typical level of fit residuals might be added here. page 5, figure 3: not all elements of the diagram are explained. page 9, line 201: here and throughout the text, mbar, not mBar or mBars page 9, line 291: cite doi.org/10.5194/amt-13-2797-2020 for matrix effect in mid-IR analysers page 14, line 295: purity <=99.99% ? page 16, table 5: briefly explain the errors page 17, line 364: cite doi.org/10.5194/amt-11-6189-2018 page 27, figure A1: no caption

---

## Referee Comment (RC2) · David Griffith (Referee) · 18 Dec 2020

**Review of Steuer et al., Simultaneous measurement of _13C, _18O and _17O of atmospheric CO2 - Performance assessment of a dual-laser absorption spectrometer.**

This paper provides a technically detailed description of a QCL laser-based analyser for the isotopic analysis of atmospheric $CO_2$, with a focus on calibration precision and accuracy. The increasing availability and usage of optical methods (as distinct from isotope-ratio mass-spectrometric methods, IRMS) for isotopic analysis of trace gases in air makes this study both timely and valuable. Optical methods provide many advantages for isotopic analysis not easily accessible by IRMS, such as minimal sample-preparation and continuous analysis in whole air at sub-minute timescales. However optical methods currently have lower precision and accuracy than IRMS, approaching but not meeting the compatibility requirements set out by WMO's Global Atmosphere Watch (GAW). This paper provides a detailed assessment of a state of the art laser spectrometer for precision and accuracy using two different approaches to calibration. It is clearly suitable, valuable and desirable for publication in AMT, but I do have some reservations about the detailed treatment of the calibration schemes as described in the submitted paper and suggest major revisions before publication. I also provide technical comments. When addressed I believe both will improve the paper substantially.

**General comments**

General comments and suggestions for revision are centred around the comparison between the ratio method (RM) and isotopologue method (IM) for the calibration strategy. These methods have been discussed and dissected in detail in the past few years, culminating at the 2017 GAW GGMT meeting in Dubendorf. Several papers from that time and earlier have now been published but, given the central focus of this paper on the calibration methods, they are not well presented or critically reviewed here. In particular for the more novel IM, papers by Wehr et al (Ag. For Met 2013, including authors common to this paper) Tans et al (AMT 2017) and Griffith, (AMT 2018) are not cited or discussed. (Disclosure: The reviewer is the author of Griffith, AMT 2018. See that paper for references to these and other relevant papers.) The analysis schemes used in those studies are equivalent to that of Flores et al (Anal Chem 2017 and Griffith 2012) which are referenced here. Griffith 2018 supplemented and extended Flores et al and included a review and analysis of the two calibration strategies. Given the central focus of ratio vs isotopologue calibration in this paper, I recommend adding a short but critical review (or at least a comprehensive literature survey) of existing work on calibration methods as they relate to the present work.

The paper is generally technically well-detailed and comprehensive in describing the analysis, after addressing technical comments and suggestions listed separately below. The major exception, based on the presented material, is the treatment of the isotopologue method (IM) for calibration. Since both RM and IM use the same raw measured data with the same uncertainties and noise characteristics, I would intuitively expect both calibration methods in principle to deliver the same accuracy and precision if applied with equal rigour. So I ask why the accuracy and precision as presented is significantly lower for IM than RM? I think the answer lies in the detailed application of the two treatments, with and without consideration of drift, non-linearity and concentration dependence in the calibrations, as described by Wen et al (2013, referenced):

- Non-linearity, due to imperfect spectroscopy and spectral fitting (eg imperfect knowledge of line shapes, pressure and temperature dependence, approximations in the spectrum model used for the spectra). Non- linearity is small, it applies to both RM and IM, but is only dealt

with for the RM, not IM. The section on concentration dependence (3.1) quantifies the non-linearity, but in the calibration section 3.3 non linearity is only considered and dealt with for RM, while IM is based on simple 2-point linear calibration for each isotopologue.
- Non-zero offset in the individual instrument response to each isotopologue. This is included explicitly in the linear calibrations for IM, and implicitly through the empirical non-linearity / concentration dependence correction in RM.
-

In RM, the sample is measured alternately with WG and QC gases, emulating practice in IRMS. The WG and QC gases provide both an interleaved constant reference R, eq. 5 and 6, as well as to derive and apply a real-time empirical concentration dependence, eq. 9, for each measurement sequence. This scheme appears to work well, taking account of both instrument drift and changing characteristics, as well as $CO_2$ dependence and non-linearity, continuously.

In IM, however, the description provided is much less detailed but from what is presented it appears that only a single linear 2-point calibration, determined once for each isotopologue,  is applied to relate the instrument response ($X_m$ in eq. 10) to the actual isotopologue amount ($X_a$). This ignores both non linearity and any change or drift in the calibration coefficients c and d with time, both of which are implicitly corrected in the RM approach. The root cause of any change in calibration with time is the same in both methods – it is embodied in the non zero intercept and quadratic terms in the effective calibration relationship between instrument response and actual isotopologue amount. As this section 3.3.2 is far less detailed than 3.3.1 and I have less information to work with, I invite the authors to correct me if I am wrong, but in any case extend the description of the method for IM in 3.3.2 to address this lack of detail.

I suspect this unequal treatment of calibration changes with time may be the root cause of the different precisions and accuracies found for the two methods in measurements of flasks and other tested samples. The measurements are of good quality, extensive and time consuming and presumably cannot be repeated easily. But I ask if the existing co-incident WG and QC measurements collected in each measurement run could be used to effectively re-derive or correct the coefficients c and d in eq 10 in real time to address stability and drift, as they are for RM through the R gas and $CO_2$ dependence corrections.

Until this is resolved, in the paper as written the comparison between RM and IM is not an equal one. When addressed, this may (or may not) change the final conclusions from the paper.  I suspect that the precision and accuracy for each method applied with equal rigour may be quite similar. The discussion would then centre around the relative measurement effort required by each method to achieve the required precision and accuracy rather than a simple one-is-better-than-the-other conclusion.

In the interest in providing an on-time review, I do not provide comments on the results, discussion and conclusions sections since they may change substantially.

**Technical comments**

Abstract: In the interest of readability, I think best editorial practice is to avoid abbreviations in the abstract, and to introduce them at first use in the main text. I agree with reviewer 1 that there is some over-use of abbreviations and acronyms. Depending on the response to the general comments above, the key conclusions from the study may change.

L27: Low variability is due to the large size of the carbon reservoir and the long lifetime of $CO_2$ in the atmosphere, not the high mole fraction. (Note: mole fraction is strictly the correct term, not mixing ratio)

L31: Can this reference to WMO 2016 be updated to the latest (from 2019 meeting).

L49: Optical methods to which this paper applies include non-laser methods such as FTIR and even NDIR. Remove (laser), it is in any case appropriately mentioned later in the sentence.

Section 1 General – see general comments, please include a literature survey and preferably some critical analysis of existing work in the calibration topic.

L65: replace "consists (among others of)" with "includes"

L68: In contrast to Reviewer #1 I am comfortable with the Hitran notation – it is well established (in optical spectroscopy at least) and quicker to type and read than full molecular labels, especially in equations.

L69: (with a sweep frequency of…) Add sweep to avoid potential confusion with optical frequency.

L74:     Please specify that this is the path inside the cell (the outside path is dealt with in the next paragraph, but it is ambiguous here)

L79: How accurate is the temperature control? This is relevant in assessing the causes of drift which are important to the whole calibration strategies. Likewise pressure control (see line 201).

Figure 3: requires a full caption with all labels and abbreviations spelt out. It is quite difficult to follow the text on p5 with the current level of detail provided. L91 – where is the inlet valve, for example?

L98: mbar not mBar, and please specify this is gauge pressure, not absolute.

L109 and Fig 4. What are the units of the Y axes of plots? For the upper plots they should be ppm$^2$ but I cannot relate this to the mimimum figures given in the in-plot tables. The lower plots are presumably ‰$^2$ . Also the abbreviation "St. dev"; these are not standard deviations, they are the minimum Allan deviations (ie sqrt(AllanVariance)), which is strictly not quite the same thing. Finally, unless the amounts for the isotopologues are scaled by their natural abundances, the Allan variances in ppm$^2$ will have quite different absolute values.

L143: please replace "n" with "number of measurements"

L143: should "lower" read "higher"?

L129: what is "demi-water"

L152: This is not a fair comparison, or is a misleading statement. It is only to be expected that ratios will show lower Allan Variance than individual isotopologues because some sources of noise and drift in the ratioed quantities are correlated and cancel. When isotopologue amounts are ratioed later in a calibration calculation, as in the IM, these correlated noise sources will also cancel. This should be made clear.

Table 1: These are relative standard deviations in ‰, please add "relative" to the caption and make clear in the case of the ratios that it is not the delta value itself but the relative std dev of it.

L180, eq. 3. Although this equation is only used for illustrative purposes, it is confused by the true calibration equation (10). The difference is between $X_{CO2}$ and $X_a$. $X_{CO2}$ is itself a function of $X_*$ (or $X_m$) through the relevant isotope ratios so the coefficients $a$ here and $c$ in Eq 10 are different. It would be clearer to use the same equation as (10) here.

L214: Griffith2018 showed that a pseudo linear + inverse relationship to $CO_2$ amount is both theoretically expected and fits data in that paper in practice. The quadratic fit is only an approximation to this.

Section 3.2.1 and Fig 5. I have difficulty following this section and figure. It may be better titled "Spectroscopic non-linearities of isotope measurements". If the "residuals" are the differences between a linear regression of $X_*$ against X626, they should be in units of ppm, not ‰ in Fig 5. I do not understand the sentence on L 223 "The CO2 mole fraction…"

L249: There should be no apostrophe in "deltas". An apostrophe is used to indicate possessive, not plural. There are many examples through the MS.

Fig 6. See comment for L214, it would be interesting to see these $\delta$ data plotted against $X_{CO2}$ and $1/X_{CO2}$. Does this improve the comparisons in Table 3? The description of this table is also difficult to follow in the text.

L245: Please clarify the sentence "From now on we use the $\delta$ with an isotopologue superscript…." and/or give an example. It is quite unclear to me.

L256-283: This section is important in the context of general comments above on correcting for stability, non linearity and drift. It points out that the calibration changes with time, but my understanding of the analysis is that this is compensated in RM but not IM. This might be clarified in dealing with earlier comments on the IM in 3.3.2.

3.3.2 Isotopologue method. Please see earlier general comments on this section. This does not have the same level of detail as 3.3.1 and if I understand it correctly the IM does not include the equivalent of the interleaved comparison with a reference gas R, or any correction for changes in the coefficients c and d in Eq 10. These changes are the root of any calibration drift in both RM and IM . As this section is written they appear to be corrected in RM but not IM.

4, 5, Results, discussion and conclusions. I do not comment on these sections because they may change substantially if the comments around the treatment of IM can be addressed.

---

## Referee Comment (RC3) · Edgar Flores (Referee) · 22 Dec 2020

General comments

Over recent years the introduction of optical systems based on various spectroscopic techniques has revolutionized stable isotope analysis in the atmosphere requiring a full calibration process with appropriate standards that are value assigned on internationally recognized scales, both for mole fraction and isotopic composition.

This paper proposes a new optical system and compares two calibration procedures.

[Figure]

Regarding the information on the instrument this paper provides straightforward technically detailed description of a QCL laser-based analyzer. However, the text about the injection of the samples into the gas cell misses very basic information (static or dynamic flow (l/min?), type of materials, treatments, flushing and gas handling procedures (cylinder and flasks) for example).

Concerning the calibration procedures this paper presents important conclusions on methods (RM and IM), but these conclusions are based on results which miss robust uncertainty analysis. Most of the uncertainties considered in the paper are statistical (precision), without proper consideration if systematic uncertainties can be discarded (for example from calibration). Additionally, important changes in the experimental conditions , significant decrease of about 50 percent in the measured laser intensity, happened and it is not clear how it influenced the results measured during the period 2017-2019.

This paper may be suitable for publication in AMT only after major revisions are to increase its robustness.

Technical comments:

1) Page 3, line 89: Please provide details on the type of materials, treatments on lines and valves to transfer the gas to the SICAS instrument. Is the gas cell under constant gas flow? If yes please provides the flow rate.

2) Page 5, line 91: Describe in detail the flushing procedure to avoid cross contamination providing evidence that it works (supplementary information).

3) Page 5, line 97: Please explain how you took into account the changes in the internal pressure of the cylinders and their potential fractionation effects? Instrument performance

4) Page 6, line 110: "machine working gas" is not defined. In Figure 3. The term Ref/Working gas is used. Is it the same?

[Figure]

5) Page 6, line 116 is stated: "The precision became significantly worse for all species but isotopologue 627 in the time period between September 2017 and July 2019 due to a gradual but significant decrease (of about 50And in Page 6, line 127 is stated: "Hence, we were able to clean the mirrors and retrieve âĹij 80Questions: o When exactly were the mirrors cleaned? o Which measures were affected? o What was the short and long term effect on the measurements with a timeline instrument response? o What was the effect of introducing moisture and ethanol for the cleaning of the mirrors, short and long term effect?

6) Page 9, line 199: the three experiments performed over the last two years means: before or after the cleaning?

7) Cross contamination and drift are only considered as uncertainties components on the instrument performance. However the instrument was used to measure cylinders and flask that could have important differences in matrix composition. What was the pressure broadening effect on the CO2 measurements?

8) What is the real contribution of various components of the air, temperature and pressure variability into the instrument? Uncertainty analysis is missing.

9) Page 8, line 160 is stated: "A sensitivity analysis was performed and showed that this is such a small amount that scale effects due to cross-contamination are well below the precisions found in this study" Where is the sensitivity analysis? This an important effect and evidence should be shown to sustain this statement.

10) How the uncertainty of the method is constrained by the uncertainty in the reference values of the CO2 mole fractions in the calibration standards?

11) What is the effect of diluting pure CO2? Uncertainty in the mole fractions related to this? What is the contribution of the loss of CO2 on the wetted surfaces and the emptying of the flasks?

12) Page 10, line 224: "..the known CO2 mole fraction of the working gas..": "mole

fraction" is the quantity referred to in the paper. The proper unit would be $\mu$mol/mol rather than ppm.

13) Figure 5 shows the experiment 1,2 and 3 that correspond to different measurements carried out in 2017, 2018 and 2019. The authors stated in page 6, line 116 that "the precision became significantly worse for all species but isotopologue 627 in the time period between September 2017 and July 2019 due to a gradual but significant decrease (of about 50 percent) in the measured laser intensity over that period". Why do the error bars in 2017, 2018 and 2019 measurements look the same for the mole fraction range 400 to 460 ppm?

14) Page 12, Figure 5: Why there no uncertainty bars in x-axis? Where are the confidence bands that could support the statements?

15) Page 14, line 269: is stated "In our lab CO2 in air samples of the same isotope composition but deviating CO2 mole fractions are prepared manually, introducing again uncertainties, and doing these experiments regularly is therefore labor- and time intensive". However no uncertainty budget (assessment) considering all the uncertainty contributors for the manual preparation is shown. As a minimum, it is likely that air composition affects CO2 measurements and this is recognized latter in the text, line 291.

16) Page 14, line 275: the reference cylinders are mentioned for the first time in this section but those are not identified (serial number) neither their composition (air matrix). The same issue with the air samples used.

17) Page 14, line 285: A brief description of the gaseous reference materials is given in this section 3.2 with additional information page 15, line 302 and Appendix B. Nevertheless it is essential to have as much as information as possible on the reference materials for allowing the readers to reproduce such experiments.

18) Page 14, line 287: The author specify that "two tanks that are specifically used

for CMFD corrections. These latter two consist of a high mole fraction reference tank (HR) and a low mole fraction reference tank (LR) covering a great part of the CO2 mole fraction range occurring in atmospheric samples"

Such range is: 342.8 to 423.77 ppm. However in Page 9, line 195 the author state: The SICAS is designed for the measurement of atmospheric samples of which the relevant range of CO2 mole fractions is 370 – 500 ppm, and experiments were therefore for the most part conducted in this range.

The author can only warrant that the CMFD corrections of the instrument following the method proposed is valid for the range 342.8 to 423.77 ppm, not higher since there is no evidence for that.

19) Page 15, line 317: It will be important for the robustness of this paper to list, and assign an experimental uncertainty, to each of the uncertainty sources cited in this section "small leakages or other gas handling problems might be introduced".

20) Page 15, line 304 . The claimed standard uncertainties on the gas tanks that were produced in-house from dry compressed natural air, "HR 423.77±0.01 ppm" and "LR 342.81±0.01ppm", are very unlikely (considering that Y = y ± U).

During the CCQM K120.a international comparison only one laboratory submitted comparable uncertainties (NMIJ) but considering potential effects of adsorption of a proportion of the molecules onto the internal surface of a cylinder and valve a limit on the uncertainties claimed by participants contributing to the reference values on this comparison was fixed to 0.095 $\mu$mol mol-1 meaning that any uncertainty claim smaller to this value was replaced by 0.095 $\mu$mol mol-1 to calculate reference values.

The claimed standard uncertainties from the gas tanks in this work are even half of the NOAA (real air mixtures) submitted uncertainty for the CCQM‐K120.a international comparison.

(https://www.bipm.org/utils/common/pdf/final$_reports/QM/K120/CCQM-K120.pdf$)

21) Page 16, Table 5. define what are CO2 err, $\delta$13C st. err., $\delta$18O st. err. and their units. It appears very unlikely that calibrated reference materials of CO2 in air are provided with an uncertainty equal to 0.01 $\mu$mol mol-1. Therefore CO2 err is most likely a precision. Please justify why the precision is the only uncertainty contributor that maters by proper considerations on the goal of the experiment.

22) Page 17, line 368: The statement "The mole fraction (X) of the four most abundant isotopologues of a measured CO2 sample are determined using two references gases with known CO2 mole fractions and isotope compositions" must be completed by including:

- A phrase stating that the measurements are only valid for the range of 342.81 ppm to 424.52 ppm (according to Table 5, page 16) since the standards must bracket each of the three expected isotopologue mole fractions in the samples;

- an uncertainty budget including at least two components, the first related to the repeatability of the measurement results (MPI-Jena), and the second related to the stability and homogeneity of the isotope ratio values in different standards containing nominally the same CO2 gas;

- It is also crucial when proposing a calibration procedure to consider the uncertainty of the calibration standards. The uncertainties on the reference isotopologue mole fraction values need to be first estimated, and then used in an uncertainty budget reflecting a two point calibration process for each isotopologue.

23) The term "reference gases" is not clear into the document. It is first mentioned in

- Page 17, line 369; "The CO2 mole fractions are chosen such that normally occurring CO2 mole fractions in atmospheric air are bracketed by the two reference gases"

- in Page 17, line 348: "...We developed a calibration method based on the idea that including the measurement of two reference gases covering the CO2 range of the measured samples (in our case LR and HR) enables the correction of the measured

isotope ratios. . .",.. then;

- In Page 17, line 371: "..Due to the broad range of CO2 mole fractions that are cov-
ered by the reference gases, measurement of both working standards will enable the
calculation of the (linear) relation of the measured mole fraction (Xm) and the Xa,.." ..
then in;

- Page 24,line 506: "..we used natural air as reference gases (or air mixtures close to
natural air)"

- but then in Page 5, Figure 3: Ref 1 and Ref 2 appears without being defined elsewhere
in the document and the Ref/Working gases term as well.

So are Ref 1 and Ref 2, the reference gases? Same as listed in Table 5 as LR and HR
(Page 16)?

Which gases are the Ref/Working gases?

Please define; reference gases, measured samples, reference cylinders, calibration
gas mixtures, flask samples, HR cylinder, LR cylinder, Ref Working gases and ma-
chine working gas.

24) Page 17, line 370: Is Appendix 1 or Appendix A?

25) Page 18, Table 6. No units?

26) Page 19, line 407: when was the sausage series 90-94 measured in 2020 (month
exactly?)

As described in Page 20, line 434. There was an important long term effect of the
aliquot storing during 3 to 20 months for $\delta$18O measurements and this could strongly
influence the conclusions of this intercomparison.

27) Page 20, Table 7. No units? Why comparing the difference SICAS-MPI with NOAA-
MPI? Which technique was used by NOAA?

28) Page 22, line 440: figure 8 shows results outside the mole fraction range validated in this work (343-425ppm). No conclusions can be stated for measurements outside the range.

29) Page 24, line 491: The author state "In this study we show that WMO compatibility goals can be reached with our Aerodyne dual-laser absorption spectrometer for stable isotope measurements of atm-CO2 in dry whole air samples if the instrumental conditions are optimal and there is no uncertainty induced because of gas handling procedures (flask sampling for instance)" but unfortunately no uncertainty budget was shown in the paper to underpin this statement.

30) Page 24, 496: The author state "Non-linear dependencies on the CO2 mole fraction occur for measured isotopologue abundances but are insignificant in the typical ambient CO2 mole fraction range" This is relatively true if compared to the compatibility goals.

31) Page 24, line 510: The author state "From studying the results of the QC we conclude that precisions are significantly better for the RM, while measurement stability is very similar, both for the $\delta$ 13C and the $\delta$ 18O measurements.". With the current version of the paper, there is no evidence to show that the isotopologue method was applied correctly, using proper reference materials and proper measurement sequences which would ensure frequent drift corrections and two points calibration for each isotopologue. If the isotopologue method is kept after revision, and if the same observations are made, consider explaining why the two methods provide different precisions.

---

## Author Comment (AC1) · 19 Feb 2021

February 19, 2021

Response to reviewer comments for manuscript "Simultaneous measurement of  $\delta^{13}C$ ,  $\delta^{18}O$  and  $\delta^{17}O$  of atmospheric  $CO_2$  - Performance assessment of a dual-laser absorption spectrometer

We want to thank the reviewer for the constructive comments for the final version of our paper. In this document we will address the points raised by the reviewer. We use **bold text** for repeating the points of the reviewer, normal text for our answers, and *italics text* for changes made in the manuscript. Page and rule numbers apply to the new version of the manuscript.

1. The abstract of the paper ends with the sentence, which is only partially supported by the main text. Authors might want to elaborate improvements in measurement procedure, spec. fit and 17O calibration. We agree that the last sentence of the abstract is not supported by the text. We therefore removed the last sentence of the abstract. In the most recent version of the paper we included a more robust calibration of the  $\delta^{17}O$  measurements by using assigned values, which were directly and indirectly derived from measurements at the IMAU. Also, due to adjustments that were made for the IM calibration method, precisions of the  $\delta^{17}O$  improved significantly and are now closer to the required precision over the whole measurement period.

p.26 line 565: "These precisions are now not yet achieved, but the results of the IM calibrated values show that small improvements in the measurement precision of the SICAS can bring the  $\Delta^{17}O$  measurements close to the 0.01 precision. This could for instance be accomplished by deciding to conduct more iterations per measurement, if sample size allows this. In section 2.2 the contamination of the mirrors was discussed as the potential cause for the decreased signal-to-noise ratio in over the period September 2017-July 2019. Placing new mirrors in the optical cell might therefore improve the quality of the measurements further. As the quality of the  $\Delta^{17}O$  measurements depends directly on the quality of the the  $\delta^{18}O$  and the  $\delta^{17}O$  measurements, it will be important to monitor the measurement quality of both isotope values over time using the measurements of the quality control gas. If SICAS measurements are to be used for comparison with  $\Delta^{17}O$  measurements from other labs or measurement devices, it is necessary to add the error introduced by the scale uncertainties of the reference gases as well.

For both the  $\delta^{17}O$  and  $\Delta^{17}O$  these uncertainties are 0.08, as calculated with a Monte-Carlo simulation as described in section 4.2. As long as only measurements from this device are used, seasonal and diurnal cycles are measured with much lower uncertainties. The high residuals found for the quality control gas measurements of the  $\delta^{17}O$  and  $\Delta^{17}O$ show that these uncertainties are probably an underestimation, as the assigned values of the low and high reference, which were not directly measured at the IMAU, are not known with high accuracy. For reducing the combined uncertainty it is therefore crucial to have all reference gases directly determined for their  $\delta^{17}O$  and  $\delta^{18}O$  values of the reference tanks. "

2. In recent years, significant progress has been made towards high-precision optical measurements of rare 17O-CO2 isotopologue: doi.org/10.1021/acs.analchem.7b03582,

doi.org/10.1021/acs.analchem.7b02853,

doi.org/10.1021/acs.analchem.9b03316. An overview of these works could be mentioned in the introduction.

p. 3 line 65: "Recent studies already showed the effectiveness of optical spectroscopy for the measurement of  $\delta^{17}O$  in pure  $CO_2$  for various applications (Sakai et al., 2017; Stoltmann et al., 2017; Prokhorov et al., 2019)."

- 3. (a) a) what was the output power of ICLs?
  - (b) were measurements realised in static or flow-through mode? Measurements were realised in static mode. p. 4, line 99: "The gas inlet system, depicted in figure 3, is designed to measure discrete air samples in static mode, such that one can quickly switch between measurements of different samples."
  - (c) CO2 is known to be absorbed by aluminium, did authors encounter losses of gas in the optical cell? We are aware of this characteristic of the cell, and loss of  $CO_2$  is observed when letting a  $CO_2$ -in-air mixture into the cell, after the cell was flushed with a  $CO_2$ -free flush gas (in this case  $N_2$ ). We did, however, conclude that these effects are negligible for measurements conducted in the atmospheric range. If samples of strongly deviating  $CO_2$ mole fractions or isotope compositions are measured, scale contraction might occur. We believe, as this paper investigates the potential to measure atmospheric samples, investigations of this effect is outside of the scope of the paper. An extra sentence has

been added to the manuscript text regarding the surface adsorption effects. page 9 line 174: If samples of  $CO_2$  concentrations outside the range of atmospheric samples are measured it will be essential to also take into account the surface adsorption effects of the aluminum cell (Leuenberger et al., 2015), an effect that was observed clearly when measuring atmospheric samples right after the cell was flushed with ( $CO_2$ -free)  $N_2$  gas.

- (d) typical level of residuals, absorption line profiles, and spectroscopic line parameters are not mentioned in the text. Figure 2 might be improved by adding subplots with fit residuals and reporting noise level. Noise levels of the four relevant isotopologues are shown in figure 4 (upper panels). Figure 2 is adjusted and now also includes the residuals of the fits.
- 4. The authors tested the novel calibration scheme based on the isotopologue mole fraction and compare it with a conventional isotope ratio calibration. Several groups demonstrated successful application of the isotopologue mole fraction calibration re- cently (doi.org/10.1088/1681-7575/ab948cand doi.org/10.1002/rcm.8836). Not too much effort has been made towards the explanation of the advantages and disadvantages of both methods. Discussion on spectroscopic principles and limiting factors of the methods fit the scope of the pa-The discussion given in the end of 4.1 is very brief. per. Authors might want to expand this section. A general introduction before section 3.3.1 has been added to explain the main advantages/disadvantages of both calibration methods. page 17, line 355: "Two different calibration strategies are discussed in this section. The calibration strategies are based on the two main approaches for calibration of isotope measurements, as also described by Griffith et al. (2012) and, more recently by Griffith (2018), being (1) determine the isotopoloque ratios, and calibrate those, taking the introduced CMFD into account, from now on defined as the ratio method (RM), and (2) first calibrate the absolute isotopologue mole fractions individually and then calculate the isotopologue ratios, from now on defined as the isotopologue method (IM). We give a brief introduction of the two calibration methods, as described in literature and we describe the measurement procedure that is used for both calibration methods. This section ends with a detailed description of both methods as applied for the SICAS measurements. The RM, being very similar to calibration strategies applied by

isotope measurements using DI-IRMS (Meijer, 2009), is usually based on reference gases covering delta values of a range which is similar to the range of the measured samples. Determination of the CMFD can be done by measuring different tanks of varying  $CO_2$  mole fractions or by dynamical dilution of pure  $CO_2$  with  $CO_2$  free air (Braden-Behrens et al., 2017, Sturm et al., 2012, Griffith et al., 2012; McManus et al., 2015; Tuzson et al., 2008), again covering the  $CO_2$  mole fraction range of the measured samples. The IM has the advantage that there is no need to take the introduced CMFD into account (Griffith, 2018). As all isotopologues are calibrated independently, it is only necessary to use reference gases covering the range of isotopologue abundances as occurring in the samples. This can be realised by using reference gases containing  $CO_2$  of similar isotope composition but varying  $CO_2$  mole fractions (Griffith et al., 2012; Griffith, 2018; Flores et al., 2017). The range of delta values that is measured in samples of atmospheric background air is limited (range in unpolluted troposphere is -9.5 to -7.5 en -2 to +2 for  $\delta^{13}C$  and  $\delta^{18}O$ , respectively (Crotwell et al., 2020)), hence this also applies to the range of delta values that should be covered by the reference gases when applying the RM. We decided therefore to use the same reference gases to test both calibration methods, varying mainly in  $CO_2$  mole fraction (342.81-424.52  $\mu mol/mol^{-1}$ )."

Additional discussion on performances of both methods has been added to section 4.3. page 23 line 522: "To check the performance of the SICAS for both the IM and RM over the wide  $CO_2$  range that is covered by the ICP sausage samples, the differences between the MPI-BGC and the SICAS results are plotted in figure 8 against the measured  $CO_2$ mole fraction. Shown is that for both methods the highest differences are seen at the higher end of the  $CO_2$  mole fraction above 425 ppm, and therefore far out of the range that is covered by the HR and LR cylinders ( $\sim 343-425$  ppm). Extrapolation of the calibration methods outside the  $CO_2$  mole fraction range of the reference cylinders yields worse compatibility with MPI-BGC, possibly due to the non-linear character of both the isotopologue  $CO_2$  dependency and the ratio  $CO_2$  dependency. It should therefore be concluded that, to achieve highly accurate results of isotope measurements over the whole range of  $CO_2$  mole fractions found in atmospheric samples, the range covered by the reference cylinders would ideally be changed to 380-450 ppm. The results of the IM are slightly better in the  $CO_2$  range above 425 ppm, specifically the point closest to 440 ppm shows a significantly smaller residual  $(\sim 0.1 \text{ less})$  than the RM. The better result of extrapolation of the determined calibration curves for the IM method could be due to the lesser degree of non-linearity of the measured isotopologue abundances as a function of the assigned isotopologue abundances, in comparison to the non-linearity of the measured isotope ratios as a function of the  $CO_2$ mole fraction. More points in this higher range are needed, however, to draw any further conclusions on this matter. "

Also in section 4.4 additional discussion on performances of both methods has been added. page 25 line 550: "All results show too enriched values according to the assigned values, which is probably due to the fact that the assigned  $\delta^{17}O$  values of the low and high references have been determined indirectly, as discussed in section 3.2. A direct determination of the  $\delta^{17}O$  values of our low and high references would supposedly improve the accuracy of both methods. The  $\Delta^{17}O$  accuracy is dependent on both the  $\delta^{17}O$  and  $\delta^{18}O$  results, where  $\Delta^{17}O$  values will deviate more if those results deviate in opposite directions and vice versa. Furthermore, it is striking that the mean standard errors of measurement periods 2 and 3 are twice as low for the IM than for the RM. The  $r^{627}$ , used for the RM, is calculated by dividing  $X_{627}$ , derived from laser 1, by  $X_{626}$  derived from laser 2. It can be that the two lasers do not drift in the same direction and the advantage of cancelling out these drifts by dividing the two measured values will not apply. The outlier analysis of the IM might in that case be more effective as it is performed on both the measured 16O and 17O abundances, while for the RM it is only performed on the  $r^{627}$ . A comparison of the correlation coefficients between the 627 peak results and the 626 peak results from both lasers shows no significant difference (and a value of  $\sim 0.65$ ), meaning that using the 626 peak of laser 1 for the  $\delta^{17}O$  calibration will not improve the precision of the RM results. "

5. This paper might attract more readers if it ends with a crisp recommendation summary on how to operate the laser-based isotope ratio spectrometer in practice. The findings reported in the paper are sufficient for this. We added all calculation steps for both calibration methods in the appendix. In addition, a section on the used measurement procedure was added to the text, so future users will have a better view on how to use this device in practice. page 17 line 376: "The measurement procedure that is used for both calibration methods is based on the alternating measurements of samples/reference gases and the WG, so the drift corrected measurement value can be calculated as in equation 1. Per sample/reference gase measurement,

there are 9 iterations of successive sample and WG measurements, from now on called a measurement series, before switching to the next sample/reference gas measurement series. One measurement series lasts ~ 30 minutes. Sample series are conducted once, while the reference gases series (LR and HR) are repeated 4 times throughout a measurement sequence. The QC, a gas of known isotope composition which is not included in the calibration procedure, is also measured 4 times throughout the measurement sequence. One measurement sequence in which 12 samples are measured lasts therefore ~12 hours. For the 9 measurement values of each measurement series outliers are determined using the outlier estimation method for very small samples by Rousseeuw and Verboven (2002), and the mean values of the measurement series are calculated. For a complete step-by-step guide of all calculation steps for both calibration methods, please see Appendix C"

- 6. The paper would benefit from reduced use of abbreviations. I also suggest using roman typesetting for chemical formula and conventional (not AFGL) notation for isotopologues, e.g.,  ${}^{12}C^{16}O_2$  instead of 626. Captions of the figures and tables might be extended for Figs. 1 3, and Tabs. 1 4. The use of abbreviations has been reduced, for instance by not abbreviating the names of the reference cylinders anymore. We continue to use the HITRAN notation, as this is in our view an accepted notation in the field of spectroscopy and is much shorter. The captions of figures 1-3 have been extended, as well as for tables 1-4. Figure 1 has been changed to a more schematic view of the components on the SICAS optical bench for clarity.
- 7. Parts of the main text with technical details that are not directly related to the main subject of the paper, e.g., p.9 starting line 199, might be moved to the Appendix. The following section has been moved to Appendix A: "The pure CO2 aliquots were prepared by connecting a 20 mL flask containing a pure CO2 local reference gas to a calibrated adjustable volume. The required amount of CO2 in the adjustable volume could be determined by measuring the pressure at a resolution of 1 mBar using a pressure sensor (Keller LEO 2). Both the sample flask and adjustable volume were connected to a vacuum (3.3 \* 10-5 mBar) glass line. The CO2 in the adjustable volume was transferred cryogenically (using liquid nitrogen) into a small glass tube shape attachment on the side of the evacuated sample flask which was custom-made for this purpose and subsequently the zero-air dilutor gas was added. The dilutor gas consists of natural air scrubbed of

 $CO_2$  and  $H_2O$  using Ascarite® (sodium hydroxide coated silica, Sigma-Aldrich) and Sicapent® (phosphoric anhydride, phosphorus(V) oxide), which results in dry,  $CO_2$ -free natural air. For experiment 2, additional samples were prepared using synthetic air mixtures with and without 1% Argon as dilutor gas for evaluation of the effect of air composition on the CMFD (see also section 3.1.6). After closing the flask, the mixture was put to rest for at least one night before measurement to ensure the  $CO_2$  and the dilutor were completely mixed. "

Some technical corrections:

page 2, line 40: isobaric interferences of m/z = 46 with m/z = 45? Please elaborate What we mean to say here is that the  ${}^{12}C^{17}O^{16}O$  has the same mass as  ${}^{13}C^{17}O^{16}O$  and will interfere when doing IRMS measurements. Idem for the  ${}^{13}C^{17}O^{16}O$  and  ${}^{12}C^{17}O^{17}O$  with  ${}^{12}C^{18}O^{16}O$ . We do believe this is stated clearly and correctly and therefore no changes to the text are made.

page 3, line 80: citation to out-dated HITRAN version. This reference is updated to Gordon et al., 2017.

page 4, figure 1: label in the figure contradicts main text (QCL vs ICL). The label in the figure is adjusted.

page 4, figure 2: typical level of fit residuals might be added here. Fit residuals are added to the figure.

page 5, figure 3: not all elements of the diagram are explained. The caption of the figure is extended and a legend is added to the figure to explain all elements of the diagram.

page 9, line 201: here and throughout the text, mbar, not mBar or mBars Is adjusted as suggested.

page 9, line 291: cite doi.org/10.5194/amt-13-2797-2020 for matrix effect in mid-IR analysers This reference is added, and the text has been changed slightly. page 15 line 308 in modified manuscript : It is known that for laser spectroscopy the composition of the sample air affects the absorption line profiles by pressure broadening effects ("matrix effects"), with non-negligible consequences (Nakamichi et al., 2006, Nara et al., 2012, Harris et al., 2020). Hence, it is likely that air composition affects  $CO_2$  isotope measurements for the SICAS as well. page 15, line 311: purity  $\langle =99.99\%$ ? page 15, line 313 in modified manuscript changed to purity  $\rangle = 99.99\%$

**page 16, table 5: briefly explain the errors** Errors are now explained in the table caption.

page 17, line 364: cite doi.org/10.5194/amt-11-6189-2018 This reference has been added in section 3.3.

Sincerely,

Farilde Steur

---

## Author Comment (AC2) · 19 Feb 2021

**Response to D. Griffith's comments for the manuscript "Simultaneous measurement of $\delta^{13}C$, $\delta^{18}O$ and $\delta^{17}O$ of atmospheric $CO_2$ - Performance assessment of a dual-laser absorption spectrometer**

We want to thank David Griffith for thoroughly reading the manuscript and giving important comments. We used his comments to adjust the manuscript, which changed, and we think improved, considerably thanks to his constructive review.

In this document we will address all the points that were raised. We use **bold text** for repeating the points of the reviewer, normal text for our answers, and *italics text* for changes made in the manuscript. Page and rule numbers apply to the new version of the manuscript.

General comments

**The reviewer suggests to add a literature overview of existing work on calibration methods as they relate to the present work.** An introduction in both calibration methods is added in section 3.3, including most of the work that was suggested by the reviewer. *page 17 line 355: "Two different calibration strategies are discussed in this section. The calibration strategies are based on the two main approaches for calibration of isotope measurements, as also described by Griffith et al. (2012) and, more recently by Griffith (2018), being (1) determine the isotopologue ratios, and calibrate those, taking the introduced CMFD into account, from now on defined as the ratio method (RM), and (2) first calibrate the absolute isotopologue mole fractions individually and then calculate the isotopologue ratios, from now on defined as the isotopologue method (IM). We give a brief introduction of the two calibration methods, as described in literature and we describe the measurement procedure that is used for both calibration methods. This section ends with a detailed description of both methods as applied for the SICAS measurements. The RM, being very similar to calibration strategies applied by isotope measurements using DI-IRMS (Meijer, 2009), is usually based on reference gases covering delta values of a range which is similar to the range of the measured samples. Determination of the CMFD can be done by measuring different tanks of varying $CO_2$ mole fractions or by dynamical dilution of pure $CO_2$ with $CO_2$ free air (Braden-Behrens ewt*

*al., 2017; Sturm et al., 2012; Griffith et al., 2012; McManus et al., 2015; Tuzson2008, again covering the $CO_2$ mole fraction range of the measured samples. The IM has the advantage that there is no need to take into account the introduced CMFD (Griffith, 2018). As all isotopologues are calibrated independently, it is only necessary to use reference gases covering the range of isotopologue abundances as occurring in the samples. This can be realised by using reference gases containing $CO_2$ of similar isotope composition but varying $CO_2$ mole fractions as described in Griffith (2018) and successfully implemented in Griffith et al. (2012), Flores et al. (2017) and Wehr et al. (2013). The range of delta values that is measured in samples of atmospheric background air is limited (range in unpolluted troposphere is -9.5 to -7.5‰ en -2 to +2‰ for $\delta^{13}C$ and $\delta^{18}O$, respectively (WMO, 2016), hence this also applies to the range of delta values that should be covered by the reference gases when applying the RM. We decided therefore to use the same reference gases to test both calibration methods, varying mainly in $CO_2$ mole fraction (342.81-424.52 µmol/mol$^{-1}$)."*

**The reviewer asks why the accuracy and precision as presented is significantly lower for IM than for RM.** We agree that intuitively the precision should be similar for the RM and IM. We realised that the difference that was shown in the original manuscript was caused by the way the standard deviation of the measurements were calculated. The ratio method is more successful in reaching high precisions because using ratios has the advantage that drifts, that occur at the same rate and direction for both isotopologue abundance measurements, are cancelled out. The standard deviation of a ratio measurement will therefore be, in most cases, more stable than the measurement of individual isotopologue abundances. In the original manuscript the standard deviation of the delta values derived with the isotopologue method were calculated by:

$$\sqrt{sd_{626}^2 + sd_{rare}^2} \tag{1}$$

With $sd_{626}$ being the standard deviation the 626 isotopologue measurement and $sd_{rare}$ being the standard deviation of the rare isotopologue measurement. This seemed to be logical as all isotopologues are calibrated individually, and we treated the uncertainties of those isotopologues as different components when calculating the measurement precision. The reviewer points out in his report, as in the comment for L152: "It is only to be expected that ratios will show lower Allan Variance than individual isotopologues because some sources of noise and drift in the ratioed quantities are correlated and cancel. When isotopologue amounts are ratioed later in a calibration calculation, as in the IM, these correlated noise sources will also cancel. This

should be made clear." This statement is true, and we changed the calculation of the measurement precision accordingly. There are 9 iterations of a sample measurement conducted, and the mean of those 9 measurements will eventually be expressed as a delta value. Initially, we used the mean of the individually calibrated isotopologues for the calculation of one delta value. Now, we use all 9 measurements to calculate 9 delta values, and we calculate the mean of those 9 delta values. The measurement error can therefore be derived by calculating the standard deviation of those 9 delta values. This improves the measurement precision significantly, resulting in similar measurement precisions for $\delta^{13}C$ and $\delta^{18}O$ for the RM and IM (see table 6 in the modified manuscript). (This is no surprise, really, as now the difference between the IM and RM data treatment is much smaller than in our original implementation of IM.) For the $\delta^{17}O$ results we even observe a better measurement precision than the RM. We argue that this difference is possible by a low correlation between the 627 and 626 measurement results. In that case, the IM shows to perform better due to a more effective outlier analysis. All calculation steps for both calibration methods are added to the manuscript in Appendix C.

The accuracy results from the intercomparison with MPI-BGC results showed to be worse for the IM in the original manuscript. This is, as the reviewer suggests, due to the non-linearity which was dealt with for the RM, but not for the IM. In the first part of the manuscript, our study on the $CO_2$ dependency showed non-linearities of both the rare isotopologue abundances as a function of the measured abundance of the abundant isotopologue, and the measured isotopologue ratio as a function of the measured $CO_2$ mole fraction. These studies were mainly focussed on getting an idea about the $CO_2$ dependency of the isotope ratios and we applied these findings of non-linearity for determining the $CO_2$ correction for the RM. Although a study of the non-linearities of measured isotopologues as a function of the true isotopologue abundance has not been performed ($CO_2$ samples that were used for the dependency experiments were not determined for their $CO_2$ mole fraction), this will occur as the result of 'imperfect spectroscopy and spectral fitting', as stated by the reviewer. We therefore adjusted the determination of the calibration curves for the IM by using the hypothetical measurement of the working gas, which will always have a drift corrected value of 1, for doing a quadratic fit instead of a linear fit using only the low and high reference measurements. This reduced the residuals of the SICAS measurement using the IM and the MPI-BGC results considerably.

[Figure]

Results intercomparison in the original manuscript

[Figure]

Results intercomparison in the adjusted manuscript

The reviewer thereby suggests that for the RM real-time empirical concentration dependency is determined, while for the IM merely one two-point calibration curve was determined per isotopologue. Actually, we use the mean of the isotopologue/delta measurement of the reference gases over a whole measurement sequence for determining the calibration curves of both the RM and the IM. We think this approach is more practical than a real-time calibration as a deviating measurement of a reference gas will not affect the sample results as this measurement is usually filtered out by the outlier analysis. We added all calculations applied for both methods in Appendix C for clarification. We stick to this approach in the modified manuscript.

Technical comments

**Abstract: In the interest of readability, I think best editorial practice is to avoid abbreviations in the abstract, and to introduce them at first use in the main text. I agree with reviewer 1 that there is some over-use of abbreviations and acronyms. Depending on the response to the general comments above, the key conclusions from the study may change.** Abbreviations are now avoided in the abstract. We reduced the use of abbreviations in the text by writing out the full names of the calibration tanks as we agree this contributes to the readability of the text. The conclusions have changed as a result of the general comments that were given by the reviewer as described above. *Main conclusion that were adjusted in the manuscript, page 1, line 9: "Measurements of the quality control tank show that the measurement precision and accuracy of both methods is of similar quality for $\delta^{13}C$ and $\delta^{18}O$ measurements. In optimal measurement conditions the precision and accuracy of the quality control tank reach WMO compatibility requirements, being 0.01‰ for $\delta^{13}C$ and 0.05‰ for $\delta^{18}O$, respectively. Uncertainty contributions of the scale uncertainties of the reference gases add another 0.03 and 0.05‰ to the combined uncertainty of the sample measurements. Hence, reaching WMO compatibility for sample measurements on the SICAS requires reduction of the scale uncertainty of the reference gases used for calibration. An inter-comparison of flask samples over a wide range of $CO_2$ mole fractions has been conducted with the Max Planck Institute for Biogeochemistry resulting in a mean residual of 0.01 and -0.01‰ and a standard deviation of 0.05 and 0.07‰ for the $\delta^{13}C$ measurements calibrated using the ratio method and the isotopologue method, respectively. The $\delta^{18}O$ could not be compared due to depletion of the $\delta^{18}O$ signal in our sample flasks because of too long storage times. Finally, we evaluate the potential of our $\Delta^{17}O$ measurements as a tracer for gross primary production by vegetation through photosynthesis. Here, a measurement precision of <0.01‰*

*would be a prerequisite for capturing seasonal variations in the $\Delta^{17}O$ signal. The isotopologue method performs better for the measurement precision of the $\delta^{17}O$ and $\Delta^{17}O$ with standard errors not exceeding 0.02‰, showing that the IM is close to reaching the high precision requirement for capturing seasonal trends in the $\Delta^{17}O$ measurements. The accuracy results show consequently too enriched results for both the $\delta^{17}O$ and $\Delta^{17}O$ measurements for both methods. The ratio method shows residuals ranging from 0.06 to 0.08‰ and from 0.06 to 0.1‰ for the $\delta^{17}O$ and $\Delta^{17}O$ results, respectively. The isotopologue method shows residuals ranging from 0.04 to 0.1‰ and from 0.05 and 0.13‰ for the $\delta^{17}O$ and $\Delta^{17}O$ results, respectively. Direct determination of the $\delta^{17}O$ of all reference gases would improve the accuracy of the $\delta^{17}O$, and thereby of the $\Delta^{17}O$ measurements."*

**L27: Low variability is due to the large size of the carbon reservoir and the long lifetime of CO2 in the atmosphere, not the high mole fraction. (Note: mole fraction is strictly the correct term, not mixing ratio)** *Page 2, line 35: "Due to the large size of the carbon reservoir of the atmosphere and the long lifetime of $CO_2$ in the atmosphere, the effects of sources and sinks on the atmospheric composition are heavily diluted."*

**L31: Can this reference to WMO 2016 be updated to the latest (from 2019 meeting).** The reference is updated to Crotwell et al., 2020, in the whole text

**L49: Optical methods to which this paper applies include non-laser methods such as FTIR and even NDIR. Remove (laser), it is in any case appropriately mentioned later in the sentence.** *Page 3 line 58: "Optical spectroscopy now offers this possibility following strong developments in recent years especially for the laser light sources, to perform isotopologue measurements showing precisions close to, or even surpassing IRMS measurements (Tuzson et al., 2008; Vogel et al., 2013; McManus et al., 2015). "*

**L65: replace "consists (among others of)" with "includes"** *Page 3 line 77: "The optical bench as depicted in figure 1 includes the two lasers, several mirrors to combine and deflect the laser beams, the optical cell and two detectors. "*

**L68: In contrast to Reviewer 1 I am comfortable with the Hitran notation – it is well established (in optical spectroscopy at least) and quicker to type and read than full molecular labels, especially in equations.** We stick to the HITRAN notation in the paper, as suggested here.

**L69: (with a sweep frequency of. . . ) Add sweep to avoid potential confusion with optical frequency.** *Page 3 line 81:"Application of a small current ramp causes small frequency variations so the lasers are swept (with a sweep frequency of 1.7kHz) over a spectral range in which ro-vibrational transitions of the isotopologues occur with similar optical depths (Tuzson et al., 2008)."*

**L74: Please specify that this is the path inside the cell (the outside path is dealt with in the next paragraph, but it is ambiguous here)** *Page 3 line 86:"The total path length of the laser light in the optical cell is 36 meters."*

**L79: How accurate is the temperature control? This is relevant in assessing the causes of drift which are important to the whole calibration strategies. Likewise pressure control (see line 201).** We determined the maximum fluctuations of the cell temperature over a measurement period of 12 hours at normal circumstances. The sentence was adjusted additionally, as it stated that the the chiller controlled the temperature inside and outside the housing, which is incorrect, it only controls the temperature inside the housing. *Page 3 line 90: "The temperature within the housing is controlled using a re-circulating liquid chiller set at a temperature of $20^\circ C$ to keep the temperature in the cell stable. Within a measurement sequence (12 hours) the temperature does normally not fluctuate more than $0.05^\circ C$."*

**Figure 3: requires a full caption with all labels and abbreviations spelt out. It is quite difficult to follow the text on p5 with the current level of detail provided. L91 – where is the inlet valve, for example?** The figure has been simplified by removing elements that are not relevant for this study. A legend has been added for clarity and the caption of figure 3 has been extended as below:

[Figure]

Gas inlet system of the SICAS with one VICI multivalve inlet port, connected to three high pressure natural air tanks and 12 free ports for samples. The includes an extra inlet port for the machine working gas, also a high pressure natural air tank.

**L98: mbar not mBar, and please specify this is gauge pressure, not absolute.** It is absolute pressure, now indicated in the text and updated to more recent user values. We did not see any influence of the sample pressure, except when too low or too high (very slow filling times and low repeatability of the cell pressure, respectively) so this range is suitable. *Page 6 line 112:"For the cylinders, single stage pressure regulators are in use (Rotarex, model SMT SI220), set at an outlet pressure of 600-1000 mbar (absolute)."*

**L109 and Fig 4. What are the units of the Y axes of plots? For the upper plots they should be ppm2 but I cannot relate this to the mimimum figures given in the in-plot tables. The lower plots are presumably $\%^2$ . Also the abbreviation "St. dev"; these are not standard deviations, they are the minimum Allan deviations (ie sqrt(AllanVariance)), which is strictly not quite the same thing. Finally, unless the amounts for the isotopologues are scaled by their natural abundances, the Allan variances in ppm2 will have quite different absolute values.** The units of the y-axis of the upper figures are indeed $ppm^2$, now added in the axis title, and the units of the y-axis of the

lower figures were in the original manuscript the absolute deviation of the measured ratio. We adjusted this to ‰$^2$ for clarity by multiplying the values by 1000. The adjusted figure is shown below.

[Figure]

The Allan variance as a function of the integration time in seconds for a single gas measurement plotted for both the measured isotopologue abdundance (top) and the isotope ratios (bottom) at September 2017 (left) and July 2019 (right). The best achieved precisions and corresponding integration times are shown as a table in the plots.

**L143: please replace "n" with "number of measurements"**

**L143: should "lower" read "higher"?** *Page 6 line 159: "It is expected that, if the drift correction is effective, the standard deviation does not get worse with a higher number of measurements, and that the standard deviations of the uncorrected values are higher than the corrected values."*

**L129: what is "demi-water"** Demi-water is replaced by ultrapure water.

**L152: This is not a fair comparison, or is a misleading statement. It is only to be expected that ratios will show lower Allan Variance than individual isotopologues because some sources of noise and drift in the ratioed quantities are correlated and cancel. When**

**isotopologue amounts are ratioed later in a calibration calculation, as in the IM, these correlated noise sources will also cancel. This should be made clear.** We realise this, and therefore we decided to only show the results of the ratios, as in the end, all amounts are ratioed later.

**Table 1: These are relative standard deviations in ‰ please add "relative" to the caption and make clear in the case of the ratios that it is not the delta value itself but the relative std dev of it.** This has been adjusted in the table and in the caption, as shown below:

| All st. dev. in ‰ | n=5 | | n=10 | |
|---|---|---|---|---|
| | uncor | cor | uncor | cor |
| r636 | 0.036 | 0.020 | 0.055 | 0.025 |
| r628 | 0.046 | 0.021 | 0.104 | 0.029 |
| r627 | 0.060 | 0.018 | 0.177 | 0.031 |

Relative standard deviations for n=5 and n=10 of uncorrected (uncor) and corrected (cor) isototope ratio sample measurements. Sample measurements were always bracketed by measurements of the working gas. Standard deviations of the uncorrected measurements only use the sample measurements, standard deviations of the corrected measurements use drift corrected (equation 1) sample measurements using the working gas measurements.

**L180, eq. 3. Although this equation is only used for illustrative purposes, it is confused by the true calibration equation (10). The difference is between XCO2 and Xa. XCO2 is itself a function of X\* (or Xm) through the relevant isotope ratios so the coefficients a here and c in Eq 10 are different. It would be clearer to use the same equation as (10) here.** In the adjusted manuscript there is not used a linear fit for the calibration of the isotopologue measurements, but a quadratic fit. We assume, therefore, that the difference between the calibration equation in equation 8 and the illustrative example in equation 3 is clear. To avoid double use of a and b, the a and b in equation 3 were adjusted to the greek symbols $\alpha$ and $\beta$.

**L214: Griffith2018 showed that a pseudo linear + inverse relationship to CO2 amount is both theoretically expected and fits data in that paper in practice. The quadratic fit is only an approximation to this.** *Page 10 line 226 "Griffith (2018) showed that a combination of a linear and inverse relationship to the $CO_2$ mole fraction is theoretically expected, and this relationship fitted the data used in his study in practice. As we expect to have a relation of the measured delta values and the $CO_2$ mole fraction which is close to linear, we use a quadratic relation which approxi-*

*mates this expected theoretical relation closely. "*

**Section 3.2.1 and Fig 5. I have difficulty following this section and figure. It may be better titled "Spectroscopic non-linearities of isotope measurements". If the "residuals" are the differences between a linear regression of X\* against X626, they should be in units of ppm, not in Fig 5. I do not understand the sentence on L 223 "The CO2 mole fraction..."** The title has been changed as suggested. The residuals are calculated as the difference between a linear regression of $X_*$, expressed as $M_{S(t)dc}$ for drift correction, against $X_6 26$. $M_{S(t)dc}$ is in fact a ratio, which makes it, in our opinion logical express the residuals in permil and not in absolute values. The sentence L223 in the original manuscript has been rephrased to: *Page 10, line 237: "The $CO_2$ mole fraction is calculated by multiplying $M_{626(t)dc}$ by the known $CO_2$ mole fraction of the working gas."*

**L249: There should be no apostrophe in "deltas". An apostrophe is used to indicate possessive, not plural. There are many examples through the MS.** This has been adjusted throughout the text.

**Fig 6. See comment for L214, it would be interesting to see these data plotted against XCO2 and 1/XCO2. Does this improve the comparisons in Table 3? The description of this table is also difficult to follow in the text.** Data were plotted agains XCO2 and 1/XCO2, showing very similar. See the table below where the new fit results were added as fit inv:

| all values in ‰ | | $\delta^{636}$ | $\delta^{628}$ | $\delta^{627}$ |
|---|---|---|---|---|
| 4\*exp.1 (404-1025ppm) | lin | 0.871 | 0.120 | 0.376 |
| | q | 0.072 | 0.142 | 0.100 |
| | fit lin | 0.141 | 0.090 | 0.169 |
| | fit q | 0.034 | 0.092 | 0.078 |
| | fit inv | 0.063 | 0.090 | 0.092 |
| 4\*exp. 2 (313-484ppm) | lin | 0.095 | 0.181 | 0.095 |
| | q | 0.054 | 0.164 | 0.097 |
| | fit lin | 0.086 | 0.175 | 0.093 |
| | fit q | 0.049 | 0.155 | 0.093 |
| | fit inv | 0.054 | 0.017 | 0.095 |
| 4\*exp. 3 (426-522ppm) | lin | 0.075 | 0.186 | 0.048 |
| | q | 0.084 | 0.162 | 0.032 |
| | fit lin | 0.093 | 0.191 | 0.037 |
| | fit q | 0.082 | 0.161 | 0.028 |
| | fit inv | 0.085 | 0.164 | 0.029 |

The XCO2 and 1/XCO2 fit does not show significant improvements, so we decided that the quadratic fit is a good approximation for the theoretically expected relationship of the delta as a function of the XCO2. *Page 14 line 274: "The theoretically expected combination of a linear and inverse relationship as described in Griffith (2018) showed very similar results as the quadratic fit results, so we consider the quadratic fit to be a good approximation of the theoretically expected relationship."*

**L245: Please clarify the sentence "From now on we use the with an isotopologue superscript...." and/or give an example. It is quite unclear to me.** Has been removed, was unnecessary.

**L256-283: This section is important in the context of general comments above on correcting for stability, non linearity and drift. It points out that the calibration changes with time, but my understanding of the analysis is that this is compensated in RM but not IM. This might be clarified in dealing with earlier comments on the IM in 3.3.2.**

**3.3.2 Isotopologue method. Please see earlier general comments on this section. This does not have the same level of detail as 3.3.1 and if I understand it correctly the IM does not include the equivalent of the interleaved comparison with a reference gas R, or any correction for changes in the coefficients c and d in Eq 10. These changes are the root of any calibration drift in both RM and IM . As this section is written they appear to be corrected in RM but not IM.**

Calibration changes over time are dealt with for both the RM and the IM. This has been clarified in the text after earlier comments.

Sincerely,

Farilde Steur

---

## Author Comment (AC3) · 19 Feb 2021

February 19, 2021

Response to E.. Flores's comments for the manuscript "Simultaneous measurement of  $\delta^{13}C$ ,  $\delta^{18}O$  and  $\delta^{17}O$  of atmospheric  $CO_2$  -Performance assessment of a dual-laser absorption spectrometer

We want to thank Edgar Flores for reading the manuscript and for the comments he made which helped us improving the manuscript, and especially in defining the uncertainties of the measurements.

In this document we will address all the points that were raised. We use **bold text** for repeating the points of the reviewer, normal text for our answers, and *italics text* for changes made in the manuscript. Page and rule numbers apply to the new version of the manuscript.

**Technical comments**

1) Page 4, line 99: Please provide details on the type of materials, treatments on lines and values to transfer the gas to the SICAS instrument. Is the gas cell under constant gas flow? If yes please provides the flow rate. We extended this section to provide more details. Page 4 line 99: " The gas inlet system, depicted in figure 3, is designed to measure discrete air samples in static mode, such that one can quickly switch between measurements of different samples. The system consists of Swagelok stainless steel tubing and connections and pneumatic valves. There are two inlet ports (11 and 14) which are connected to the sample cross at the heart of system (from now on indicated as inlet volume). where a sample is collected at the target pressure of  $200\pm 0.25$  mbar before it is connected to the optical cell. One of the inlet ports (11) is connected to a 1/8" VICI multivalve (Valco Instruments) with 15 p otential positions for flask samples or cylinders. The cylinders depicted in figure 3 will be defined in section 2.2 and 3.2. When the VICI valve switches from position, the volume between port 10 and 9 is flushed 7 times with the sample gas to prevent memory effects due to the dead volume of the VICI valve. "

2) Page 5, line 91: Describe in detail the flushing procedure to avoid cross contamination providing evidence that it works (supplementary information). The flushing procedure has been described in the section that is showed after comment 1. We extended the section on cross-contamination slightly and provided the results of the analysis in the Appendix. Page 8 line 164:"Cross-contamination, being the dilution of a small volume of the working gas in the sample aliquot that is being measured, and vice versa, as described for a Dual-Inlet IRMS in Meijer et al. (2000), will occur in the SICAS due to the continuous switching between sample and machine working qas. If cross-contamination is not corrected for DI-IRMS measurements inaccuracies can occur when samples of a highly deviating isotope composition are measured. On the SICAS only atmospheric samples are measured that are of very similar isotope values. The  $CO_2$  mole fraction of the samples can deviate quite strongly from the machine working gas, so effects of cross-contamination will have an influence on the  $CO_2$  mole fraction in the optical cell. From experimental data we quantified the fraction of the preceding sample that affects a sample measurement to be max 0.01%. A sensitivity analysis was performed using this fraction and showed that this is such a small amount that scale effects due to cross-contamination are well below the precisions found in this study (for a detailed description of the analysis, see Appendix E). If samples of  $CO_2$  concentrations outside the range of atmospheric samples are measured it will be essential to also take into account the surface adsorption effects of the aluminum cell which is known to absorb  $CO_2$  Leuenberger et al., 2015).  $CO_2$  absorption in the cell of the SICAS was clearly visible as a drop of measured  $CO_2$  concentration when an atmospheric sample was let into the cell right after the cell was flushed with a  $CO_2$  free flush gas (hence stripped from  $CO_2$  molecules sticking to the cell surface). "

3) Page 5, line 97: Please explain how you took into account the changes in the internal pressure of the cylinders and their potential fractionation effects? Instrument performance The internal pressure of the cylinders has been well above 50 bar during the whole study period shown in this paper. This made us confident that no measures had to be taken to prevent fractionation due to cylinder depletion, as this is known to show pronounced effects below 4 bar. Moisture inside the cylinder is known to cause instability in the isotope composition of  $CO_2$  in the cylinder (Socki et al., 2020). All cylinders were evacuated overnight before filling and were filled with dried air. While working with low sample consumption and at high internal cylinder pressure, we did not evaluate potential fractionation effects thoroughly. For future use it would be interesting to evaluate the potential fractionation effects due to storing cylinders in vertical position, and consider to store long-lasting reference cylinders in horizontal position. Although essential for long-term measurement stability, we consider cylinder treatment out of the scope of the paper, which is mainly on evaluating the measurement performance of our laser absorption spectrometer and the calibration method for determining the triple stable isotope composition.

4) Page 6, line 110: "machine working gas" is not defined. In Figure 3. The term Ref/Working gas is used. Is it the same? The term machine working gas is replaced by working gas, which is defined in section 2.2. In section 3.2 the other cylinder names are defined.

5) Page 6, line 116 is stated: "The precision became significantly worse for all species but isotopologue 627 in the time period between September 2017 and July 2019 due to a gradual but significant decrease (of about 50And in Page 6, line 127 is stated: "Hence, we were able to clean the mirrors and retrieve..Questions: o When exactly were the mirrors cleaned? o Which measures were affected? o What was the short and long term effect on the measurements with a timeline instrument response? o What was the effect of introducing moisture and ethanol for the cleaning of the mirrors, short and long term effect? The cleaning procedure took place at the  $31^{st}$  of October 2019, meaning that the measurements presented in the results and discussion section took all place after the procedure and we cannot relate any measurement instabilities from those long-term results to the change in laser signal. Note that that all results presented in section 2 and 3 are from the period before the cleaning procedure. The measurements did not improve due to the cleaning procedure, indicating other issues played a role, maybe already before the cleaning procedure, or as a result of the cleaning and following realignment procedure. We added this extra information in the text. This section was added to the manuscript to provide the reader with extra information on potential reasons for mirror contamination, being most likely the cause for the observed decrease in laser intensity. Presenting the effects of contamination and finding the best way to clean the mirrors is, however, not our aim. Page 7, line 144: "This procedure, performed at the  $31^{st}$  of October in 2019, deviates from the recommended mirror cleaning instructions in which it is advised to use ethanol only to clean the mirrors. The additional use of distilled water was in our case necessary since the precipitated aerosols were not dissolved in ethanol and were therefore not removed when we used ethanol only. Despite the increase of the laser signal due to the cleaning procedure, precisions did not improve as a consequence of it. This indicates that other, still unidentified, issues played a role in the decrease of measurement precision."

6) Page 9, line 199: the three experiments performed over the last two years means: before or after the cleaning? All three experiments presented in this section were performed before the cleaning procedure.

7) Cross contamination and drift are only considered as uncertainties components on the instrument performance. However the instrument was used to measure cylinders and flask that could have important differences in matrix composition. What was the pressure broadening effect on the CO2 measurements? All our cylinders and flasks, also the ones that were prepared by dilution of  $CO_2$  with  $CO_2$  free air, are natural air mixtures. We do not use any synthetic reference or sample gases, except for the samples presented in table 4, which shows the results of an experiment to test possible effects of differences in matrix composition. The composition of natural air is very stable, except for trace gases as  $CO_2$ ,  $CH_4$  and  $N_2O$ , etc.. Differences in  $CO_2$  mole fractions are taken into account in the calibration schemes. We typically collect our reference gases at near background, or slightly enhanced trace gas concentrations. Therefore we consider the other trace gases of such low mole fractions that we consider the air matrix to have a negligible influence on our isotope measurements.

8) What is the real contribution of various components of the air, temperature and pressure variability into the instrument? Uncertainty analysis is missing. We show measurement stability over a whole measurement sequence (12 hours), and we show the effect of measuring a working gas alternately with every sample gas measurement. We realise that more insight in the repeatability and accuracy, as well as uncertainty effects of our reference gases used will help the reader getting insight in the performances nad the potential of this instruments. We therefore added section 4.2 in which we show how we derive the combined uncertainty of our measurements. Although insight in the real contributions of parameters that were given here by the reviewer are of great importance as this would help in the development of an instrument of higher measurement quality, we want to present the measurement and calibration quality as users of the instrument instead of developers. We therefore think that presenting the measurement errors and repeatability is enough to give readers an idea about the potential of the measurement device for the application for atmospheric isotope research. Page 20, line 454: "A combined uncertainty consisting of measurement uncertainties and scale uncertainties is calculated for the sample measurements. Measurement uncertainties include the standard error of the sample measurement, the repeatability of all (usually four) measurements of the quality control gas throughout the measurement sequence, and the residual of the mean of the quality control gas measurements from the assigned value. The measurement uncertainties will therefore vary with each measurement/measurement sequence. We observe a high repeatability of all sequences included in the analysis of figure 7 (8 in total); with standard errors ranging between 0.005 and 0.03‰ and a mean of 0.014‰ for  $\delta^{13}C$ , and standard errors ranging between 0.011 and 0.04% and a mean of 0.012% for  $\delta^{18}O$ , for both methods. The residuals in these sequences show a higher contribution to the combined uncertainty and a small difference between the two calibration methods. The absolute residuals of the RM range between 0.0009 and 0.07%with a mean of 0.026% for  $\delta^{13}C$ , and between 0.007 and 0.06% with a mean of 0.04‰ for  $\delta^{18}O$ . For the IM the residuals range between 0.002 and 0.05‰ with a mean of 0.023‰ for  $\delta^{13}C$ , and between 0.012 and 0.05‰ with a mean of 0.03\% for  $\delta^{18}O$ . Hence, the RM shows slightly higher contributions to the combined uncertainty as a result of the accuracy of the quality control gas measurements. The scale uncertainties, which are fixed for all measurement sequences in which the working gas, low reference and high reference are used as calibration gases, were simulated using the Monte Carlo method. Input values were generated by choosing random numbers of normal distribution with the assigned value and uncertainty as in table5 being the mean and the standard deviation around the mean, respectively. As the RM and IM follow different calibration schemes, the Monte Carlo simulations are discussed separately; for the RM the scale uncertainties of the assigned delta values result in an uncertainty in the calculated residuals which are quadratically fitted against the measured  $CO_2$  mole fraction. The average uncertainties in the calibrated delta values of the 5 simulations are 0.03 and 0.05% for  $\delta^{13}C$ , and  $\delta^{18}O$ , respectively. Besides the uncertainties introduced by the scale uncertainties of the delta values, the calibrated measurements of the IM are also affected by the scale uncertainties of the  $CO_2$  mole fractions. Both the uncertainties in the delta values and in the  $CO_2$  mole fractions affect the calculated assigned isotopologue abundances, which are quadratically fitted against the measured isotopologue abundances. The uncertainties in the assigned delta values result in average uncertainties of 0.03 and 0.06\u03 for  $\delta^{13}C$  and  $\delta^{18}O$ , respectively. The uncertainties in the assigned  $CO_2$  mole fractions result in uncertainties of 0.005 and 0.018% for  $\delta^{13}C$  and  $\delta^{18}O$ , respectively, and are small compared to the uncertainties of the assigned delta values. Reducing the combined uncertainty of the  $\delta^{13}C$  and  $\delta^{18}O$  measurements of the SICAS will be most effective by determining the isotope composition of the reference gases with a lower uncertainty on the VPDB- $CO_2$  scale.

9) Page 8, line 160 is stated: "A sensitivity analysis was performed and showed that this is such a small amount that scale effects due to cross-contamination are well below the precisions found in this study" Where is the sensitivity analysis? This an important effect and evidence should be shown to sustain this statement. Please see Appendix E and experimental data. 10) How the uncertainty of the method is constrained by the uncertainty in the reference values of the CO2 mole fractions in the calibration standards? Please see the added section 4.2, as already showed in point 8.

11) What is the effect of diluting pure CO2? Uncertainty in the mole fractions related to this? What is the contribution of the loss of CO2 on the wetted surfaces and the emptying of the flasks? We did not measure the absolute  $CO_2$  values of the diluted samples, as we use in the analysis the measured  $CO_2$  mole fractions. The dilution process was done, keeping in mind the range of  $CO_2$  mole fractions that we would like to test, not trying to produce samples of a  $CO_2$  mole fraction with high accuracy. Flasks that were used for the CMFD experiments were dry, we made sure to evacuate the glass system that was used for the preparation of the samples for at least one night before starting the procedure. What the uncertainties are exactly is hard to say, and would only be possible to check by doing a so called closed loop experiment: diluting a well-known pure  $CO_2$  reference gas to atmospheric concentrations, extract it again and measure it to check whether any fractionation occured. We did not include this work in this paper, as this is still ongoing work.

12) Page 10, line 224: "..the known CO2 mole fraction of the working gas..": "mole fraction" is the quantity referred to in the paper. The proper unit would be *mumol/mol* rather than ppm. We added an extra sentence to the introduction to clarify that we express the  $CO_2$  mole fraction in  $\mu mol/mol$ , also referred to as ppm. Page 3 line 70: We report  $CO_2$  mole fractions in  $\mu mol/mol$ , also referred to as ppm.

13) Figure 5 shows the experiment 1,2 and 3 that correspond to different measurements carried out in 2017, 2018 and 2019. The authors stated in page 6, line 116 that "the precision became significantly worse for all species but isotopologue 627 in the time period between September 2017 and July 2019 due to a gradual but significant decrease (of about 50 percent) in the measured laser intensity over that period". Why do the error bars in 2017, 2018 and 2019 measurements look the same for the mole fraction range 400 to 460 ppm? The statement applies to figure 4, in which it is clearly visible that measurement precision of a single gas measurement gets worse over time. The figure the reviewer is referring to is figure 5, in which the mean values of drift corrected measurements will introduce uncertainty as well. Apparently, the reduction of the quality of a single gas measurement,

which was probably caused due to reduction of the laser intensity, is of less importance than the repeatability of the measurements.

14) Page 12, Figure 5: Why there no uncertainty bars in x-axis? Where are the confidence bands that could support the statements? Typical measurement uncertainties of  $CO_2$  concentrations of the SICAS are  $\sim 0.2$  ppm. These uncertainties would fall within the size of the marker and would therefore not be visible. The fits that are showed here have the aim to show the non-linearities of the rare isotopologue abundances as a function of the abundant isotopologue, not to quantify these relations exactly.

15) Page 14, line 269: is stated "In our lab CO2 in air samples of the same isotope composition but deviating CO2 mole fractions are prepared manually, introducing again uncertainties, and doing these experiments regularly is therefore labor- and time intensive". However no uncertainty budget (assessment) considering all the uncertainty contributors for the manual preparation is shown. As a minimum, it is likely that air composition affects CO2 measurements and this is recognized latter in the text, line 291. We used natural air, only scrubbed from  $CO_2$  using Ascarite for the preparation. We assume therefore that the effect of air composition will be negligible, see also my answer on point 8.

16) Page 14, line 275: the reference cylinders are mentioned for the first time in this section but those are not identified (serial number) neither their composition (air matrix). The same issue with the air samples used. We don't see the need for providing the serial numbers as we identify the tanks by name, not by serial number. As we already elaborated in point 8 all our reference cylinder contain dried natural air.

17) Page 14, line 285: A brief description of the gaseous reference materials is given in this section 3.2 with additional information page 15, line 302 and Appendix B. Nevertheless it is essential to have as much as information as possible on the reference materials for allowing the readers to reproduce such experiments. All the reference materials that we use are dried natural air mixtures. Please see also our answer on point 7.

18) Page 14, line 287: The author specify that "two tanks that are specifically used for CMFD corrections. These latter two consist of a high mole fraction reference tank (HR) and a low mole fraction reference tank (LR) covering a great part of the CO2 mole fraction

range occurring in atmospheric samples" Such range is: 342.8 to 423.77 ppm. However in Page 9, line 195 the author state: The SICAS is designed for the measurement of atmospheric samples of which the relevant range of CO2 mole fractions is 370 - 500 ppm, and experiments were therefore for the most part conducted in this range. The author can only warrant that the CMFD corrections of the instrument following the method proposed is valid for the range 342.8 to 423.77 ppm, not higher since there is no evidence for that. We agree with this statement. In the modified manuscript measurements that are done outside the range are left out of the analysis with the argument that extrapolation of calibration curves should at all times be avoided. One of our main conclusion is that extending the range of the reference cylinders will make the instrument better suitable for measuring the isotope composition of the whole range of atmospheric  $CO_2$  mole fraction samples. We sentence in which the range of 370-500 ppm is mentioned applies to the experiments on determining the CMFD of our instrument, in which no calibration, only drift correction, is applied. The sentence has been rephrased to make this clear. Page 10 line 211:"The SICAS is designed for the measurement of atmospheric samples of which the relevant range of  $CO_2$  mole fractions is  $\sim 370 - 500$  ppm, and CMFD experiments were therefore for the most part conducted in this range."

19) Page 15, line 317: It will be important for the robustness of this paper to list, and assign an experimental uncertainty, to each of the uncertainty sources cited in this section "small leakages or other gas handling problems might be introduced". As per reference gas five flasks were sampled, gas handling problems and small leakages will appear in the combined uncertainty which is given in table 5. As uncertainties are rather small, we don't think gas handling problems would have caused major problems. We are mainly interested in the total uncertainty of our measurements, as the overall uncertainty is acceptable we are less interested in the exact contribution of uncertainty of the filling process.

20) Page 15, line 304. The claimed standard uncertainties on the gas tanks that were produced in-house from dry compressed natural air, "HR 423.770.01 ppm" and "LR 342.810.01ppm", are very unlikely (considering that Y = y U). During the CCQM K120.a international comparison only one laboratory submitted comparable uncertainties (NMIJ) but considering potential effects of adsorption of a proportion of the molecules onto the internal surface of a cylinder and valve a limit on the uncertainties claimed by participants contributing to the reference values on this comparison was fixed to 0.095 mol mol-1 meaning that any uncertainty claim smaller to this value was replaced by 0.095 mol mol-1 to calculate reference values. The claimed standard uncertainties from the gas tanks in this work are even half of the NOAA (real air mixtures) submitted uncertainty for the CCQMâAR K120.a international comparison. These values were indeed incorrect. The 0.01 ppm uncertainty applies to the measurement uncertainty, while the scale uncertainty was not considered, while very important for determining the calibration uncertainty. We expanded the text in section 3.2 and changed the uncertainties to the correct values after this comment of the reviewer. Page 15 line 330: The  $CO_2$  mole fraction of the tanks was measured on a PICARRO G2401 gas mole fraction analyzer and calibrated using in-house working standards, linked to the WMO 2007 scale for  $CO_2$  with a suite of of four primary standards provided by the Earth System Research Laboratory (ESRL) of the National Oceanic and Atmosphere Administration (NOAA). The uncertainty of the WMO 2007 scale was estimated to be 0.07  $\mu$ mol/mol-1. The typical measurement precision of the PICARRO G2401 measurements is 0.01  $\mu$ mol-1 resulting in a combined uncertainty of 0.07  $\mu$ mol/mol-1 for the assigned CO2 mole fraction values of the calibration tanks, while difference between the two cylinders is known with a much lower uncertainty.

21) Page 16, Table 5. define what are CO2 err, 13C st. err., 18O st. err. and their units. It appears very unlikely that calibrated reference materials of CO2 in air are provided with an uncertainty equal to 0.01 mol mol-1. Therefore CO2 err is most likely a precision. Please justify why the precision is the only uncertainty contributor that maters by proper considerations on the goal of the experiment. We agree, see our comments after point 20.

22) Page 17, line 368: The statement "The mole fraction (X) of the four most abundant isotopologues of a measured CO2 sample are determined using two references gases with known CO2 mole fractions and isotope compositions" must be completed by including: - A phrase stating that the measurements are only valid for the range of 342.81 ppm to 424.52 ppm (according to Table 5, page 16) since the standards must bracket each of the three expected isotopologue mole fractions in the samples; We agree, and added therefore the following phrase: Page 19 line 416: "The CO2 mole fractions are ideally chosen such that normally occurring CO2 mole fractions in atmospheric air are bracketed by the two reference gases. The low and high reference gases cover the range between 324.81 and 424.52 ppm, meaning that this method is only valid for samples within that range of  $CO_2$  concentrations. "

- an uncertainty budget including at least two components, the first related to the repeatability of the measurement results (MPI-Jena), and the second related to the stability and homogeneity of the isotope ratio values in different standards containing nominally the same CO2 gas; The first component is now included in the manuscript, see section 4.2. The second component is very important for long-term measurement stability, however, the reference gases have not been in use for such a long time that we have sufficient information to answer this question. We will, of course, do a re-calibration of our reference gases in the near future, and we will be able to address this issue accordingly.

- It is also crucial when proposing a calibration procedure to consider the uncertainty of the calibration standards. The uncertainties on the reference isotopologue mole fraction values need to be first estimated, and then used in an uncertainty budget reflecting a two point calibration process for each isotopologue. We added section 4.2, in which we consider these uncertainties. See also our answer on point 8.

23) The term "reference gases" is not clear into the document. It is first mentioned in - Page 17, line 369; "The CO2 mole fractions are chosen such that normally occurring CO2 mole fractions in atmospheric air are bracketed by the two reference gases" - in Page 17, line 348: ": We developed a calibration method based on the idea that including the measurement of two reference gases covering the CO2 range of the measured samples (in our case LR and HR) enables the correction of the measured isotope ratios: ",... then; - In Page 17, line 371: "...Due to the broad range of CO2 mole fractions that are covered by the reference gases, measurement of both working standards will enable the calculation of the (linear) relation of the measured mole fraction (Xm) and the Xa,..." .. then in; - Page 24, line 506: "...we used natural air as reference gases (or air mixtures close to natural air)" - but then in Page 5, Figure 3: Ref 1 and Ref 2 appears without being defined elsewhere in the document and the Ref/Working gases term as well. So are Ref 1 and Ref 2, the reference gases? Same as listed in Table 5 as LR and HR (Page 16)? Which gases are the Ref/Working gases? Please define; reference gases, measured samples, reference cylinders, calibration gas mixtures, inC ask samples, HR cylinder, LR cylinder, Ref orking gases and machine working gas. We now define the term reference gas as: Page 14, line 291: In the daily procedure of the SICAS there are at least two  $CO_2$ -in-air reference gases (in short reference gases), high pressurized cylinders containing gas of known isotope composition and  $CO_2$  mole fraction..." The term working gas is defined earlier as: Page 6 line 129: "The cylinder used for drift correction which we define as the working gas contains natural air of which the isotope composition and the  $CO_2$  concentration is known."

Form these definitions one can conclude that the working gas is a reference gas.

We decided to not use the abbreviations WH, LR and HR in the text for clarity. Instead we use the, earlier explained term working gas, and low reference and high reference. Low and high reference are defined as: Page 15 304: "two tanks containing a high mole fraction reference gas and a low mole fraction reference gas, from now defined as the high reference and the low reference, which can thus be used for CMFD corrections. The high and low reference cover a great part of the  $CO_2$  mole fraction range occurring in atmospheric samples."

Also the terms used in figure 3 were changed to the terms defined above.

24) Page 17, line 370: Is Appendix 1 or Appendix A? Changed to Appendix B.

25) Page 18, Table 6. No units? Has been added, as well as in table 7,8 and 9.

26) Page 19, line 407: when was the sausage series 90-94 measured in 2020 (month exactly?) Page 22 line 490: "SICAS measurements took place in the period from December 2019 to April 2020 As described in Page 20, line 434. There was an important long term effect of the aliquot storing during 3 to 20 months for 180 measurements and this could strongly influence the conclusions of this intercomparison. Because of the observed effect of drifting oxygen isotope values cannot compared the  $\delta^{18}O$  measurements, and only conclusions are drawn for the  $\delta^{13}C$  measurements as no observed drift was observed for the  $\delta^{13}C$  values.

27) Page 20, Table 7. No units? Why comparing the difference SICAS-MPI with NOAA-MPI? Which technique was used by NOAA? Units are added to table 7. Comparing the difference SICAS-MPI with NOAA-MPI has the aim to put our isotope measurement results into context of the performance of other labs. As NOAA is known to have a good reputation in stable isotope measurements of atmospheric samples, we think showing both comparisons: SICAS-MPI and NOAA-MPI gives a good idea on how we are doing in comparison with expected results from the stable isotope community.

28) Page 22, line 440: figure 8 shows results outside the mole fraction range validated in this work (343-425ppm). No conclusions can be stated for measurements outside the range. We agree, the results out of the range are kept out of the analysis with NOAA, and the figure has been adjusted so it is clear to the reader which measurements are out of the range.

29) Page 24, line 491: The author state "In this study we show that WMO compatibility goals can be reached with our Aerodyne duallaser absorption spectrometer for stable isotope measurements of atm-CO2 in dry whole air samples if the instrumental conditions are optimal and there is no uncertainty induced because of gas handling procedures (flask sampling for instance)" but unfortunately no uncertainty budget was shown in the paper to underpin this statement. We agree that this statement only applies to the measurement precision and accuracy of our quality control gas measurements, and not to the combined uncertainty. We therefore do not claim to reach WMO compatibility goals but give values for all components of the uncertainty analysis in the conclusion. Page 27, 592: In optimal measurement conditions, precisions and accuracies of < 0.01 and < 0.05% for  $\delta^{13}C$  and  $\delta^{18}O$  are reached for measurements of the quality control tank for both calibration methods. The combined uncertainty of the measurements includes also the repeatability of the four quality control gas measurements throughout the measurement sequence, with mean values of 0.014 and 0.012%. The last components in the combined uncertainty calculation are caused by scale uncertainties of the reference gases used for the sample calibration, which are 0.03 and 0.05%for  $\delta^{13}C$  and  $\delta^{18}O$  of the RM, respectively and 0.03 and 0.06% for  $\delta^{13}C$  and  $\delta^{18}O$  of the IM, respectively. "

30) Page 24, 496: The author state "Non-linear dependencies on the CO2 mole fraction occur for measured isotopologue abundances but are insignificant in the typical ambient CO2 mole fraction range" This is relatively true if compared to the compatibility goals. We do not state this anymore, as we see that doing a quadratic fit improves the accuracy of the results for both methods.

31) Page 24, line 510: The author state "From studying the results of the QC we conclude that precisions are significantly better for the RM, while measurement stability is very similar, both for the 13C and the 18O measurements.". With the current version of the paper, there is no evidence to show that the isotopologue method was applied correctly, using proper reference materials and proper measurement sequences which would ensure frequent drift corrections and two points calibration for each isotopologue. If the isotopologue method is kept after revision, and if the same observations are made, consider explaining why the two methods provide different precisions. The IM is revised after comments of D. Griffith, and we now see similar precision results as for the RM, as one would indeed expect. Only the precision calculation was adjusted. We think the application of the IM was already correct, as we apply continuous drift corrections and all isotopologues were calibrated with a 2-point calibration curve. We now adjusted that to a 3-point, quadratic calibration curve which improved the measurement results of the IM more.

Sincerely,

Farilde Steur

References:

Richard Socki, Matt Matthew, James McHale, Jun Sonobe, Megumi Isaji, and Tracey Jacksier: Enhanced stability of Stable Isotopic Gases, ACS Omega, 5(29), 17926-17930, 10.1021/acsomega.0c00839, 2020

---

## Author Response (AR2)

Response to final comments Associate Editor Tim Arnold on manuscript Steur et al., 2021

We want to thank Tim Arnold for his comments on the manuscript, as well his comments for earlier versions of the manuscript. Please see below a point by point answer on the comments. In grey the comments, in black the responses.

Sincerely,

Farilde Steur

**Point by point answers to comments:**

L10 - 'methods' should read 'calibration methods'

Changed as suggested.

L11 - I agree with the referee report regarding the phrase 'optimal conditions'. This implies that the steps to reach these conditions are well know, but I don't think this is the case. Instead of 'optimal..' suggest 'During one specific measurement period..'

Line 12: "During one specific measurement period the precision and accuracy of the quality control tank reach WMO compatibility requirements, being 0.01‰ for δ13C and 0.05‰ for δ18O, respectively. "

~L70 - there is reference to chapters - please correct

Line 68: "In this paper we present the performance, in terms of precisions and accuracy, of an Aerodyne dual laser optical spectrometer (CW-IC-TILDAS-D) in use since September 2017, for the simultaneous measurement of δ13C, δ18O and δ17O of atm-CO2, 70 which we refer to as "Stable Isotopes of CO2 Absorption Spectrometer" (SICAS). The instrument performance over time is discussed, followed by an analysis of the CO2 mole fraction dependency of the instrument. We report CO2 mole fractions in μmol/mol, also referred to as ppm. The actual ways of performing a calibrated measurement using either individual isotopologue measurements or isotope ratios is discussed and whole air measurement results of both calibration methods are evaluated for their compatibility with IRMS stable isotope measurements. Conclusively, the usefulness of the triple oxygen
75 isotope measurements for capturing signals of atmospheric CO2 sources and sinks is evaluated. "

Eq1: I don't really understand this equation. Mwg(t) is calculated and Ms(t) is measured. But Ms(t)dc can't be the ratio of these?

Ms(t)dc has been changed to Is(t) as suggested by D. Griffith, to clarify that the Is(t) is calculated from the Ms(t) and Mwg(t). Ms(t) and Mwg(t) are both measured values, while Is(t) is the calculated ratio from those two values.

L156-157: 'The effectiveness of this drift correction method was tested for the measured

isotope ratios, as for calibration of the measured isotopologue abundances to delta values, isotope ratios will always be used.' I can't make sense of this sentence - please revise.

Line 158: "The effectiveness of this drift correction method was tested for the measured isotope ratios only. Although one of the tested calibration methods uses isotopologue abundances for the initial calibration, the isotope composition is expressed 160 as a delta value and will therefore eventually be calculated using isotope ratios (section 3.3.3). The precision of the isotope ratios will therefore always determine the measurement precision."

Figure 5: Can you be clear what is actually being calculated in the residual - the use of per mil on the y axis is confusing if relating to amount fraction difference. Also 'darkblue' should be 'dark blue' etc.

Caption figure 5: "Figure 5. Residuals (expressed in ‰ relative to the measured amount fraction) of the linear fit of the rare isotopologue abundancies as a function of the X626 and the quadratic fit on the residuals. From top to bottom: Experiment 1, experiment 2 and experiment 3. The colours red, dark blue and light blue are used for the isotopologues 636, 628 and 627 respectively…."

Figure 6: In Figure 5 amount fractions go above 1000 ppm so why is that range not also obvious in Figure 6?

Not all points of experiment 1 are shown in figure 6 as this reduces the readability of the figure since the range of the other experiments is much lower.

285-295: mole or molar fraction? - be consistent throughout text

Changed to mole fraction throughout the text.

Figure 7's caption: H or M measurement period? Perhaps relate back to the text section where this is explained. Reader needs to be able to relate captions to the figures easily, or at least know where to find missing info.

Changed caption of figure 7: "…Colour of the points indicates whether the measurements were performed in a High quality (green), Medium quality (black) or Low quality (red) measurement period (see section 4.1 for definitions).

L458: I don't see 0.005 in the table - should these values correspond to those in the table?

The numbers in the text were not updated to the latest version of the table. This has been changed and now the numbers correspond to the table.

L583: 'We developed two different calibration methods based..' I think the methods are

developed elsewhere, implementation here is new. Perhaps 'we implemented two different calibration methods'

Changed as suggested.

Response to final comments D. Griffith on manuscript Steur et al., 2021

We want to thank D. Griffith for his final comments on the manuscript. The comments were used to adjust the manuscript to the final version as uploaded now. His thorough reading and critical review helped us to improve the manuscript significantly.

Please find the point by point answers below. In grey the points made by D. Griffith, in black the response.

Sincerely,

Farilde Steur

**Point by point answers to comments D. Griffith:**

L8: The references are also calibrated for mole fractions, which is required for the IM. Please specify that here.

Line 8: "Calibration with the ratio method and isotopologue method is based on three different assigned whole air references calibrated on the VPBD and the WMO 2007 scale for their stable isotope compositions and their CO2 mole fractions, respectively. "

L28: I am not convinced that the differences in quoted repeatabilities between the two methods IM and RM are really statistically significant. It would be better to avoid value judgements about which method is "better" and simply state the repeatabilities as observed under the measurement conditions.

Line 25: "The ratio method shows residuals ranging from 0.06 to 0.08‰ and from 0.06 to 0.1‰ for the $\delta^{17}O$ and $\Delta^{17}O$ results, respectively. The isotopologue method shows residuals ranging from 0.04 to 0.1‰ and from 0.05 and 0.13‰ for the $\delta^{17}O$ and $\Delta^{17}O$ results, respectively."

Are they really the "optimal" conditions (L11)?

Line 12: "During one specific measurement period the precision and accuracy of the quality control tank reach WMO compatibility re- quirements, being 0.01‰ for δ13C and 0.05‰ for δ18O, respectively. "

L58: FTIR has also been developed recently for this purpose, especially at BIPM as is referenced in this paper. Perhaps reword this as "Optical (infrared) spectroscopy now offers this possibility following strong developments in recent years in FTIR and especially for the laser light
Sources, …"

Line 58: "Optical (infrared) spectroscopy now offers this possibility following strong developments in recent years in FTIR and espe- cially for the laser light sources, to perform isotopologue measurements showing precisions close to, or even surpassing IRMS measurements (Tuzson et al., 2008; Vogel et al., 2013; McManus et al., 2015). "

Figure 2. From the plots it is clear that the linewidths in the calculated (fitted) spectrum do not match those measured, leading to the typical "second derivative" shaped residuals (and a position error in the case of the 636 line). Is this due to the accuracy of the Hitran widths, or the linewidth model employed in TDLWintel? Could the Aerodyne authors comment, perhaps in the Figure caption. It would be desirable to reassure the reader that the resultant errors in retrieved amounts are systematic and will calibrate out under identical measurement conditions, but they may lead to dependence on such conditions (eg. temperature, pressure) if they vary.

Added in caption of figure 2: "The residuals show systematic deviations at the line positions. These deviations are primarily due to the use of the Voigt lineshape function in the spectral fitting model, rather than a more complex lineshape function such as Hartmann- Tran. Careful analysis has shown that the use of the more convenient Voigt lineshape function does not add noise, drift or calibration error as implemented in the isotope analyser. "

L125: strictly speaking, the r18 and r17 isotope ratios differ from the 628/626 and 627/626 ratios by a factor of 2.

Added as a footnote to line 126: "Note that the r628 and r627 differ strictly speaking from the isotope ratios (r18 and r17) by a factor 2."

Figure 4: I cannot understand the relationship of the third column in the table (labelled std dev (‰)) to the Y axis on each figure, ppm2 or ‰2. I would expect this to be the square root of Allan deviation at the minimum, in units of ppm (upper plots) or ‰ (lower plots). These are sqrt(AV) at a particular averaging time, not strictly the same as standard deviations. But the numbers do not line up:
For example for 626 in the top left plot, the minimum AV is ~20 ppm2 at 16 s, so sqrt(AVmin) = 4 ppm. The given value in the plot table is 0.01 ‰. 4 ppm seems rather high (1% or 10‰ of 400 ppm) for the laser measurement.
Please explain and/or correct the numbers in the table

This was indeed incorrect due to use of the incorrect unit on the y-axis. The numbers of the y-axis were in ppb, while the axis title says ppm. I changed the numbers to the correct (ppm) unit, and now the numbers in the table correspond to the numbers in the graph.

Eq 1. It is confusing to use the same symbol (M) for the measured amounts (RHS) and the ratio to the WG measurement (LHS) – the former has the units of an amount of gas, the latter is dimensionless. Could you be clear in the text by using a different symbol for the

LHS, say M', or R for ratio, or I for index and referring to the appropriate quantity in the text strictly according to which is being used (for example L 235 and following).

The ratio has now been changed to the symbol I as suggested, and this has been adjusted for the use of it in the rest of the text.

L159 et seq: As stated in the text, the std dev of the corrected isotope ratio should be lower than that for the uncorrected ratio if the drift correction is effective, by an amount that depends on the drift, but only after taking into account that the random error in the ratio (M') will be sqrt(2) times larger than in either of the individual M values (so if there is no drift, the std dev of M' would be 1.4 times the std dev of M). But the statement line 160-164 that the std dev should not get larger with n is not correct – if the correction procedure were perfect the std dev should decrease by sqrt(2) or a factor of 1.4 from n=5 to n=10 measurements. In fact it increases in every case, so the drift correction is not perfect over that timescale. Why?

We show here the standard deviation, not the standard error, so it will decrease not with a factor of 1.4 from n=5 to n=10. However, it is right that in the case of a perfect drift correction, the standard deviation would decrease. This implies the drift correction is effective, but not perfect. This has been changed in the text. Line 165: "It is expected that, if the drift correction is effective, the standard deviations of the uncorrected values are higher than the standard deviations of the corrected values. The drift correction is effective as the standard deviations of the corrected values are always lower than of the uncorrected values. Although the drift correction procedure is not perfect, as we see a small increase of the standard deviation between n=5 and n=10 between 0.005 and 0.012‰, we can still conclude that the drift correction will result in a better repeatability of the isotope ratios."

Further, in Table 1 caption, are these really RELATIVE standard deviations (in ‰ of ‰), or are they actual standard deviations of the isotope ratios in ‰? This was changed from the original version. Please be very wary of using ‰ for a relative value of 2 quantities in isotope work because it is so easily confused with a delta value itself, also quoted in ‰.

These are relative standard deviations, which is now explicitly written in the caption.

L174, L1276: should this be adsorption, not absorption?

Yes, this was changed to adsorption throughout the text.

L188: please refer to comment on Figure 2 above where such a lack of fit is evident in the linewidths.

We now refer to the comments in the caption of figure 2, line 194: "Capturing the true absorption spectrum is very complicated, due to among others line broadening effects of the various components of the air, far wing overlap of distant but strong absorptions, temperature and pressure variability and the choice of lineshape function (see figure 2 and caption).

L197: should this read "when eq 3 is brought into eq. 2..."?
Yes, this has been adjusted.

L198: ... for either or both of those ...

Line 203: "When equations 3 is brought into 2 for either or both of the rare and the abundant isotopologue mole fraction, and $\beta$ is non-zero for one of those..."

L236: This should read either "The CO2 mole fraction is calculated by multiplying M'626(t)dc by the known M626(t)WG of the working gas" or more strictly "The CO2-626 mole fraction is calculated by multiplying M'626(t)dc by the known M626(t)WG of the working gas" (These will be the same for the same isotopic composition).

Line 234:"The 626 mole fraction is calculated by multiplying I626(t)dc by the known 626 mole fraction of the working gas. "

L258: the meaning is not clear – what quantity is "The WG(t)"

Line 264: "The $r^*_{WG(t)}$ is calculated using the same method as $M_{WG(t)}$ is calculated in equation 1. "

L263: "As measurements were not conducted on CO2 of similar isotope composition" Not clear, of similar composition to what?

Line 269:As the CO2 used for the different experiments was not of similar isotope composition, the δ636 measurements in figure 6 were normalized such, that at the CO2 mole fraction of 400 ppm all ratios are 1.

L330: The Picarro analysis is based on 626, not whole CO2 – there is an inherent assumption that the 626/totalCO2 ratio is constant. This is a potential source of error if the isotopic composition of different reference gases varies significantly, but from Table 5 the variation in isotopic composition is not significant for this error (Griffith 2018). However the narrow range of delta13C across the reference gases does not provide a wide span for calibration using the RM – for IM it does not matter. See L390.

Line 342: "The PICARRO analysis is based on the 626 isotopologue mole fraction, not on whole CO2. This is a potential source of error if the isotope composition of different reference gases varies significantly. As the isotope compositions of the used reference gases are close (see table 5), the variation is not significant for this error (Griffith, 2018). "

The comment on the isotope range of the reference gases used for the RM is correct, but this is covered in section 3.3.

L370, 371: Tans, Crotwell and Thoning (2017) also successfully implemented the IM and should perhaps be referenced here – it is used routinely at NOAA in the generation of reference gases.

Is added to the references.

L584: For the benefit of those reading only the abstract, intro and conclusions, please spell out Ratio Method and Isotopologue method here.

Is spelled out now.